# Online Mixture of Experts: No-Regret Learning for Optimal Collective Decision-Making

**Larkin Liu***
Technische Universität München
Riebaki AI
larkin.liu@tum.de

**Jalal Etesami**
Technische Universität München
j.etesami@tum.de

## Abstract

We explore the use of expert-guided bandit learning, which we refer to as online mixture-of-experts (OMoE). In this setting, given a context, a candidate committee of experts must determine how to aggregate their outputs to achieve optimal results in terms of aggregate accuracy. We propose two algorithms to address this problem. The first algorithm combines aggregate voting with UCB-driven successive elimination, efficiently pruning suboptimal exploration actions. The second algorithm employs an online weighted-majority-voting mechanism, leveraging the respective voting power of each expert proportional to their predictive power. We derive theoretical guarantees for the regret properties in the bandit setting under ideal circumstances, and empirical results are provided accordingly. As a modern study on applications, these methods are applied to the online fine-tuning of a set of expert large language models (LLMs), where after each response, the generative LLM dynamically reweighs its set of experts and/or selects the optimal committee of experts to generate the most accurate response. Our results introduce new methodologies and no-regret guarantees for combining multiple experts to improve on the performance of the an aggregate model overall.

## 1 Introduction

The mixture-of-experts (MoE) model is a powerful concept in applied machine learning and social choice, leveraging the collective decision-making capabilities of a group of experts to yield improved predictions or more computationally efficient models. The core idea is that within a collection of $N$ experts, $\mathcal{E}$, there exists at possible a subgroup, $\mathcal{E}^* \subseteq \mathcal{E}$, whose combined predictions are well-optimized on some well-defined metric (e.g. collective accuracy, model efficiency etc.). Classically, MoE models involved the design of a gating mechanism (a.k.a router) that aims to optimally transform the output of many experts into a single more optimal output. Traditionally, this involved training an offline model from labelled data to optimize some loss function via supervised learning [1, 2, 3]. Concerning neural architecture design for offline learning, the MoE framework has been wide integrated to several state-of-the-art LLMs (e.g. DeepSeek-MoE, Mixtral) for token prediction tasks [4, 5, 6, 7, 8]. More recently, pertinent to applications on AI alignment, the online version of MoE has garnered significant interest. This is particularly useful when agents interact with an environment in a repeated manner, receiving feedback in an online fashion, and needing to optimize in real-time.

**Online Mixture of Experts:** Historically, online MoE was framed as a *prediction with expert advice* problem, which has been extensively studied in the learning literature [9, 10, 11, 12]. The core principle involves aggregating predictions from multiple experts, where each expert is assigned a weight proportional to its historical predictive performance. Previous online MoE learning algorithms, such as EXP4, are theoretically guaranteed to converge to the best single expert in hindsight [13, 14, 9]. While this property ensures asymptotic optimality relative to the best individual expert, it introduces a concerning limitation. Specifically, such algorithms fail to account for scenarios

---

*Corresponding author.

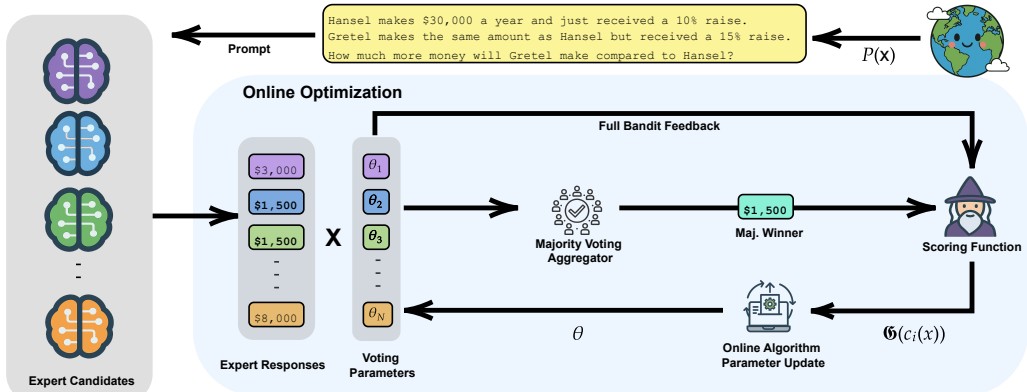

Figure 1: **Online Mixture of Experts via Majority Voting.** At each timestep, a context x is sampled from a distribution $P(\mathbf{x})$. A prompt derived from x is used to query a set of expert candidates $\mathcal{E}_{\texttt{cand}}$. The responses are aggregated via a majority voting mechanism parameterized by $\theta$, yielding a single output. All expert responses and the aggregated output are evaluated by a scoring function (e.g., an oracle or human). The voting parameters $\theta$ are updated online to maximize expected score, using full-bandit feedback over a finite time horizon.

where an aggregate outcome of the experts' predictions—such as a majority voting, plurality consensus, or any form of preference aggregation—could yield superior performance compared to any single expert. In social choice, this phenomenon is often referred to as the *wisdom-of-the-crowd*. If there exists potential improvement from collective decision-making, traditional online MoE methods, such as EXP4, would not capture this. This gap highlights a key question, *could we design an algorithm that provides convergence to the best collective decision-making committee rather than to the best single expert?* In our setting, we aim to learn both the competencies of the experts, and a sufficiently optimal strategy to combine the experts aggregate outcome in a real-time online environment. Evidently this also has important ramifications to generative LLMs, allowing for adaptively improved responses to tasks across various domains, when multiple LLMs are available.

**Key Contributions:** We make the following contributions in our work,

> **Online Expert Aggregation:** We introduce a family of methods that dynamically aggregate expert outputs in real-time through a combination of majority voting and online learning. This provably optimizes the collective aggregate competency of the expert set.

> **LLM Application:** We empirically validate the effectiveness of our method across realistic problem domains via online majority voting mechanisms to aggregate responses from multiple LLMs serving as experts, yielding improved answer quality.

> **No-Regret Guarantees:** We provide theoretical guarantees with respect to the sublinear finite-time regret of the learning algorithm relative to the optimal solution under perfect information, bridging online learning methods to social choice theory.

## 2 Problem Setting

To formally describe the problem setting, consider a learning model where the output is generated by combining the results of multiple weak learners through an aggregation function. These weak learners, referred to as *experts*, produce individual outputs, and the aggregation function determines how these outputs are combined. Let $\mathcal{E}$ denote the set of expert with the size $N$. Provided a context $\mathbf{x} \in \mathcal{X}$, expert $i \in \mathcal{E}$ makes a prediction $\boldsymbol{c}_i(\mathbf{x}) \in \mathcal{C}_i$ which represents an abstract categorical ordering or a singleton. Since $\{\mathcal{C}_i\}_{i \in \mathcal{E}}$ might be different sets (e.g., $\mathcal{C}_i = \{\texttt{cat}, \texttt{dog}, \texttt{mouse}\}$ and $\mathcal{C}_j = \{\texttt{mouse}, \texttt{eagle}, \texttt{fly}\}$), we introduce a *standardizer* $\mathfrak{S}(\cdot)$ that maps an arbitrary collection of prediction spaces $\{\mathcal{C}_i\}_{i \in \mathcal{E}}$ to a unified output space $\mathcal{C}$. Formally, the standardizer satisfies,

$$\bigcap_{i \in \mathcal{E}} \mathcal{C}_i \subseteq \mathcal{C} := \mathfrak{S}(\{\mathcal{C}_i\}_{i \in \mathcal{E}}) \subseteq \bigcup_{i \in \mathcal{E}} \mathcal{C}_i,$$

We refer to the size of the standardized output as $\mathsf{D}_{\mathcal{C}} := |\mathfrak{S}(\{\mathcal{C}_i\}_{i \in \mathcal{E}})|$.

**Aggregation function:** It is $\mathfrak{A}_\theta : \bigotimes_{i \in \mathcal{E}} \mathcal{C}_i \mapsto \mathcal{C}$ and characterized by a set of parameters $\theta = \{\theta_i\}_{i \in \mathcal{E}}$. It takes as input the experts' predictions $\{c_i(\mathbf{x})\}_{i \in \mathcal{E}}$ and produces a new prediction $\tilde{c} \in \mathcal{C}$.

**Scoring function:** The *goodness* of the aggregation function's output is valued by the scoring function, $\mathfrak{S} : \mathcal{C} \mapsto \mathbb{R}$ which assigns a real-value score in $\mathbb{R}$ to it's prediction. Suppose a series of contexts, $\mathbf{x}$, are randomly drawn from a distribution $P$. This contextual information is passed into each expert, and the predictions are subsequently aggregated parametrically. As indicated in Eq. (2.1), the prediction itself may be context-dependent. The scoring function evaluates the accuracy or quality of the prediction. The performance of a given aggregation function $\mathfrak{A}_\theta$ is measured by the average scoring function,

$$\mathbb{E}_{\mathbf{x} \sim P}[\mathfrak{S} \circ \mathfrak{A}_\theta] = \int_{\mathbf{x}} \mathfrak{S} \circ \mathfrak{A}_\theta \left(\{c_i(\mathbf{x})\}_{i \in \mathcal{E}}\right) \, dP(\mathbf{x}), \tag{2.1}$$

where $\mathfrak{S} \circ \mathfrak{A}_\theta$ denotes the composition of the scoring and aggregation functions. We aim is to maximize the average scoring function, formulated as, $\theta^* \in \arg\max_{\theta \in \boldsymbol{\theta}} \mathbb{E}_{\mathbf{x} \sim P}[\mathfrak{S} \circ \mathfrak{A}_\theta]$. The properties of the utility function, such as commutativity or monotonicity, can depend on the structure of the aggregation outcomes. Our goal is to design an algorithm that learns the optimal parameters for the aggregation algorithm in an online setting.

**Remark 1. Catalogue Standardizer:** *The standard dimension represents unified responses across experts (e.g., a fixed set of answers for LLMs). We distinguish two approaches: (1) **catalogue-based**, the model produces a catalogue of items where each expert selects from, and (2) **model-based**, where experts (e.g., transformers, MLPs) project arbitrary outputs to a standardized logit or token space $\mathbb{R}^D$. We applied the **catalogue** based standardizer for our experiments.*

The score of individual expert, expert's *competency*, can be obtained from the scoring function by computing the expectation over all possible contexts distributed according to $P(\mathbf{x})$ as follows,

$$p_i := \int_{\mathbf{x}} \mathfrak{S}\left(c_i(\mathbf{x})\right) \, dP(\mathbf{x}). \tag{2.2}$$

In more straightforward terms, the competency of an expert refers to its individual ability to maximize the scoring-aggregation function over a singleton committee consisting of itself - i.e. the ability of the expert to individually predict the correct answer among candidate solutions. In voting theory, this is an important piece of information measuring potential effectiveness of aggregating experts. The average scoring function is the expectation over the composition of the aggregation first $\mathfrak{A}(\cdot)$, followed by the scoring function $\mathfrak{S}(\cdot)$, according to distribution $P(\mathbf{x})$ over all contexts, Eq. (2.1). Which we denote as as the scoring-aggregation function, or aggregation function for brevity.

**Assumption 2.1. $\tilde{\epsilon}$-Incrementality (Heterogeneous Expert Competencies):** *Every expert has a different $p_i$, offset by at least $\tilde{\epsilon}$ increment, that is $\inf_{i \neq j} (p_i - p_j) \geq \tilde{\epsilon}, \ \forall i, j \in 1 \ldots N$.*

**Extensions of Condorcet's Jury Theorem:** The celebrated Condorcet's Jury Theorem [15] states that with competent experts ($p_i > 0.5$), majority voting accuracy approaches 1 as $N \to \infty$. List and Goodin [16] extends this to plurality voting, showing asymptotic accuracy can hold even with $p_i < 0.5$ under strict reliability assumptions, i.e. that the probability of any expert selecting the correct class is greater than selecting any incorrect class. However, when these assumptions fail, incompetent experts can degrade aggregate accuracy, highlighting the need for robust expert aggregation in our setting. Our work applies epistemological social choice theory [17, 16] to MoE models, focusing on maximizing predictive accuracy with a modest, finite set of experts ($\sim$20), where aggregate accuracy guarantees may or may not hold [16, 18, 19]. Unlike traditional social choice settings, we prioritize maximizing output accuracy (and accordingly minimizing regret) under imperfect information. Key challenges include dynamically inferring heterogeneous expert competencies and aggregating unreliable multi-class predictions under sample efficiency constraints.

## 2.1 Egalitarian Majority Voting

Consider a special setting in which experts' scored prediction are binary $\mathtt{Im}(\mathfrak{S}) = \{0, 1\}$, where 1 indicates that the expert is deemed correct and 0 if incorrect. Suppose that the score-aggregation function is the majority voting with a random tie-breaking. In this setting, the composition of the

scoring and aggregation function is expressed in Eq. (2.3). In words, it is 1 whenever majority of the experts give one and zero otherwise.

$$\mathfrak{G} \circ \mathfrak{A}(\{\boldsymbol{c}_i(\mathbf{x})\}_{i \in \mathcal{E}}) = \mathbb{I}\left(\sum_{i \in \mathcal{E}} \mathfrak{G}(\boldsymbol{c}_i(\mathbf{x})) > \frac{N}{2}\right) + \mathbb{I}\left(\sum_{i \in \mathcal{E}} \mathfrak{G}(\boldsymbol{c}_i(\mathbf{x})) = \frac{N}{2}\right) \cdot Y, \qquad (2.3)$$

where $\mathbb{I}(\cdot)$ is the indicator function and $Y \sim Be(1/2)$ is a Bernoulli random variable with parameter $1/2$. Let $\mathcal{X}_i := \{\boldsymbol{x} = 1 | \boldsymbol{c}_i(\mathbf{x})\}$ to be the set of contexts for which expert $i$ gives a correct prediction and $\mathcal{X}_i^c$ its complement. Note that the measure of $\mathcal{X}_i$ is $p_i$. Using this definition, the average scoring function can be rewritten as follows,

$$\mathbb{E}[\mathfrak{G} \circ \mathfrak{A}] = \mathbb{P}\left(\bigcup_{\substack{S \subseteq \mathcal{E} \\ |S| > N/2}} \bigcap_{i \in S} \mathcal{X}_i \bigcap_{j \in S^c} \mathcal{X}_j^c\right) + \frac{1}{2}\mathbb{P}\left(\bigcup_{\substack{S \subseteq \mathcal{E} \\ |S| = N/2}} \bigcap_{i \in S} \mathcal{X}_i \bigcap_{j \in S^c} \mathcal{X}_j^c\right).$$

**Remark 2.** *This setting can be extended to non-binary prediction set, i.e.,* $Im(\mathfrak{G}) = \{1, 2, ..., K\}$ *for some $K$ and the output of the aggregation function is $i \in Im(\mathfrak{G})$ whenever a majority of experts vote for $i$. In the event of a tie among the experts, the aggregation function outputs one of the tied items uniformly at random. That is the Bernoulli variable in* (2.3) *will be replaced with a uniform random variable over the set of items that received the highest (tied) number of votes.*

**Independent Experts:** Suppose that the experts are independent, i.e., $P(\bigcap_{i \in S} \mathcal{X}_i) = \prod_{i \in S} P(\mathcal{X}_i)$ for all $S \subseteq \mathcal{E}$. In this case, the average scoring function will be,

$$\mathbb{E}[\mathfrak{G} \circ \mathfrak{A}] = \sum_{\substack{S \subseteq \mathcal{E} \\ |S| > N/2}} \prod_{i \in S} p_i \prod_{j \in S^c} (1 - p_j) + \frac{1}{2} \sum_{\substack{S \subseteq \mathcal{E} \\ |S| = N/2}} \prod_{i \in S} p_i \prod_{j \in S^c} (1 - p_j). \qquad (2.4)$$

Evaluating the above averaging function is combinatorially complex as its complexity grows exponentially with the number of experts $N$. Combinatorial multi-armed bandit (CMAB) have been applied for this purpose in [20, 21, 22] . Such methods assume the learning algorithm has access to an offline $(\alpha\beta)$-approximation oracle oracle where, with probability $\beta$, it outputs a solution whose value is at least $\alpha$ proportional of the maximum reward. Specifically, when the reward function is monotone and submodular, such an oracle can be given where $\alpha = 1 - 1/e$. In our setting, the learner does not have access to such an $(\alpha\beta)$-oracle. For instance, adding experts an existing committee does not result in any generalizable behaviour in our majority voting setting. Throughout this paper, we assume that the experts are independent. However, as it is discussed in Section 3, our proposed algorithms assemble their committees solely using the marginal experts' competencies and subsequently, the theoretical guarantees remain valid even if the experts are correlated.

**Optimal Egalitarian Committee (OEC):** The optimal egalitarian committee (EC), also known as the binary weighted participation problem, refers to the formation of a committee in which each expert receives exactly one vote. Examples of such committees include corporate boards, academic panels, and juries. In machine learning, similar structures appear in ensemble methods such as Random Forests, boosting algorithms, deep learning ensembles, and Mixture-of-Experts (MoE) models.

While online learning algorithms like EXP4 typically converge to the best expert in hindsight, prior work has shown that it is often possible to achieve better performance by identifying an optimal egalitarian committee—that is, a subset of experts whose aggregated votes outperform the best expert. In this setting, the final prediction is determined by majority vote, and the resulting combined estimate can exceed the accuracy of any individual expert. We denote this committee as $\mathcal{E}^* \subseteq \mathcal{E}$.

When considering the multi-class classification problem, for a majority vote to pass, more than some minimum amount, referred to as a *quota*, of the experts should have voted for the correct answer, each occurring with probability $p_i$. For majority voting, we set the quota to be half of the number of experts (Optionally, the quota could also be reduced to allow for the experts to arrive at a clear option as a winner from a choice of options [17] ). We denote the average scoring function of a candidate committee set $\mathcal{E}_{\text{cand}} \subseteq \mathcal{E}$ by $\mathbb{P}_{\text{maj}}(\mathcal{E}_{\text{cand}})$ and is given by,

$$\mathbb{P}_{\text{maj}}(\mathcal{E}_{\text{cand}}) := \sum_{\substack{S \subseteq \mathcal{E}_{\text{cand}} \\ |S| \geq |\mathcal{E}_{\text{cand}}|/2}} \prod_{i \in S} p_i \prod_{j \in S^c} (1 - p_j) + \frac{1}{2} \sum_{\substack{S \subseteq \mathcal{E}_{\text{cand}} \\ |S| = |\mathcal{E}_{\text{cand}}|/2}} \prod_{i \in S} p_i \prod_{j \in S^c} (1 - p_j). \qquad (2.5)$$

Accordingly, the optimal committee is $\mathcal{E}^* \in \arg\max_{\mathcal{E}_{\text{cand}} \subseteq \mathcal{E}} \mathbb{P}_{\text{maj}}(\mathcal{E}_{\text{cand}})$.

**Successive Expert Elimination Function:** To form th OEC, we can intuitively successively include each expert sorted by their respective $p_i$, to a candidate expert set $\mathcal{E}_{\text{cand}}$, given that such an inclusion would improve the overall accuracy of the committee. For any subset $\mathcal{E}' \subseteq \mathcal{E}_{\text{cand}}^c$, we define the advantage function as follows,

$$\mathfrak{F}(\mathcal{E}_{\text{cand}}, \mathcal{E}') := \mathbb{P}_{\text{maj}}(\mathcal{E}_{\text{cand}} \cup \mathcal{E}') - \mathbb{P}_{\text{maj}}(\mathcal{E}_{\text{cand}}), \quad \text{and} \quad \mathbb{P}_{\text{maj}}(\emptyset) := 0. \tag{2.6}$$

Clearly, when the above function is negative, the addition of $\mathcal{E}'$ to $\mathcal{E}_{\text{cand}}$ will not produce $\mathcal{E}^*$, i.e.,

$$\mathfrak{F}(\mathcal{E}_{\text{cand}}, \mathcal{E}') \leq 0 \implies \mathcal{E}^* \neq \mathcal{E}_{\text{cand}} \cup \mathcal{E}'.$$

**Lemma 2.1.** ***Top-K Ordinality of Experts:*** *The optimal expert committee $\mathcal{E}^* \subseteq \mathcal{E}$ is a subset of Top-K experts based on their competencies, for $0 < K \leq N$. (Proof can be found in Appendix A.1.)*

**Elimination of Experts:** To build the OEC under the perfect information, i.e., experts' competencies are known, we propose a constructive algorithm, where we start with $\mathcal{E}_{\text{cand}} = \mathcal{E}$, and successively eliminate admissible subsets $\mathcal{E}'$, until there is no further possible improvement via the $\mathfrak{F}(\cdot)$. Given the result of Lemma 2.1, starting with all experts, $\mathcal{E}$, the set of admissible experts must fall within the set of Top-$K$ experts for some unknown $0 < K \leq N$, i.e. $\mathcal{E}' \subseteq \text{TopK}(\mathcal{E})$. The benefit of Lemma 2.1 allows us to efficiently compute the OEC, avoiding combinatorial search over all possible subsets. Under perfect information, this allows us to construct the OEC via a greedy algorithm. (See Algorithm 3 in Appendix.)

## 2.2 Weighted Majority Voting

We extend from the egalitarian committee in Section 2.1 to a scenario where voters in a committee have unequal voting power. Intuitively, more accurate experts should have stronger voting power to sway the accuracy of the majority voting system. In a *weighted majority voting* (WMV) system, the voting mechanism is a decision-making system where each expert has a specific level of influence, determined by an assigned weight. Previously [13] investigated the problem providing only an optimal error guarantee limited to binary predictions, in the offline setting where voters can only produce one of two outputs $\{0, 1\}$. We aim to extend the setting to multiple class predictions with no-regret guarantees in the online setting.

We denote the weight of expert $i \in \mathcal{E}$ by $\theta_i \in \mathbb{R}$. These weights often reflect factors such as expertise, historical performance, or stake in the outcome. Each expert votes for a candidate class, and the total weight of votes for each candidate is calculated. The option with the weight that surpasses a quota $Q$ is chosen as the final decision. This approach extends simple majority voting by allowing experts to have unequal influence, making it useful in contexts like corporate governance, expert panels, and machine learning ensembles, where experts vary in reliability or expertise. Let us define $\mathbb{P}_{\text{maj}}(\mathcal{E}, \theta)$ as the aggregate scoring function of a committee of voters with unequal weights denoted by $\theta = (\theta_i)_{i \in \mathcal{E}}$.

$$\mathbb{P}_{\text{maj}}(\mathcal{E}, \theta) := \sum_{S \subseteq \mathcal{E}} \mathscr{F}_N(S, \theta) \left( \prod_{i \in S} p_i \prod_{j \in S^c} (1 - p_j) \right), \quad \text{where} \quad \mathscr{F}_N(S, \theta) := \mathbb{I}\left( \sum_{j \in S} \theta_j > Q \right),$$

$$(2.7) \qquad\qquad\qquad\qquad\qquad\qquad\qquad\qquad\qquad\qquad (2.8)$$

where $\sum_{j \in \emptyset} \theta_j := 0$. In words the above selection function decides whether the configuration $S$ of the committee $\mathcal{E}$ meets the feasibility requirements for the vote to pass. In the special case where all weights are one, then (2.7) reduces to (2.4) with an appropriate tie-breaking. The weighted majority voting problem with quota $Q$ can be stated as,

$$\max_{\theta \in \mathbb{R}_+^N, \|\theta\|_1 \leq 2Q} \mathbb{P}_{\text{maj}}(\mathcal{E}, \theta). \tag{2.9}$$

In our experiments, for simplicity, we used $Q = N/2$, ensuring majority consensus. This guarantees a clear winner when the majority of experts predict optimally. Should a majority decision be unreachable, a random class is sampled uniformly as the winner. To help further simplify the solution to Eq. (2.9), via Lemma 2.2, we can solve for $\theta$ at equality as opposed to constrained cap on $Q$, constituting a more convenient expression encompassing the latter.

**Lemma 2.2.** *Weights Satisfied at Equality: When solving for the optimal weights for $\theta$-weighted majority voting, the constraint $Q \leq \|\theta\|_1 \leq 2Q$, from Eq. (2.9) can be replaced with $\|\theta\|_1 = 2Q$. (Proof is available in Appendix A.5.)*

### 2.2.1 Solving the Optimal Weighted Voting Committee

A key objective is optimizing the parameters, $\theta$, to maximize majority voting accuracy in Eq. (2.7). While convex combinations of expert outputs with non-zero weights may yield higher optima, the problem involves iterative combinatorial complexity. For binary outcomes, i.e. only two classes to vote for, optimal weights follow the log-odds ratio: $\theta_i^* = \log(p_i/1 - p_i)$, $\forall \theta_i \in \mathbb{R}$ [23, 24] (see Appendix A.13 for a discussion and proof). For multi-class settings, the specifications of the problem challenges the aforementioned existing theoretical guarantees. Everaere, Konieczny, and Marquis [18] and Abramowitz, Lev, and Mattei [25] provide further insight, yet optimality guarantees still require binary outcomes or large expert pools. Karge and Rudolph [19] provides non-asymptotic bounds via a competency gap, but remains limited for small-scale online settings. Therefore, our problem requires novel approaches balancing accuracy maximization with sample efficiency and expert heterogeneity. First, we leverage potential simplifications in optimal weighted majority voting problem structure.

**Proposition 2.1.** *Discrete Image: Many configurations of $\theta$ can lead to the same result for $\mathbb{P}_{maj}(\mathcal{E}, \theta)$. (Proof is provided in Appendix A.3.).*

The results of Prop. 2.1 are rather intuitive, stating that many different configurations of weights could lead to the same expected accuracy via majority voting. One can prove this constructively by considering a scenario where two or more experts do not contribute to the optimal solution. So long as their weights do not contribute to swinging a majority vote for any possible outcome (i.e. this could be an infinitesimally small weight), it could be a valid optimal solution. Next, we provide an algorithm which for any given ordering over the competencies, denoted as $\text{Ord}(p) := (p_1, p_2, \ldots p_N)$, such that $p_1 \geq p_2 \geq \ldots p_N$, computes an optimal set of weights.

**Mixed Integer Program Formulation:** The exact formulation of the optimal weights in the voting problem can be formulated as a mixed-integer program (MIP), governed by the properties outlined in this section. Let $\boldsymbol{\theta} = [\theta_1, \theta_2, \ldots, \theta_N]$ be a positive vector, where $\theta_i \geq 0$ for all $i$. Let $\mathbf{x} = [\mathbf{x}_1, \mathbf{x}_2, \ldots, \mathbf{x}_{2^N}]$ be a predefined vector where $\mathbf{x}_S \in \{0, 1\}^N$ and represents configuration $S \subseteq \mathcal{E}$, i.e., $[\mathbf{x}_S]_j = 1$ if and only if expert $j$ is voted correctly in configuration $S$. We also define $\mathsf{Z}_S$ as a binary variable corresponding to $S$, determined by the following if-else condition,

$$\mathsf{Z}_S := \begin{cases} 1 & \text{if } \mathbf{x}_S \cdot \boldsymbol{\theta} > Q, \\ 0 & \text{otherwise}, \end{cases}$$

where $Q \in \mathbb{R}^+$ is the given quota. The problem in (2.9) can be formulated as an MIP with the following objective function,

$$\max_{\theta \in \mathbb{R}_+^N} \sum_{S \subseteq \mathcal{E}} \mathsf{Z}_S \prod_{i \in S} p_i \prod_{j \in S^c} (1 - p_j). \tag{2.10}$$

This MIP computes the expected value over all voting outcomes using a binary selection variable and possibility variable $\mathsf{Z}_S$. When maximizing for the MIP objective, we must formulate a set of valid constraints. Leveraging Lemma 2.2, we tighten the quota condition, introducing constraint $\sum_j \theta_j = 2Q$. Assumption 2.1 enforces $\theta_i \neq \theta_j$ as a constraint. Importantly, we enforce logical consistency by ensuring that complementary voting outcomes cannot simultaneously satisfy both an outcome and its complement (e.g. if scenario $S = [0, 1, 1, 0]$ is possible and $\mathsf{Z}_S = 1$, then its complement $S^c = [1, 0, 0, 1]$ is not, implying $\mathsf{Z}_{S^c} = 0$.). These constraints collectively validate voting configurations, enforce expert competency distinctions, and eliminate contradictory outcomes, while maximizing for a global solution. We provide the details for this MIP formulation in Appendix A.8.

**Lemma 2.3.** *Majority Voting Dominance over Egalitarian Voting: For any set of experts, the optimal majority weighted voting method will always yield a stronger or equal predictive accuracy than the plurality vote of the committee. (Proof can be found in Appendix A.2.)*

Lemma 2.3 posits that given the same committee of experts, utilizing the $\theta$-WMV algorithm (Alg. 2), to aggregate experts' output will always yield a stronger result than using egalitarian voting.

Therefore, if the computational resources are available the $\theta$-WMV algorithm will produce a better accuracy. However, this depends on the systems ability to serve all experts and provide a solution to the MIP expressed in Eq. (2.10). If the goal is to learn an efficient representation of experts (i.e. a sparse set of experts) then SEE (in Alg. 1) can be a better choice to retain efficiency.

# 3 Online Learning

In previous work, the online MoE problem was typically framed as a no-regret problem relative to the best single expert in hindsight (Hannan consistency). Recent work on online MoE learning focuses on identifying top-$K$ experts based on their competencies. A prominent gap remains in efficiently learning optimal aggregation with heterogeneous expert competencies *without* combinatorial explosion. Prior work assumes either pairwise feedback (duelling) or submodularity, yet there remains a focal body of work addressing optimal MoE aggregation in an online learning setting. Key previous approaches include duelling bandits, where pairwise comparisons identify dominant experts [26, 27] , and submodular bandits, which treat committee formation as a matroid optimization problem with greedy guarantees [28, 29] . Alternatively, satisficing methods [30, 31, 32] trade optimality for efficiency, while Top-K bandits [33, 34, 35] leverage competency ordering for successive acceptance and elimination of arms. Despite the ample amount of research in the online Top-K arm-selection area, a prominent research gap yet remains which bridges voting theory to online no-regret guarantees.

**Notion of Regret:**   As our proposed algorithm is a no-regret learner, we formally introduce the notion of regret in Def. 3.1. Cumulative regret, $R_T$, is defined as the difference between the expected majority voting aggregate accuracy, $\mathbb{P}_{\texttt{maj}}(\mathcal{E}, \theta^*)$, under parameters, $\theta^*$, subtracted by the expected aggregate accuracy under the current parameters $\mathbb{P}_{\texttt{maj}}(\mathcal{E}, \theta^t)$.

**Definition 3.1.** *Regret: The cumulative regret of an online algorithm over a time horizon of length $T$ that selects a set of weights $\theta^t$ at round $t$ is given by $R_T := \sum_{t=1}^{T} \mathbb{P}_{maj}(\mathcal{E}, \theta^*) - \mathbb{P}_{maj}(\mathcal{E}, \theta^t)$.*

## 3.1 Successive Expert Elimination (SEE)

The idea of successive elimination in online learning was demonstrated in Even-Dar et al. [33] where the successive elimination bandit algorithm with a PAC bound on sample complexity was proposed, along with both a gap dependent and gap independent versions based on the UCB. Rejwan and Mansour [35] presents a successive elimination algorithm in the full-bandit-feedback setting (a minor detail is that only the sum of rewards is available, which is an additive result from the individual rewards). Yet, it does not consider the optimal Top-K expert selection (only that K is given).To distinguish, in our situation, the aggregate reward, $\mathbb{P}_{\texttt{maj}}$ function is not necessarily additive, and the Top-$K$ is not given beforehand and must be determined. We first propose an algorithm based on the UCB, where there is some confidence range between our estimate of $\hat{p}_i$ and the true value for each expert $p_i^*$. This confidence bound should shrink over time with increasing samples.

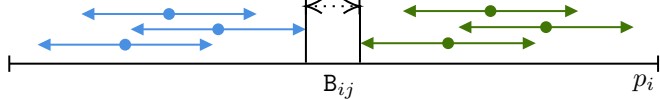

Figure 2: Illustration of a breakage event $\texttt{B}_{ij}$.

**Definition 3.2.** *Breakage Event: A breakage event at round $t$ denoted as $\texttt{B}_{ij}^t$ between experts $i$ and $j$ denotes an event where the upper confidence limit of the empirical estimation of $j$'s competency at round $t$, $\hat{p}_j^t$, is less than the lower confidence limit of $\hat{p}_i^t$, i.e.,*

$$\texttt{B}_{ij}^t : \hat{p}_j^t + \texttt{UCB}_j^t < \hat{p}_i^t - \texttt{UCB}_i^t.$$

**Intuition of Successive Expert Elimination (SEE):**   The key idea of SEE is that we can use the confidence bounds to successively eliminate experts from a candidate committee until we arrive at a final set of optimal candidates, with high confidence. The algorithm begins with the complete set of candidate committee equal to the set of all experts $\mathcal{E}_{\texttt{cand}} \leftarrow \mathcal{E}$ and all experts are played at each round until a breakage event occurs $\texttt{B}_{ij}^t$. At the event of a breakage, we first order the experts whose estimated competencies are lower than expert $j$, say $j_1, j_2, ..., j_k := j$, such that $\hat{p}_{j_1}^t < \hat{p}_{j_2}^t \leq ... \leq \hat{p}_j^t$ (pictured as blue points in Diagram 2). Then, we define $\texttt{TopK}(\mathcal{E}_{\texttt{cand}}^{\leq j})$ as the set $\left\{ \{j_k\}, \{j_k, j_{k-1}\}, ..., \{j_1, ..., j_k\} \right\}$. To perform a removal test, for each element $\mathcal{E}'$ in $\texttt{TopK}(\mathcal{E}_{\texttt{cand}}^{\leq j})$,

we consult the advantage function to see if $\mathfrak{F}(\mathcal{E}_{\text{cand}}, \mathcal{E}') \leq 0$; that is no advantage can be obtained by including $\mathcal{E}'$ with high confidence. If so, all experts in $\mathcal{E}_{\text{cand}}^{\leq j} := \{j_1, ..., j_k\}$ could be subsequently eliminated from $\mathcal{E}_{\text{cand}}$ (i.e. removal of blue points from the previous union of blue and green w.r.t Diagram 2).

---

**Algorithm 1** Successive Expert Elimination (SEE)

---

1: $\mathcal{E}_{\text{cand}} \leftarrow \mathcal{E}$              ▷ Set the candidate expert set to the full set of experts.
2: **for** $t = 1$ to $T$ **do**
3:      Estimate $\hat{p}_i^t$ and upper confidence bound, $\text{UCB}_i^t$, for each expert in $\mathcal{E}_{\text{cand}}$.
4:      **if** any $\text{B}_{ij}^t$ event is observed **then**            ▷ Breakage event occurs.
5:          **for** every $\text{B}_{ij}^t$, in descending order **do**
6:              $\mathcal{E}_{\text{cand}}^+ \leftarrow \mathcal{E}_{\text{cand}}^{>j}$
7:              **if** $\mathfrak{F}(\mathcal{E}_{\text{cand}}^+, \mathcal{E}') \leq 0, \quad \forall \mathcal{E}' \in \text{TopK}(\mathcal{E}_{\text{cand}}^{\leq j})$ **then**      ▷ Perform removal test.
8:                  $\mathcal{E}_{\text{cand}} \leftarrow \mathcal{E}_{\text{cand}} \setminus \mathcal{E}_{\text{cand}}^{\leq j}$.           ▷ Expert set truncation.
9:              **end if**
10:          **end for**
11:      **end if**
12: **end for**
13: **return** $\mathcal{E}_{\text{cand}}$

---

**Lemma 3.1.** *PAC Sample Complexity: Suppose that the competency of each expert $i$ is modeled as a sub-Gaussian random variable with mean $p_i$ and variance $\sigma_i^2 \leq \sigma^2$. Under Assumption 2.1, for any two experts $i$ and $j$, if the number of samples $t$ and shrinkage term $\text{UCB}_i^t$ satisfy*

$$\text{UCB}_i^t := \sqrt{\frac{2\sigma_i^2 \log(4/\delta)}{t}}, \quad t \geq t_0 := \frac{32\sigma^2 \log(4/\delta)}{\tilde{\epsilon}^2},$$

*then, with probability at least $1 - \delta$, $\delta \in (0, 1)$, the breakage condition, $\text{B}_{ij}^t$, will be met. (Proof provided in Appendix A.6)*

**Theorem 1.** *Regret Bound for* SEE*: The total regret of* SEE *Algorithm in 1 is bounded by:*

$$R_T \in \mathcal{O}\left(N \log(T)/\tilde{\epsilon}^2\right). \tag{3.1}$$

*Sketch of Proof:* The bound follows a PAC elimination sample complexity (Lemma 3.1), followed by a union bound over $\mathcal{O}(N^2)$ pairwise comparisons. Next, we set the failure probability $\delta = 1/T$ to control catastrophic regret. The dominant term arises from aggregating $N-1$ elimination rounds, each contributing $\mathcal{O}\left(\log(T)/\tilde{\epsilon}^2\right)$ regret. (For complete proof see Appendix A.7.)

### 3.2 Online Weighted Majority Voting Solution (Full Bandit Feedback)

**Full Bandit Feedback:** As discussed in previous section, the optimal weighed majority voting problem ($\theta$-WMV) can be cast as an MIP under perfect information. Consequently, this can be integrated into UCB style bandit algorithms. To implement a no-regret learning algorithm, we solve for the best- and worst-case outcomes under uncertainty in the experts' competencies via the MIP, and quantify the corresponding shrinkage in the bounded simple regret.

---

**Algorithm 2** $\theta$-Weighted Majority Voting with Full Bandit Feedback ($\theta$-WMV)

---

**Require:** Set of experts $\mathcal{E}$ with $n \in \{1, \ldots, N\}$.
1: $\mathcal{E}_{\text{cand}} \leftarrow \mathcal{E}$              ▷ Set the candidate expert set to the full set of experts.
2: **for** $t = 1$ to $T$ **do**
3:      Estimate $\hat{p}_i^t$ and upper confidence bound, $\text{UCB}_i^t$ using the history for all experts in $\mathcal{E}_{\text{cand}}$.
4:      Solve for $\bar{\theta}^*$ with optimistic estimates, $\hat{p}^t + \text{UCB}_i^t$, via MIP.
5:      Elicit all responses from $\mathcal{E}$, and combine responses using weighted majority voting with $\bar{\theta}^*$.
6:      Receive feedback from scoring function, and add it to the history.
7: **end for**
8: **return** $\bar{\theta}^*$

---

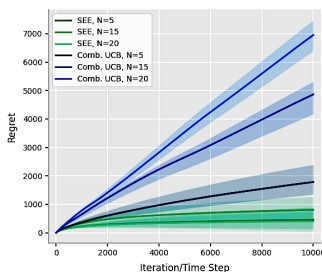
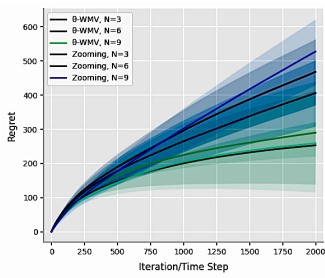
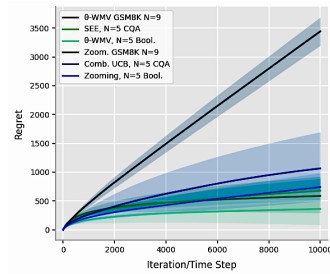

| (a) Optimal Egalitarian Voting. | (b) Optimal Weighted Voting. | (c) LLM Experts. |

Figure 3: **Comparison of Cumulative Regret for MoE Voting Environments.** Fig. 3a: SEE on simulated Bernoulli experts; Fig. 3b: $\theta$-WMV on simulated Bernoulli experts; Fig. 3c: LLM experts across domains (e.g., GSM8k, BoolQ etc.), with tasks sampled from each domain. Shaded regions show $\pm$UCB uncertainty. Evidently, SEE and $\theta$-WMV consistently outperform baselines in regret minimization.

| Algorithm | $N$ | $\lvert\mathcal{E}^*\rvert$ | $\mathbb{P}^*_{\text{maj}}$ | $\tilde{\epsilon}$ | $R_T$ | $\%R\downarrow$ | $\hat{P}_{\text{maj}}$ | Domain |
|---|---|---|---|---|---|---|---|---|
| SEE | 20 | 9 | 0.992 | 0.010 | **546.2** | 0.921 | **0.980** | Bernoulli |
| Comb. UCB | 20 | 9 | 0.992 | 0.010 | 6932 | - | 0.370 | Bernoulli |
| $\theta$-WMV | 9 | 9 | 0.927 | 0.027 | **587.0** | 0.829 | **0.927** | GSM8K |
| Zooming | 9 | 9 | 0.927 | 0.027 | 3443 | - | 0.604 | GSM8K |
| SEE | 5 | 3 | 0.810 | 0.031 | **676.8** | 0.364 | **0.810** | CommonsenseQA |
| Comb. UCB | 5 | 3 | 0.810 | 0.031 | 1065 | - | 0.664 | CommonsenseQA |
| $\theta$-WMV | 5 | 5 | 0.735 | 0.013 | **360.9** | 0.512 | **0.735** | BoolQ |
| Zooming | 5 | 5 | 0.735 | 0.013 | 740.3 | - | 0.694 | BoolQ |

Table 1: Multi-armed bandit experiment results showing consistent performance across environments, with time horizon $T \leq 10^4$. Performance metrics show reduced cumulative regret ($R_T$) and increased empirical accuracy ($\hat{P}_{\text{maj}}$) across all baselines. Extended empirical results can be found in Appendix C.5.

**Theorem 2.** *Regret Bound for $\theta$-WMV: Under the Assumption A.1, Algorithm 2 achieves a regret bound of $R_T \in \mathcal{O}\left(\sqrt{NT\log(T)}\right)$. (Proof in Appendix A.11.)*

*Sketch of Proof:* We show that the MIP solution to Eq. (2.10) under $\bar{\theta}^*$ misspecification is bounded by the difference in outcome scenario probabilities, which is also bounded by the uncertainty estimates of the expert's competencies, $p_i \pm$ UCB (see Lemma A.1 in Appendix). Subsequently, using standard PAC concentration bound arguments (i.e. Lemma 3.1), we derive the expected cumulative regret. A looser alternative bound under relaxed assumptions is also provided in Appendix A.16.

**Targeted-$m$ Online Majority Expert Voting:** In MoE applications, computational constraints often limit the number of experts that can be queried at each timestep to at most $m$. We address this extension to the problem by partitioning the $N$ experts into $N/m$ groups and running a windowed phase of $t_0$ rounds per group. This ensures high-probability breakage detection while preserving the original regret bounds, scaled by a multiplicative $N/m$ factor (Lemma 3.1). Specifically, the targeted-$m$ variant of SEE maintains $\mathcal{O}\left(N^2\log T/m\tilde{\epsilon}^2\right)$ regret, matching the original setting's guarantees with respect to $T$ despite the query limitation.

**Corollary 3.1.** *Targeted-$m$ WMV: An algorithm exists that allows for the regret bounds of Theorems 1 and 2 to hold with an multiplicative factor of $N/m$. (Proof can be found in Appendix A.12.)*

## 4 Empirical Study

We evaluate online learning in both model-based and data-derived settings. In the model-based setting, $N$ Bernoulli experts with fixed success probabilities test each algorithm's ability to distinguish and optimize for varying competencies under noise. In the data-derived setting, we sample tasks from diverse domains: GSM8K [36] (mathematical reasoning), CommonsenseQA [37] (implicit knowledge), and BoolQ [38] (evidence-based binary inference), each presenting unique challenges. Experts are instantiated as LLMs with varying reasoning capabilities (see Appendix C.1).

**Propose-then-Vote:** Datasets used in our experiments, such as CommonsenseQA [37], BoolQ [38] etc., constitute structured labelled data, that is the catalogue of answers is provided to the LLM to choose from (note Remark 1). But as we tackle unstructured open-ended tasks for experts, we need to standardize the vote aggregation functions with proper structure in order to elicit the appropriate expert feedback. One solution is to use the MoE to generate each individually their top responses to a prompt. By default, this is added to the admissible catalogue and the expert votes for their own response among the candidate responses. This, *propose-then-vote*, mechanism essentially uses the committee to first determine the answer catalogue, and then vote on it. (For additional details and a complete set of comprehensive experiments please refer to Appendix C.5.)

**Results Summary:** We analyze SEE (Algorithm 1) and $\theta$-WMV (Algorithm 2) in terms of expected aggregate accuracy $\mathbb{P}_{\text{maj}}(\mathcal{E}, \theta^T)$ (Eq. (2.7)) and cumulative regret $R_T$ (Def. 3.1). As baselines, we compare against combinatorial UCB bandits and the zooming algorithm [39], which serve as no-regret benchmarks. Further details – including prompting strategies (chain-of-thought), expert LLM specifications, baseline implementations, and ablation studies – are provided in Appendix C.

## 5 Conclusion

We introduce a new no-regret learning framework for online decision-making with voting-based expert ensembles, unifying online learning and social choice theory as a key contribution. Our algorithms—SEE for egalitarian voting and $\theta$-WMV for weighted majority voting—provide provable cumulative regret guarantees with UCB-style analysis, accompanied by definitive empirical performance. There are of course, also some limitations. First, although we consider up to a modest number of experts, the $\theta$-WMV algorithm depends on the solution to a mixed-integer program which could become overly complex when a large number of experts is introduced. Second, we did not consider the additional use of contextual and/or side information for the MoE framework to enhance regret minimization performance. Nevertheless, both learning algorithms offer practical advantages for ensemble MoE systems, particularly in LLM applications as demonstrated. This work establishes both theoretical foundations and practical tools for voting-based ensemble learning, opening new directions at the intersection of online learning and collective decision-making.

## Acknowledgements

We would like to thank Nicholas Mattei, Ben Abramowitz and Ariel D. Procaccia, as well as the reviewers for their insightful feedback and suggestions for our work.

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

# A  Proofs for Core Theory

## A.1  Proof for Lemma 2.1

**Top-K Ordinality of Experts:** The optimal expert committee $\mathcal{E}^* \subseteq \mathcal{E}$ is a subset of Top-K experts based on their competencies, for $0 < K \le N$.

*Proof.* Suppose there is an expert $i$ in the optimal committee $\mathcal{E}^*$ and another expert $j$ with a higher competency, i.e., $p_i < p_j$ who is not included in the optimal expert committee. In this case, replacing experts $i$ and $j$ results in a new committee with higher average score, leading to a contradiction. We prove it for when the size of the optimal committee is odd. Similar proof can be written for when it is even. Let $|\mathcal{E}^*| = 2s + 1$ for some $s$, $i \in \mathcal{E}^*$ but $j \notin \mathcal{E}^*$ and $p_i < p_j$. We form a new committee $\mathcal{E}_0 = \mathcal{E}^* \cup \{j\} \setminus \{i\}$. Let $S_i(k)$ be the event that precisely $k$ experts in $\mathcal{E}^*$ are correct including expert $i$. We also define $S_i^c(k)$ to be the event that $i$ is incorrect but $k - 1$ other experts voted correctly in $\mathcal{E}^*$. Similarly, we define $S_j^c(k)$ and $S_j^c(k)$ in committee $\mathcal{E}_0$. It is straightforward to see that the probability of either $S_i^c(k)$ or $S_i(k)$ occurs in committee $\mathcal{E}^*$ is equal to the probability of either $S_j^c(k)$ or $S_j(k)$ occurs in $\mathcal{E}_0$ for all $k \in \{s + 2, ..., 2s + 1\}$. This implies the sum of these events are the same for both committees in (2.5). Note that when $k = s + 1$, both events $S_i^c(k)$ and $S_j^c(k)$ will not appear in the average scoring in (2.5) as in both of these events $s$ experts are correct and $s + 1$ are incorrect. Moreover, the probability that event $S_j(s + 1)$ occurs is higher than the probability of the event $S_i(s + 1)$ since in the former expert $j$ is correct and in the latter expert $i$ and $p_j > p_i$. □

## A.2  Proof for Lemma 2.3

**Majority Voting Dominance over Egalitarian Voting:** For any set of experts, the optimal majority weighted voting method will always yield a stronger or equal predictive accuracy then the plurality vote of the committee.

*Proof.* Suppose $\mathcal{E}^*$ denotes the OEC where $|\mathcal{E}^*| = M$. In this case, we define $\theta$ such that $\theta_i = 1$ if $i \in \mathcal{E}^*$ and zero, otherwise. Next, we show that $\mathbb{P}_{\text{maj}}(\mathcal{E}, \theta) \ge \mathbb{P}_{\text{maj}}(\mathcal{E}^*)$, where the quota is $Q$ rather than $|\mathcal{E}^*|/2$ as in (2.5). Let $S_1 = S \cap \mathcal{E}^*$, $S_2 = S \cap (\mathcal{E}^*)^c$, then,

$$\mathbb{P}_{\text{maj}}(\mathcal{E}, \theta) = \sum_{S \subseteq \mathcal{E}} \mathscr{F}_N(S, \theta) \left( \prod_{i \in S} p_i \prod_{j \in S^c} (1 - p_j) \right)$$

$$= \sum_{S_1 \subseteq \mathcal{E}^*} \sum_{S_2 \subseteq (\mathcal{E}^*)^c} \mathscr{F}_N(S_1 \cup S_2, \theta) \left( \prod_{i \in S_1} p_i \prod_{j \in \mathcal{E}^* \setminus S_1} (1 - p_j) \right) \left( \prod_{i \in S_2} p_i \prod_{j \in (\mathcal{E}^*)^c \setminus S_2} (1 - p_j) \right)$$

Note that,

$$\mathscr{F}_N(S, \theta) = \mathbb{I} \left( \sum_{j \in S} \theta_j > Q \right) = 1 \Leftrightarrow |S \cap \mathcal{E}^*| = |S_1| > Q,$$

This implies that the nonzero terms in the above summation are those with $|S_1| > Q$. Therefore, we can simplify the summation as

$$\mathbb{P}_{\text{maj}}(\mathcal{E}, \theta) = \sum_{\substack{S_1 \subseteq \mathcal{E}^* \\ |S_1| > Q}} \sum_{S_2 \subseteq (\mathcal{E}^*)^c} \left( \prod_{i \in S_1} p_i \prod_{j \in \mathcal{E}^* \setminus S_1} (1 - p_j) \right) \left( \prod_{i \in S_2} p_i \prod_{j \in (\mathcal{E}^*)^c \setminus S_2} (1 - p_j) \right)$$

$$= \sum_{\substack{S_1 \subseteq \mathcal{E}^* \\ |S_1| > Q}} \left( \prod_{i \in S_1} p_i \prod_{j \in \mathcal{E}^* \setminus S_1} (1 - p_j) \right) \sum_{S_2 \subseteq (\mathcal{E}^*)^c} \left( \prod_{i \in S_2} p_i \prod_{j \in (\mathcal{E}^*)^c \setminus S_2} (1 - p_j) \right)$$

$$= \sum_{\substack{S_1 \subseteq \mathcal{E}^* \\ |S_1| > Q}} \left( \prod_{i \in S_1} p_i \prod_{j \in \mathcal{E}^* \setminus S_1} (1 - p_j) \right) = \mathbb{P}_{\mathtt{maj}}(\mathcal{E}^*).$$

$\square$

## A.3 Proof for Proposition 2.1

**Discrete Image:** Many configurations of $\theta$ can lead to the same result for $\mathbb{P}_{\mathtt{maj}}(\mathcal{E}, \theta)$.

*Proof.* $\mathbb{P}_{\mathtt{maj}}(\mathcal{E}, \theta)$ is a function with finite number of discontinuities (jumps), which is bounded by $2^N$, the number of possible committees.

Let $\theta = [\theta_1, ..., \theta_N]$ be a set of weights and $\mathcal{S} = \{S_1, ..., S_k\}$ be all the set of configurations for which $\sum_{j \in S_i} \theta_j > Q$ for all $i \in [K]$. Let $S_1$ be the configuration which has the lowest sum of weights, i.e., $S_1 = \arg \min_i \sum_{j \in S_i} \theta_j$. Then, define $\theta^\epsilon$ to be a new set of weights in which some of the weights in $S_1$ are disturbed by $\epsilon$ such that $\sum_{j \in S_1} \theta_j = \sum_{j \in S_1} \theta_j^\epsilon$ and the remaining weights are left unchanged. After this distribution, some weights in $S_1$ are reduced by $\epsilon$, some increased, and some are left unchanged. Let $S_0 = \arg \max_{S \notin \mathcal{S}} \sum_{j \in S} \theta_j$. Then, set $\epsilon \leq \min\{\sum_{j \in S_1} \theta_j - Q, Q - \sum_{j \in S_0} \theta_j\}/N$. Clearly, by this choice of $\epsilon$, all the configurations in $\mathcal{S}$ will pass the aggregation criterion and no other configurations will be added to the $\mathcal{S}$, that is $\mathbb{P}_{\mathtt{maj}}(\mathcal{E}, \theta) = \mathbb{P}_{\mathtt{maj}}(\mathcal{E}, \theta^\epsilon)$. $\square$

## A.4 Remark 3: Wedge Property

**Remark 3.** *Wedge Property: Provided expert competencies, $p_1, p_2 \ldots, p_N$, with the ordering $p_1 \geq p_2 \cdots \geq p_N$, there exists an optimal solution with parameters $\theta^*$ that maximizes $\mathbb{P}_{maj}(\mathcal{E}, \theta)$ and are constrained by the ordering, $\theta_1 \geq \ldots, \geq \theta_N$.*

The intuition behind Remark 3 hinges on the idea that more competent voters would contribute more in terms of voting power in the expert committee compared to less competent experts. Voting power can be measured by various metrics such as the Shapley-Shubik Index or Banzhaf Power Index, which measures the amount of contribution that each voter contributes to swinging the collective decision. Having more weight, or allocated votes, to the expert reflects this property. Therefore we can expect a monotone increasing wedge property on the expert weight $\theta_i$ proportional to the ordering on $p_i$. Enabling this as a constraint for any optimization program will tighten the search space for optimal solutions, making it more computationally efficient.

## A.5 Proof for Lemma 2.2

**Weights Satisfied at Equality:** When solving for the optimal $\theta^*$ for $\theta$-weighted majority voting, we can replace the constraint $Q \leq \sum_{i=1}^N \theta_i$, from Eq. (2.8), with a binding equality $\sum_{i=1}^N \theta_i = 2Q$, to obtain a valid solution. (Proof is available in Appendix A.5.)

*Proof.* We can begin by considering $2Q$, or two times the quota, as the total number of votes in an electorate. We would like to allocate the total number of allowable votes among the voters to ensure total representation, allocating any less could end up with suboptimalities. If the quota, $Q$, is too high in comparison to the total number of votes allotted, then it could be the case that some participants who would have contributed optimally to the voting committee, are unable to participate due to quota limitations. In order to ensure majority voting, more than half of the total votes must pass for any certain choice - by definition. Therefore, between $Q$ and $2Q$, the objective $\mathbb{P}_{\mathtt{maj}}(\theta)$ is strictly non-decreasing, as more useful voters could potentially be included in the voting pool without sacrificing exclusion of others. Since the objective function is strictly non-decreasing w.r.t. in put $\theta$, then for any mixed-integer program, the constraint $Q \leq \sum_{i=1}^N \leq 2Q$ can be replaced by a binding constraint $\sum_{i=1}^N \theta_i = 2Q$.

$\square$

## A.6 Proof for Lemma 3.1

**PAC Sample Complexity:** For any two distributions for experts $i$ and $j$, if the number of samples $t$ and shrinkage term $\mathsf{UCB}_i(t)$, satisfies,

$$\mathsf{UCB}_i^t = \sqrt{\frac{2\sigma_i^2 \log(4/\delta)}{t}}, \quad t_{i,j} \geq \frac{16(\sigma_i^2 + \sigma_j^2) \log(4/\delta)}{\tilde{\epsilon}^2}.$$

then, with probability at least $1 - \delta$, $\delta \in (0, 1)$, the breakage condition, $\mathsf{B}_{ij}^t$, will be met.

*Proof.* Without loss of generality, let us consider $i = 1$ and $j = 2$. Let there exist two sub-Gaussian distributions parameterized with means $p_1, p_2$ and variances $\sigma_1^2, \sigma_2^2$, respectively. Given $\delta \in (0, 1)$, we seek the minimal number of samples $t$ required to distinguish between the two distributions with probability at least $1 - \delta$. Due to Hoeffding's inequality, with respect to the number of samples $t$, the empirical means $\hat{p}_1^t$ and $\hat{p}_2^t$ must satisfy,

$$\mathbb{P}\left(|\hat{p}_i^t - p_i| \geq \mathsf{UCB}_i^t\right) \leq 2 \exp\left(-\frac{t\,(\mathsf{UCB}_i^t)^2}{2\sigma_i^2}\right), \quad i \in \{1, 2\}.$$

Let us set the confidence bound shrinkage parameter $\mathsf{UCB}_i^t$ as,

$$\mathsf{UCB}_i^t := \sqrt{\frac{2\sigma_i^2 \log(4/\delta)}{t}}, \quad i \in \{1, 2\}.$$

The condition for breakage, $\mathsf{B}_{ij}^t$, is indicated by non-overlapping confidence intervals, can be expressed as,

$$|\hat{p}_1^t - \hat{p}_2^t| > \epsilon_1(t) + \epsilon_2(t).$$

Therefore, with probability at least $1 - \delta$, the intervals are separated when

$$\Delta_{12} \geq 2\sqrt{\frac{4(\sigma_1^2 + \sigma_2^2) \log(4/\delta)}{t}} \geq 2\left(\epsilon_1(t) + \epsilon_2(t)\right).$$

Solving for $t_{i,j}$ yields the sample complexity:

$$t_{i,j} \geq \frac{32\sigma^2 \log(4/\delta)}{\tilde{\epsilon}^2} \geq \frac{16(\sigma_i^2 + \sigma_j^2) \log(4/\delta)}{\tilde{\epsilon}^2}.$$

where $\Delta_{ij} := |p_i - p_j|$, $\tilde{\epsilon} = \min_{i \neq j} \Delta_{ij}$, and $\sigma_i \geq \sigma$ for all $i$. $\qquad\square$

## A.7 Proof for Theorem 1

**Regret Bound for** SEE **Algorithm:** The SEE algorithm achieves a total regret $R_T$ bounded by,

$$R_T \in \mathcal{O}\left(\frac{N}{\tilde{\epsilon}^2} \log T\right).$$

*Proof.* **Pairwise Comparison Regret:** During execution, there exist $\binom{N}{2}$ possible pairwise comparisons leading to expert elimination, but the algorithm would encounter at most $N - 1$ elimination events. For each elimination round, we can expect the most samples to be drawn when $\Delta_{ij}$ between experts, $i$ and $j$, is minimized, therefore, the maximum sample complexity to achieve full breakage among all experts is upper-bounded by the sum over $t_{i,i+1}$ from $i$ goes from 1 to $N - 1$. Furthermore, we also impose conditions to limit the sub-optimal inclusion of experts under conservative advantage function estimates (Def. A.2), so long as the UCB is sufficiently sampled such that $2\,\mathsf{UCB} < \epsilon_{\mathtt{maj}}$ (illustrated by Condition A.1). As a consequence Lemma 3.1 will still hold by the same derivation, but we take the smaller of two gaps, where we define $\epsilon_{\mathtt{min}}$ as,

$$\epsilon_{\mathtt{min}} := \min\{\tilde{\epsilon}, \epsilon_{\mathtt{maj}}\}.$$

where $\tilde{\epsilon} = \min_{i \neq j} \Delta_{ij}$ and $\epsilon_{\mathtt{maj}}$ can be computationally derived per Def. A.3. Leveraging the sample complexity result from Lemma 3.1, the regret decomposes as,

$$R_T \leq \sum_{i=1}^{N-1} \left( \underbrace{\frac{16(\sigma_i^2 + \sigma_{i+1}^2)\log(4/\delta)}{\epsilon_{\mathtt{min}}^2}}_{\text{Exploration Cost}} + \underbrace{\delta T \bar{\Delta}}_{\text{Failure Cost}} \right). \tag{A.1}$$

Here, we apply the fact that the optimality gap is upper bounded, $\mathbb{P}_{\mathtt{maj}}(\mathcal{E}, \theta^*) - \mathbb{P}_{\mathtt{maj}}(\mathcal{E}, \theta_t) \leq 1$.

In order to translate the sample complexity result from Lemma 3.1 into a regret bound, we must realize that we expect to play suboptimally with probability $1 - \delta$. Next, $\delta$ represents the probability where a catastrophic breakage even could happen (i.e. the algorithm eliminated expert $i$ from the committee, even though they were supposed to be part of the optimal expert committee). If catastrophic breakage were to occur, we could obtain a upper bounded worst-case regret of $T\bar{\Delta}$, where $\bar{\Delta} \leq 1$ (the maximum expert competency difference) - so essentially a regret of $T$. Therefore, we select $\delta = 1/T$ to offset this issue. By selecting $\delta = 1/T$, the failure cost is effectively an upper bounded by a constant (in this case 1), and the exploration cost scales logarithmically with the number of samples $T$. We can further express Eq. (A.1) as,

$$R_T \leq \sum_{i=1}^{N-1} \left( \frac{32\sigma^2}{\epsilon_{\mathtt{min}}^2} \log(4T) + 1 \right) \in \mathsf{O}\left( \frac{N\sigma^2}{\epsilon_{\mathtt{min}}^2} \log(T) \right).$$

$\square$

## A.8 Mixed Integer Program Formulation for Optimal Weights

**Problem Formulation:** Let $\boldsymbol{\theta} = [\theta_1, \theta_2, \ldots, \theta_N]$ be a continuous positive vector, where $\theta_i \geq 0$ for all $i = 1, 2, \ldots, 2^N$. Let $\mathbf{x} = [x_1, x_2, \ldots, x_{2^N}]$ be a given fixed vector. Define $\mathbf{z} = [\mathsf{Z}_1, \mathsf{Z}_2, \ldots \mathsf{Z}_{2^N}]$ as a binary vector corresponding to each scenario $S$, as determined by the following if-else condition,

$$\mathsf{Z}_S = \begin{cases} 1 & \text{if } \mathbf{x}_i \cdot \boldsymbol{\theta} > Q \\ 0 & \text{otherwise.} \end{cases}$$

where $Q \in \mathbb{R}^+$ is a given constant, and $\mathsf{Z}_S$ is the indicator of the specific scenario. Then, the objective in (2.10) can be written as the dot product $\mathbf{z}^\top \mathbf{p}$, where $\mathbf{p} = [p_1, \ldots, p_{|\mathcal{S}|}]$ and $p_S$, with a slight abuse of notation, is defined by,

$$p_S := \prod_{i \in S} p_i \prod_{j \in S^c} (1 - p_j), \quad S \in \mathcal{S},$$

where $\mathcal{S}$ is the size of all scenarios of size $2^N$.

**Mixed Integer Linear Program Formulation:** The problem can be formulated as a mixed-integer linear program (MIP) as follows:

$$\max_{\theta \in \mathbb{R}^N} \sum_{S \in \mathcal{S}} \mathsf{Z}_S \prod_{i \in S} p_i \prod_{j \in S^c} (1 - p_j) \tag{A.2}$$

subject to:

$$\sum_{j=1}^{N} x_j \theta_j - Q \geq \epsilon_{\texttt{MIP}} \mathsf{Z}_S, \qquad \forall S \in \mathcal{S} \tag{A.3}$$

$$\sum_{j=1}^{N} x_j \theta_j - Q \leq M \mathsf{Z}_S, \qquad \forall S \in \mathcal{S} \tag{A.4}$$

$$\sum_{j=1}^{N} \theta_j = 2Q, \tag{A.5}$$

$$\theta_j - \theta_k \geq \tilde{\epsilon}, \qquad \forall j, k \in \{1, 2, \ldots, N\}, j < k, \tag{A.6}$$

$$\mathsf{Z}_S + \mathsf{Z}_{S^c} \leq 1, \qquad \forall S \in \mathcal{S}, \text{ where } \mathbf{x}_k = \mathbf{1} - \mathbf{x}_i, \tag{A.7}$$

$$\mathsf{Z}_S \in \{0, 1\}, \qquad \forall S \in \mathcal{S}. \tag{A.8}$$

As standard notation for indicator variable $\mathsf{Z}_s$, we define $M$ as a very large number, and $\epsilon_{\texttt{MIP}}$ as a very small number, such that the indicator condition in Eq. (A.2) will hold as specified.

**Description of Constraints** This MIP is formulated with the objective variable as the dot product between a binary selection variable which dictates the possible voting outcomes, and a *possibility variable* $z_i$ which dictates the possibility of this outcome among the constraints . This produces a valid sum over all possibilities and their individual expected values, over which the summation over indicates the expectation over the configuration provided a set of weights $\theta$.

- **Correct Vote Indicators:** The two constraints, (A.3) and (A.4), ensure that $z_i = 1$ if $\sum_{j=1}^{N} x_{ij} \theta_j \geq Q + \epsilon$, or $z_i = 0$ otherwise.

- **Electorate Approval Condition:** the two constraints (A.3) and (A.8), the variable $z_i$ is binary and is determined by the condition $\sum_{j=1}^{N} x_{ij} \theta_j \geq Q + \epsilon$.

- **Equality at Quota:** The constraint (A.5) ensures the total sum of $\theta_j$ is fixed to $2Q$. Lemma 2.2 provides constraint tightening which allows to solve address the quota inequality with a more convenient constraint.

- **$\tilde{\epsilon}$-Incrementality (Heterogeneous Expert Competencies) and $\theta$-Wedge Property:** (A.6) enforces the Assumption 2.1, and the $\theta$-Wedge property in Remark 3, imposing a minimum difference between all pairs of $\theta_i, \theta_j$.

- **Proper Outcome Complements:** The constraint (A.7) ensures that complementary binary vectors $\mathbf{x}_i$ and $\mathbf{x}_k$ cannot both have $z_i = 1$ and $z_k = 1$. (e.g. a specific outcome where expert 1,3,4 votes correctly, cannot have any of 1,3, or 4 also voting incorrectly)

## A.9  Assumption on Summation of Competencies

Let us denote in shorthand, $\mathbf{z}(\mathbf{p}) := \arg\max_{\mathbf{z}} \mathbf{z}^\top \mathbf{p}$ such that $\mathbf{z}(\mathbf{p})$ is the value of $\mathbf{z}$ that maximizes the dot product of any given $\mathbf{p}$. Let $\bar{\mathbf{p}}$ denote the optimistic estimate estimates and $\underline{\mathbf{p}}$ represent the pessimistic estimates of scenario probabilities based on UCB uncertainty for expert competencies.

**Assumption A.1.** *For scenario probabilities $\mathbf{p}$ and scenario indicator vector $\mathbf{z} \in \{0,1\}^{|\mathcal{S}|}$, the inequality holds such that,*

$$\sum_{i=1}^{N} \bar{p}_i - \mathbf{z}^\top \bar{\mathbf{p}} \geq \sum_{i=1}^{N} \underline{p}_i - \mathbf{z}^\top \underline{\mathbf{p}}, \tag{A.9}$$

*where, $\bar{p}_i$ and $\underline{p}_i$ refer to the maximum and minimum probabilities subject to UCB bounds over competency estimates, i.e. $p_i \pm \mathsf{UCB}, \forall p_i \in p_1, \ldots, p_N$.*

To provide additional discourse on Assumption A.1, we assume that the variation in $\mathbf{z}(\bar{\mathbf{p}})^\top \bar{\mathbf{p}} - \mathbf{z}(\underline{\mathbf{p}})^\top \underline{\mathbf{p}}$ is sufficiently small compared to the fluctuation in the shrinking UCB. In the worst case scenario, for each $2^{N-1}$ scenario probabilities, we could add a deviation term of $N \times \mathsf{UCB}$ for each scenario (although this is a highly conservative estimate).

**Bonferroni Expansion:** The *Bonferroni inequalities* provide a sequence of upper and lower bounds on the probability of the union of $N$ events, refining the standard inclusion-exclusion principle. These inequalities alternate between over- and under-estimates as more terms are included, ultimately converging to the exact probability when all $N$ terms are considered.

For events $A_1, A_2, \ldots, A_N$, the exact probability of their union is given by:

$$P\left(\bigcup_{i=1}^{N} A_i\right) = \sum_{k=1}^{N} (-1)^{k+1} \mathsf{S}_k,$$

where $\mathsf{S}_k$ represents the sum of the probabilities of all $k$-wise intersections:

$$\mathsf{S}_k = \sum_{1 \leq i_1 < i_2 < \cdots < i_k \leq N} P\left(A_{i_1} \cap A_{i_2} \cap \cdots \cap A_{i_k}\right).$$

**General $k$-th Order Bonferroni Inequality:** The Bonferroni inequalities state that truncating the inclusion-exclusion series at the $k$-th term yields an upper or lower bound depending on whether $k$ is odd or even. If $k$ is odd, the partial sum is an upper bound,

$$P\left(\bigcup_{i=1}^{N} A_i\right) \leq \sum_{j=1}^{k} (-1)^{j+1} \mathsf{S}_j.$$

If $k$ is even, the partial sum is a lower bound,

$$P\left(\bigcup_{i=1}^{N} A_i\right) \geq \sum_{j=1}^{k} (-1)^{j+1} \mathsf{S}_j.$$

Let $(A_1, A_2, A_3, \ldots, A_N)$ be $N$ independent events with success probabilities $(P(A_1), P(A_2), P(A_3), \ldots, P(A_N))$,

$$P\left(\bigcup_{i=1}^{N} A_i\right) = P(A_1) + P(A_2) + P(A_3) \cdots + P(A_N) - P(A_1 \cap A_2) - P(A_1 \cap A_3) - P(A_2 \cap A_3) \cdots$$
$$+ P(A_1 \cap A_2 \cap A_3) \cdots - \cdots \pm P(A_1 \cap A_2 \cap A_3 \cdots \cap A_N)$$

Given expert competencies, we could express,

$$P\left(\bigcup_{i=1}^{N} A_i\right) = \underbrace{p_1 + p_2 + p_3 \cdots + p_N}_{\mathsf{S}_1} - \underbrace{p_1 p_2 - p_1 p_3 - p_2 p_3 \ldots}_{\mathsf{S}_2} + \underbrace{p_1 p_2 p_3 \ldots}_{\mathsf{S}_3} - \cdots \pm \underbrace{\prod_{i=1}^{N} p_i}_{\mathsf{S}_N}$$

If we were to vary $p_i \pm \mathsf{UCB}$, we would incorporate this modification into all of the terms in the Bonferroni expansion. Given the alternating $\pm$ structure of the Bonferroni expansion, Ass. A.1 posits that the difference between, any two MIP solutions under $\bar{p}_i$ versus $\underline{p}_i$ would entail a difference of,

$$\mathbf{z}^\top \bar{\mathbf{p}} - \mathbf{z}^\top \underline{\mathbf{p}} = \sum_{i=1}^{N} \mathsf{Z}_s \bar{p}_i \ - \mathsf{Z}_s \bar{p}_1 \bar{p}_2 - \mathsf{Z}_s \bar{p}_1 \bar{p}_3 - \mathsf{Z}_s \bar{p}_2 \bar{p}_3 \cdots + \mathsf{Z}_s \bar{p}_1 \bar{p}_2 \bar{p}_3 \cdots - \cdots \pm \mathsf{Z}_s \prod_{i=1}^{N} \bar{p}_i$$
$$- \sum_{i=1}^{N} \mathsf{Z}_s \underline{p}_i \ \underbrace{+ \mathsf{Z}_s \underline{p}_1 \underline{p}_2 + \mathsf{Z}_s \underline{p}_1 \underline{p}_3 + \mathsf{Z}_s \underline{p}_2 \underline{p}_3 \cdots - \mathsf{Z}_s \underline{p}_1 \underline{p}_2 \underline{p}_3 \cdots + \cdots \pm \mathsf{Z}_s \prod_{i=1}^{N} \underline{p}_i}_{\text{Higher order terms.}}$$

Where $\bar{p}_i = p_i + \mathsf{UCB}$, and $\underline{p}_i = p_i - \mathsf{UCB}$, and $\mathsf{Z}_s$ is the indicator variable for whether the specific scenario is possible, and equivalent in both the upper and lower bounding formulations. Ass. A.1 posits that the higher order terms of the Bonferroni expansion serve to cancel each other out, such that as $T \to \infty$, and $\mathsf{UCB} \to 0$,

$$\mathbf{z}^\top \bar{\mathbf{p}} - \mathbf{z}^\top \underline{\mathbf{p}} \le \sum_{i=1}^{N} \bar{p}_i - \sum_{i=1}^{N} \underline{p}_i. \tag{A.10}$$

This is true in most cases where the values of the expert competencies are spread out over $[0, 1]$ and not highly concentrated around a single number, which is supported by Ass. 2.1. It also tends to be true as we have a modest size of experts, as in our setting. It is possible to check, whether Ass. A.1 holds provided $\bar{p}_i$ and $\underline{p}_i$ as UCB-based estimates input into the MIP. Our empirical analysis has demonstrated that this assumption holds in all tested configurations of expert competencies.

## A.10 Bound on Max-Min Probabilities for Weighted Majority Voting

**Lemma A.1.** *Under the Assumption A.1, the regret of any learning algorithm (Def. 3.1) which provides bounds on the individual agent competencies $p_i$ in a majority voting setting can be bounded by,*

$$\mathtt{reg}_t \le \sum_{i=1}^{N} \bar{p}_i - \sum_{i=1}^{N} \underline{p}_i. \tag{A.11}$$

*Proof.* Under full-bandit feedback, as we draw more samples the estimate of each individual expert's competencies improve, and the confidence bound shrinks, and with high confidence the estimate of each competency is bounded by, $p_i \pm \mathsf{UCB}$. Recall that the probability of each scenario $S \in \mathcal{S}$ denoted by $p_S$ is given by,

$$p_s = \prod_{i \in S} p_i \prod_{j \in S^c} (1 - p_j), \quad S \subseteq \mathcal{S},$$

where $\mathcal{S}$ contains $2^N$ elements. Let $\mathbf{z}^*$ be the binary vector that maximizes $\mathbf{z}^\top \mathbf{p}$ for a given $\mathbf{p}$. We further subject $\mathbf{z}$ to a constraint,

$$\sum_s^{|\mathcal{S}|} Z_S \leq \frac{1}{2}|\mathcal{S}|, \quad \forall Z_S \in \mathbf{z} \tag{A.12}$$

The constraint in Eq (A.12) represents the restriction on $Z_S$, such that if $Z_S = 1$ then its complement $Z_{S^c} = 0$. Which, giving winning condition for tie-free weighted majority voting $\sum_{i=1}^N \theta_i > Q$, and $Q \geq N/2$, implies that the summation of $Z_S$ over all scenarios cannot be greater than half of all scenarios $0.5|\mathcal{S}|$, because its scenario complement, $S^c$, must be not be possible. This constraint is imposed because $Z_S \in \{0,1\}^{|\mathcal{S}|}$ represents the possibility of the scenario in the set of all possible scenarios, and any possible scenario always excludes its complement, $S^c$, therefore $Z_{S^c} = 0$, and the sum over $\mathcal{S}$ is at most one half of $|\mathcal{S}|$. Let $\mathbf{p}$ represent the probability vector of $2^N$ possible scenarios in $\mathcal{S}$. Therefore, the regret is expressed as,

$$\texttt{reg}_t = \max_{\mathbf{z}} \mathbf{z}^\top \mathbf{p}^* - \mathbf{z}(\mathbf{p})^\top \mathbf{p}^* \tag{A.13}$$

**Establishing the Inequality:** We establish the key inequality to bound our regret,

$$\mathbf{z}(\mathbf{p})^\top \underline{\mathbf{p}} \leq \mathbf{z}(\mathbf{p})^\top \mathbf{p}^* \leq \max_{\mathbf{z}} \mathbf{z}^\top \mathbf{p}^* \leq \mathbf{z}(\bar{\mathbf{p}})^\top \bar{\mathbf{p}}. \tag{A.14}$$

We consider the Bonferroni inequalities, via first order Bonferroni inequality,

$$\max_{\mathbf{z}} \mathbf{z}^\top \mathbf{p} \leq P\left(\bigcup_{i=1}^N A_i\right) \leq \sum_{i=1}^N p_i \tag{A.15}$$

Where $A_i$ are defined as probability of success of expert $i$, for $N$ experts. The probability of the union of $A_i$, that is at least one experts predicts correctly, is always greater than or equal to the probability that at least $N/2$ experts predict correctly (majority condition). Therefore, for any vector $\mathbf{z}$ adhering to constraints from Eq. (A.12), we can use the union bound over all expert competencies to upper bound the majority correct vote probability, leading to upper bounds,

$$\mathbf{z}^\top \bar{\mathbf{p}} \leq \sum_{i=1}^N \bar{p}_i, \qquad \mathbf{z}^\top \underline{\mathbf{p}} \leq \sum_{i=1}^N \underline{p}_i. \tag{A.16}$$

From the bound on Eq. (A.15), combined with constraints on the sensitivity of the competency estimates as limited by Assumption A.1 we can obtain bounds on the simple regret of the algorithm under uncertainty of expert competencies,

$$\texttt{reg}_t \leq \sum_{i=1}^N \bar{p}_i - \sum_{i=1}^N \underline{p}_i. \tag{A.17}$$

$\square$

## A.11 Proof of Theorem 2

**Regret Bound for $\theta$-WMV Algorithm:** An algorithm based on $\theta$-MIP weight assignment, such as in Algorithm 2, achieves a regret bound of $R_T \in \mathsf{O}\left(\sqrt{NT\log(T)}\right)$.

*Proof.* Provided the result from Lemma A.1, we simply account for the uncertainty difference between the two estimated sums for $\sum_{n=1}^N p_i$ and $\sum_{n=1}^N \bar{p}_i$. Let $S_\Sigma = \sum_{n=1}^N p_i$, denotes the true summed competencies, thus any deviation from $S_\Sigma$ via estimating $p_i$ will give the PAC bound of $\sum_{i=1}^N (\bar{p}_i - p_i)$. Let us express the uncertain quantity as UCB, where,

$$\mathsf{UCB}_\Sigma = \left| \sum_{i=1}^{N} \hat{p}_i - S_\Sigma \right|.$$

**PAC Bounds for Summation of Binomial Trials:** The $\mathsf{UCB}_\Sigma$ represents the radius that the estimate $\sum_{i=1}^{N} \hat{p}_i$ should fall within with probability $1 - \delta$. Consider $N$ independent binomial experiments, where the $i$-th experiment consists of $n_i$ trials with success probability $p_i$. Let $\hat{p}_i = \frac{k_i}{n_i}$ denote the empirical probability of success, where $k_i$ is the number of successes observed in the $i$-th experiment. We are interested in deriving a $1 - \delta$ PAC bound for the sum of the true probabilities $S_\Sigma = \sum_{i=1}^{N} p_i^*$.

**Hoeffding's Inequality for Unequal $n_i$:** Using Hoeffding's inequality, the deviation of the empirical sum $\sum_{i=1}^{N} \hat{p}_i$ from the true sum $S_\Sigma$ can be bounded. Let $\{X_{i,j}\}_{j=1}^{n_i}$ be independent Bernoulli random variables with parameter $p_i \in [0,1]$, for $i = 1, \dots, N$. That is, for each fixed $i$,

$$X_{i,j} \sim \text{Bernoulli}(p_i), \quad \text{i.i.d. for } j = 1, \dots, n_i,$$

and the samples are independent across different $i$. Our goal is to establish a concentration inequality for the sum of these empirical means, $S_\Sigma$, around its expectation, $\mu$. Where,

$$\hat{p}_i := \frac{1}{n_i} \sum_{j=1}^{n_i} X_{i,j}, \qquad S_\Sigma := \sum_{i=1}^{N} \hat{p}_i, \qquad \mu := \sum_{i=1}^{N} p_i. \qquad \text{(A.18)}$$

Each $\hat{p}_i$ is an average of $n_i$ independent Bernoulli random variables bounded in $[0,1]$. Hence, $\hat{p}_i$ itself lies in the interval $[0,1]$. We can write,

$$\sum_{i=1}^{N} \hat{p}_i = \sum_{i=1}^{N} \frac{1}{n_i} \sum_{j=1}^{n_i} X_{i,j}.$$

Consider the collection of all $M := \sum_{i=1}^{N} n_i$ independent random variables $\{X_{i,j}\}$. The sum $S$ is a weighted sum of these variables, where each $X_{i,j}$ is multiplied by $1/n_i$.

**Hoeffding-Type Concentration Bound:** For any $\epsilon > 0$, it holds that (see Lemma A.4),

$$\mathbb{P}\left( \left| \sum_{i=1}^{N} \hat{p}_i - \sum_{i=1}^{N} p_i \right| \geq \epsilon \right) \leq 2 \exp\left( -\frac{2\epsilon^2}{\sum_{i=1}^{N} \frac{1}{n_i}} \right).$$

To achieve a $1 - \delta$ PAC bound, we set the right-hand side to $\delta$ and solve for $\epsilon$:

$$\delta = 2 \exp\left( -\frac{2\epsilon^2}{\sum_{i=1}^{N} \frac{1}{n_i}} \right), \qquad \epsilon = \sqrt{ \frac{\ln(2/\delta)}{2} \cdot \sum_{i=1}^{N} \frac{1}{n_i} }. \qquad \text{(A.19)}$$

Thus, with probability at least $1 - \delta$:

$$\left| \sum_{i=1}^{N} \hat{p}_i - S_\Sigma \right| \leq \sqrt{ \frac{\ln(2/\delta)}{2} \cdot \sum_{i=1}^{N} \frac{1}{n_i} }. \qquad \text{(A.20)}$$

The term $\sum_{i=1}^{N} \frac{1}{n_i}$ captures the variability introduced by the unequal number of trials $n_i$. Experiments with smaller $n_i$ contribute more to the bound due to their higher uncertainty, while experiments with larger $n_i$ contribute less due to their lower uncertainty.

Suppose in the complete full bandit feedback, we always play every single arm,, therefore at each time interval, we would play each expert exactly $N$ times, therefore $n_i = t$, for all experts at round

$t$. We can then use an integral to bound a summation as,

$$\sum_{t=1}^{T} \sqrt{\frac{\ln(2/\delta)}{2} \cdot \sum_{i=1}^{N} \frac{1}{n_i}} = \sum_{t=1}^{T} \sqrt{\frac{\ln(2/\delta)}{2} \cdot \sum_{i=1}^{N} \frac{1}{t}} \leq \sqrt{N \frac{\ln(2/\delta)}{2}} \int_{1}^{T} \sqrt{\frac{1}{t}} \, dt,$$

To solve for the definite integral,

$$\int_{1}^{T} \frac{1}{\sqrt{t}} \, dt = \int_{1}^{T} t^{-1/2} \, dt \tag{A.21}$$

$$\left[ 2t^{1/2} \right]_{1}^{T} = 2T^{1/2} - 2(1)^{1/2} = 2\sqrt{T} - 2 \tag{A.22}$$

Putting this back into the original inequality,

$$\sqrt{\frac{\ln(2/\delta)}{2} \cdot \sum_{i=1}^{N} \frac{1}{n_i}} \leq \sqrt{\frac{\ln(2/\delta)}{2}} 2\sqrt{N} \left( \sqrt{T} - 1 \right) \tag{A.23}$$

Eq. (A.25) bounds the high probability estimation uncertainty of the algorithm, where at each time step the regret can be bounded in the high probability regime. Therefore, we can express the expected regret from 1 to $T$ as,

$$R_T \leq \sqrt{2N \ln(2/\delta)} \left( \sqrt{T} - 1 \right) + T \bar{\Delta} \delta \tag{A.24}$$

We can choose $\delta = \frac{1}{T}$, allowing it to vanish. Therefore, the expected regret is bounded by,

$$R_T \in \mathsf{O} \left( \sqrt{NT \log(T)} \right) \tag{A.25}$$

$\square$

**Worst-Case Bound:** It is possible to check, whether Ass. A.1 holds provided $\bar{p}_i$ and $\underline{p}_i$, and should this condition not hold, the algorithm will still retain its no-regret properties, however, it will incur a looser regret bound, illustrated in Appendix A.16.

### A.12 Proof for Corollary 3.1

**Targeted-$m$ Setting:** An algorithm exists that allows for the regret bounds of Theorems 1 and 2 to hold with an additional multiplicative $N/m$ factor (e.g., SEE's regret becomes $\mathcal{O} \left( \frac{N^2}{m\bar{\epsilon}^2} \log T \right)$).

*Proof.* The algorithms presented in the previous sections were developed under the assumption that all experts could be queried at each round. However, this assumption may be restrictive in certain applications where only $m$ experts are available at any round. We refer to this setting as targeted-$m$, and in the following, we propose an extension of the previous algorithm to address this constraint

A simple modification of the previous algorithm can be applied to the targeted-$m$ setting. Specifically, we partition the $N$ experts into $(N/m)$ groups of size $m$ each. Then, we run a burn-in period during which all experts within each group are queried for $t_0$ rounds, where $t_0$ is defined in Lemma 3.1. This results in the total $t_0 \cdot (N/m)$ number of queries. The purpose of the burn-in period is to ensure that any breakage event, if it occurs, does so with high probability. Afterwards, we apply Algorithms 1 or 2 for assembling the OEC or finding the optimal weights in the weighted majority voting problem, respectively. Therefore, the regret bounds of Theorems 1 and 2 remain valid with an additional factor of $N/m$, e.g., the regret of SEE in the Targeted-$m$ setting will be $\mathcal{O}(\frac{N^2}{m\bar{\epsilon}^2} \log T)$. $\square$

### A.13 Binary Outcome Expert Weighting

**Lemma A.2.** *Optimal Majority Voting Expert Weights for Binary Classification: From Nitzan and Paroush [24] , in a weighted majority voting scenario, as described in Sec. 2.2, when the space of outcomes is binary (binary classification), the optimal weighting that maximizes the combined predictive accuracy for all experts combined is,*

$$\beta_i = \log \frac{p_i}{1 - p_i}. \tag{A.26}$$

*Where $p_i$ is the competency of expert $i$ (the probability that the expert predicts correctly.)*

*Proof.* We begin with a few key assumptions. First as stated, that the space of predictive outcomes is binary, the expert predicts true, they receive a value of $1$, and false a value of $0$. Let $y$ denote a set of outcomes (i.e. $y = [1, 0, 1, 1, 0]$ indicates that experts 1, 3, 5 selected the correct answer out of 5 experts). Due to the binary aspect of the outcomes, if an expert consistently predicts the incorrect outcome, we can always negate their prediction (predict the opposite) to obtain a equally consistently correct outcome. Further, let us define an aggregation function $f(y) : y \mapsto \{0, 1\}$ that takes any outcome and aggregates all individual outcomes to form a single outcome in $\{0, 1\}$.

**Pairing with Compliments:** Suppose we impose an arbitrary pairing over outcomes. Let $A(y)$ denote the selection over $y$ where we impose as the positive outcomes voters. Let $B(y)$ denote its compliment. Thus for any outcome $y = A(y) \cup B(y)$. This implies that if all outcomes are 1 in $A(y)$ the outcome of $f(y)$ will result in an overall positive outcome $y = 1$, $\forall y \in A(y) \implies f(y) = 1$. The compliment, $B(y)$ is by definition mutually exclusive from $A(y)$ therefore it is not possible to include both in the selection voting criteria (either $A(y)$ or $B(y)$ for any outcome $y$.) We also can see that there is a deterministic relations that maps any selection $A(y)$ to its compliment $B(y)$ (i.e. if $A(y) = [1, 0, 1, 1, 0] \implies B(y) = [0, 1, 0, 0, 1]$.) Therefore, the set of all outcomes $\mathcal{Y}$ is paired in such a way. Let us denote this as $(A_k(y), B_k(y))$ for each pair.

**Maximizing $\mathbb{E}[f(y)]$:** In order to maximize the probability that $f(y) = 1$, for each pair $(A_k(y), B_k(y))$ we must determine a selection rule that selects the greater of $P(A_k(y))$ or $P(B_k(y))$. For notation purposes, we impose that,

$$P(A_k(y)) \geq P(B_k(y)), \quad \forall k \in \mathcal{K} \implies \mathbb{E}[f(y)] = \mathbb{E}[f^*(y)] \tag{A.27}$$

Should a selection rule be found such that Eq. (A.27) holds, then the value $\mathbb{E}[f(y)]$ is maximized. This simply holds because all outcomes are assigned to pairs, and events in each are mutually exclusive (or condition), and events between pairs are not (and condition). We can express this probability as,

$$P(A(y)) = \prod_{i \in A} p_i \prod_{j \in B} (1 - p_j) \geq \prod_{i \in A} (1 - p_i) \prod_{j \in B} p_j = P(B(y)). \tag{A.28}$$

This forms a sufficient condition, so long as the selection rule fulfills Eq. (A.28).

**Selection Rule:** It just so happens that our selection rule $f(y)$ selects $f(y) = 1$, then the weighted sum of outcomes in $A(y)$ must exceed that of $B(y)$ by definition. Thus,

$$\sum_{i \in A} \beta_i y_i \geq \sum_{j \in B} \beta_j \texttt{Flip}(y_j) \tag{A.29}$$

Where `Flip` represents the *bitwise not operator* (for example $[1, 1, 0] \rightarrow [0, 0, 1]$). This forms a necessary condition for the selection rule to output $f(y) = 1$.

**Log-Odds Weights:** Given the sufficient and necessary conditions to produce $\mathbb{E}[f^*(y)]$ outlined in Eq. (A.28) and Eq. (A.29). We can readjust Eq. (A.28) such that,

$$\prod_{i \in A} \frac{p_i}{(1 - p_i)} \geq \prod_{j \in B} \frac{p_j}{(1 - p_j)} \tag{A.30}$$

Taking the logarithm, we obtain,

$$\sum_{i \in A} \log \frac{p_i}{(1 - p_i)} \geq \sum_{j \in B} \log \frac{p_j}{(1 - p_j)} \tag{A.31}$$

The optimal weights therefore are equivalent to,

$$\beta_i = \log \frac{p_i}{(1 - p_i)} \tag{A.32}$$

Likewise for $j$ as well, mutatis mutandis. To summarize, the sufficient and necessary conditions from Eq. (A.28) and Eq. (A.29) enforce a single unique solution for the optimality condition of $\mathbb{E}[f(y)]$ expressed in (A.32).

$\square$

## A.14 Relation Between the Optimal Binary Classification and Weighted Majority Voting

The setting of Nitzan and Paroush [24] considers first, arbitrary neutral decision rules, whereas we restrict our analysis to a specific class of majority voting systems operating over sub-committees. Consequently, the binary voting framework of Nitzan and Paroush [24] encompasses a strictly larger set of decision rules, which allows the approach to achieve superior performance only in the binary classification setting.

For example, consider a simple setting in which there are only two experts with competencies $\{p_1 > p_2 > 0.5\}$. According to Nitzan and Paroush [24], the optimal neutral decision rule is given by $f(x_1, x_2) = \text{sign}(w_1 x_1 + w_2 x_2)$, where $w_i = \log(p_i/(1 - p_i)) > 0$ and $x_i \in \{1, -1\}$ is the expert $i$'s decision. The success probability of this decision rule will be,

$$P\left(f(x_1, x_2) = 1\right) = P\left(\text{sign}(w_1 x_1 + w_2 x_2) = 1\right) = P\left(x_1 = 1, x_2 = 1\right) + P\left(x_1 = 1, x_2 = -1\right)$$
$$= p_1 p_2 + p_1(1 - p_2) = p_1.$$

This performance can be achieved using the WMV scheme by considering $\theta_1 = 1$ and $\theta_2 = 0$ and quota $Q = 0.5$ in (2.7). In this case, (2.7) will be

$$\mathbb{P}_{\texttt{maj}}(\mathcal{E}, \theta) = p_1 p_2 + p_1(1 - p_2) = p_1.$$

Now, suppose $\{p_1 > 0.5 > p_2\}$ such that $w_1 < |w_2|$. For example, $p_1 = 0.6$ and $p_2 = 0.1$. In this case, $f(x_1, x_2)$ will always flip the decision of expert 2 and ignore expert 1 which leads to the success probability of $1 - p_2 = 0.9$. But this performance cannot be achieved by the WMV. Because, if the optimal committee is only expert 1, then the success probability is 0.6. If the optimal committee contains both of them, then the success is $0.6 \cdot 0.1 + 0.6 \cdot 0.9 + 0.4 \cdot 0.1 = 0.64$.

## A.15 Non-Contradiction of Expert Inclusion

**Definition A.1.** *Inclusion Signal: The inclusion signal of the advantage function $\mathfrak{F}(\mathcal{E}_1, \mathcal{E}_2)$ is defined as,*

$$\mathbb{1}(\mathfrak{F}(\mathcal{E}_1, \mathcal{E}_2) > 0).$$

Given $\mathcal{E}_1$ and $\mathcal{E}_2$ are disjoint subsets of $\mathcal{E}$, $\mathbb{1}(\mathfrak{F}(\mathcal{E}_1, \mathcal{E}_2) > 0)$ serves an indicator representing whether the advantage function is positive. This indicates if it is advantageous to retain the expert set $\mathcal{E}_2$ in the original $\mathcal{E} = \mathcal{E}_1 \cup \mathcal{E}_2$ or truncate it to $\mathcal{E}_1$.

**Definition A.2.** *Conservative Advantage Function: Given two mutually exclusive subsets $\mathcal{E}_1 \subseteq \mathcal{E}$ and $\mathcal{E}_2 \subseteq \mathcal{E}$, where $\mathcal{E}_1 \cap \mathcal{E}_1 = \emptyset$, for optimistic and pessimistic estimates expert competencies of $p_i$, denoted as $\bar{p}_i$ and $\underline{p}_i$ respectively, the conservative advantage function is defined as*

$$\mathfrak{F}(\underline{\mathcal{E}}_1, \overline{\mathcal{E}}_2),$$

*such that,*

$$\overline{\mathcal{E}} = \{\bar{p}_1, ..., \bar{p}_N\}, \qquad \underline{\mathcal{E}} = \{\underline{p}_1, ..., \underline{p}_N\}.$$

That is, $\overline{\mathcal{E}}$ represents the egalitarian voting committee substituting the optimistic estimates of the expert competencies $\bar{p}_i$, and $\underline{\mathcal{E}}$ represents the egalitarian voting committee substituting the pessimistic estimates of expert competencies, $\underline{p}_i$.

**Remark 4.** *Guarantee on Consistent Exclusion: Given a bisection $\mathscr{B}_{12}$, that divides two an ordering over a set of experts, $\mathcal{E}$, into two ordered sets, $\mathcal{E}_1$ and $\mathcal{E}_2$, where $p_i > p_j$ for all $i \in \mathcal{E}_1$ and $j \in \mathcal{E}_2$, it follows that,*

$$\mathfrak{F}(\underline{\mathcal{E}}_1, \overline{\mathcal{E}}_2) \le 0 \implies \mathfrak{F}(\mathcal{E}_1, \mathcal{E}_2) \le 0.$$

We begin by asserting that any committee with dominant competencies for all experts (i.e. $N$ experts, all experts have higher competencies in one committee than another) will result in $\mathbb{P}_{\mathtt{maj}}(\mathcal{E}) \le \mathbb{P}_{\mathtt{maj}}(\overline{\mathcal{E}})$, and it follows, $\mathbb{P}_{\mathtt{maj}}(\mathcal{E}_1, \mathcal{E}_2) \le \mathbb{P}_{\mathtt{maj}}(\mathcal{E}_1, \overline{\mathcal{E}}_2)$. When we are provided a condition $\mathfrak{F}(\mathcal{E}_1, \overline{\mathcal{E}}_2) \le 0$ this implies $\mathbb{P}_{\mathtt{maj}}(\mathcal{E}_1, \overline{\mathcal{E}}_2) \le \mathbb{P}_{\mathtt{maj}}(\mathcal{E}_1)$, by the definition from Eq. (2.6). Therefore, $\mathbb{P}_{\mathtt{maj}}(\mathcal{E}_1, \mathcal{E}_2) \le \mathbb{P}_{\mathtt{maj}}(\mathcal{E}_1, \overline{\mathcal{E}}_2) \le \mathbb{P}_{\mathtt{maj}}(\mathcal{E}_1) \implies \mathbb{P}_{\mathtt{maj}}(\mathcal{E}_1, \mathcal{E}_2) \le \mathbb{P}_{\mathtt{maj}}(\mathcal{E}_1) \implies \mathfrak{F}(\mathcal{E}_1, \mathcal{E}_2) \le 0$. We can conclude that, $\mathfrak{F}(\mathcal{E}_1, \overline{\mathcal{E}}_2) \le 0 \implies \mathfrak{F}(\mathcal{E}_1, \mathcal{E}_2) \le 0$. Notice that during breakage $B_{ij}$, the bisection property $\mathscr{B}_{12}$ naturally divides $\mathcal{E}$ into two sets at the barrier between $i$ and $j$ (see Diagram 2), and we can be free to select $\underline{\mathcal{E}}_1$ from within $\mathcal{E}_1$ without contradicting the bisection property (i.e. $p_i > p_j$ for all $i \in \mathcal{E}_1$ and $j \in \mathcal{E}_2$). Therefore, it holds that,

$$\mathfrak{F}(\underline{\mathcal{E}}_1, \overline{\mathcal{E}}_2) \le 0 \implies \mathfrak{F}(\mathcal{E}_1, \mathcal{E}_2) \le 0.$$

In other words, if a conservative advantage function is applied, when this conservative advantage function, $\mathfrak{F}(\underline{\mathcal{E}}_1, \overline{\mathcal{E}}_2)$ signals to exclude $\mathcal{E}_2$ from $\mathcal{E}_1 \cup \mathcal{E}_2$, then with high probability $1 - \delta$, this exclusion of $\mathcal{E}_2$ from the optimal committee is valid.

**Condition A.1.** *Consistent Inclusion: Provided an egalitarian voting committee with experts $\mathcal{E}$, and their respective competencies $p_1 \ldots p_N$, where two mutually exclusive subsets $\mathcal{E}_1, \mathcal{E}_2$ are formed from the breakage event, $B_{ij}$, such that $p_i > p_j$, $\forall i \in \mathcal{E}_1, \forall j \in \mathcal{E}_2$ (see Diagram 2), and sufficient samples are drawn, such that $2\mathsf{UCB} < \epsilon_{\mathtt{maj}}$, it follows that the inclusion signal (see Def. A.1) of the conservative advantage function, $\mathfrak{F}(\underline{\mathcal{E}}_1, \overline{\mathcal{E}}_2)$, must be consistent with the inclusion signal computed under the true competencies, $\mathfrak{F}(\mathcal{E}_1, \mathcal{E}_2)$, i.e.,*

$$\mathfrak{F}(\underline{\mathcal{E}}_1, \overline{\mathcal{E}}_2) > 0 \iff \mathfrak{F}(\mathcal{E}_1, \mathcal{E}_2) > 0.$$

Note that the expert elimination condition checks the condition $\mathfrak{F}(\underline{\mathcal{E}}_1, \overline{\mathcal{E}}_2) \le 0$ for any candidate set, where $p_i > p_j$, $\forall i \in \underline{\mathcal{E}}_1, \forall j \in \overline{\mathcal{E}}_2$ with high probability, $1 - \delta$. It is therefore not possible to falsely eliminate $\mathcal{E}_2$ from $\mathcal{E}$ with high confidence, as we are stipulating the lowest estimates of competence for $\mathcal{E}_1$ (group with higher competence) and highest estimates of competence for $\mathcal{E}_2$ (group with lower competence) - if no advantage exists here, then we can confidently eliminate $\mathcal{E}_2$ from $\mathcal{E}$. In the reverse situation without additional constraints, the inclusion signal under the conservative estimate of $\mathfrak{F}(\underline{\mathcal{E}}_1, \overline{\mathcal{E}}_2)$ may be inconsistent with the inclusion signal under true competencies $\mathfrak{F}(\mathcal{E}_1, \mathcal{E}_2)$ (i.e. the signs of the two signals are different). Therefore, we next define $\epsilon_{\mathtt{maj}}$ as a property of the expert configuration such that consistency is achieved, per Condition A.1.

**Definition A.3.** $\epsilon_{\mathtt{maj}}$*-Consistent Gap: We denote a consistency gap, $\epsilon_{maj} \in [0, 1]$, as the maximum value such that Condition A.1 holds.*

For any configuration of experts, there must exist a $\epsilon_{\mathtt{maj}}$ such that Condition A.1 holds, as there are no constraints on how small $\epsilon_{\mathtt{maj}}$ can be. When we draw enough samples such that $2\mathsf{UCB} < \epsilon_{\mathtt{maj}}$, then we will obtain a consistent elimination function, when our confidence of expert competencies is less than the gap needed for always computing inclusion-consistent advantage functions with high confidence. Given the properties of the problem, i.e. expert competency arrangements $\{p_i\}$, an exact computation can be performed to determine $\epsilon_{\mathtt{maj}}$, and the value of $\epsilon_{\mathtt{maj}}$ is independent of the number of the number of experts $N$ or number of samples $T$. Let $\{\mathscr{B}_{12}\}$ denote the set of bisections of $\mathcal{E}$ (i.e. division of the set of experts into two sets post breakage, where $p_i > p_j$ for all $i \in \mathcal{E}_1$ and

$j \in \mathcal{E}_2$). The range of admissible $\epsilon_{\texttt{maj}}$ values can be expressed as,

$$\mathcal{G}(\mathcal{E}) := \left\{ \mathsf{UCB} \in \mathbb{R}^+ \,|\, \mathrm{Sign}\left(\mathfrak{F}(\underline{\mathcal{E}}_1, \bar{\mathcal{E}}_2)\right) = \mathrm{Sign}\left(\mathfrak{F}(\mathcal{E}_1, \mathcal{E}_2)\right) \right\}, \quad \forall(\mathcal{E}_1, \mathcal{E}_2) \in \{\mathscr{B}_{12}\} \qquad \text{(A.33)}$$

We express $\epsilon_{\texttt{maj}}$ as,

$$\epsilon_{\texttt{maj}} = \underset{\mathsf{UCB} \in \mathcal{G}(\mathcal{E})}{\arg\max} \ \epsilon. \qquad \text{(A.34)}$$

## A.16    Alternative Bound for Theorem 2

We provide an alternative upper bound for Theorem 2 which upper bounds the regret of $\theta$-WMV algorithm. As preliminaries, we introduce the definition for $\mathcal{S}(\theta)$ and Lemma A.3 which stipulate that any two sets in $\mathcal{S}(\theta)$ cannot be disjoint.

**Definition A.4.** *For a given set of weights, we denote by $\mathcal{S}(\theta)$ the collection $\{S_1, ..., S_k\}$ of all configurations of experts such that $\sum_{k \in S_i} \theta_k > Q$ for all $i$ and $S_i \nsubseteq S_j$ for all $i \neq j$.*

**Lemma A.3.** *Any two sets in $\mathcal{S}(\theta)$ have non-empty intersection.*

*Proof.* Otherwise, we have at least $i$ and $j$ such that $S_i \cap S_j = \emptyset$. Then, $\sum_{k \in S_i \cup S_j} \theta_k > 2Q$ while due to constraint $\|\theta\| \leq 2Q$. This is a contradiction. $\qquad \square$

**Theorem 3.** *Regret Bound for $\theta$-WMV: An algorithm based on $\theta$-MIP weight assignment, such as in Algorithm 2, achieves a regret bound of $R_T \in \mathsf{O}\left(2^N \sqrt{NT \log\left(NT\right)}\right)$.*

*Proof.* Let $\hat{p}_i^t$ be the empirical estimate of $p_i$ after collecting $t$ i.i.d. samples, then due to Hoeffding's inequality, we have,

$$\mathbb{P}\left(\left|\hat{p}_i^t - p_i\right| \geq \epsilon\right) \leq 2\exp\left(-\frac{t\epsilon^2}{2\sigma_i^2}\right) \leq 2\exp\left(-\frac{t\epsilon^2}{2\sigma^2}\right).$$

where $\sigma = \max_i \sigma_i$. Recall that at each round, we collect samples from all experts. Thus, after $t$ rounds, we have collected precisely $t$ samples from each expert. Using Union bound, we have

$$\mathbb{P}\left(\left|\hat{p}_i^t - p_i\right| \leq \epsilon, \forall i \in \{1, ..., N\}\right) \geq 1 - 2N\exp\left(-\frac{t\epsilon^2}{2\sigma^2}\right). \qquad \text{(A.35)}$$

Using the inclusion-exclusion principle, and the definition of $\mathcal{S}(\theta) = \{S_1, ..., S_k\}$, we have

$$\mathbb{P}_{\texttt{maj}}(\mathcal{E}, \theta) = \sum_{i_1=1}^{k} \prod_{j \in S_{i_1}} p_j - \sum_{i_1 \neq i_2 = 1}^{k} \prod_{j \in S_{i_1} \cup S_{i_2}} p_j + ...$$

When $|p_i - \hat{p}_i^t| \leq \epsilon$ for all experts, i.e., $i \in [N]$, for any given $\theta$, we obtain,

$$\left|\mathbb{P}_{\texttt{maj}}(\mathcal{E}, \theta) - \widehat{\mathbb{P}_{\texttt{maj}}}^t(\mathcal{E}, \theta)\right| =$$

$$\left|\left(\sum_{i_1=1}^{k} \prod_{j \in S_{i_1}} p_j - \sum_{i_1 \neq i_2 = 1}^{k} \prod_{j \in S_{i_1} \cup S_{i_2}} p_j + ...\right) - \left(\sum_{i_1=1}^{k} \prod_{j \in S_{i_1}} \hat{p}_j^t - \sum_{i_1 \neq i_2 = 1}^{k} \prod_{j \in S_{i_1} \cup S_{i_2}} \hat{p}_j^t + ...\right)\right|$$

$$\leq \left(\sum_{i=1}^{k} \sum_{j \in S_i} p_j\right)\epsilon \leq \binom{N-1}{\lceil \frac{N-1}{2} \rceil}\left(\sum_{j \in S_1 \cup ... \cup S_k} p_j\right)\epsilon,$$

where $\lceil a \rceil$ denotes the ceiling of real number $a$. As shown in Remark 5 for a simple scenario, the first inequality is tight. The last inequality is due to Lemma A.3 which implies that the maximum size of $\mathcal{S}(\theta)$ is $\binom{N}{\lceil \frac{N+1}{2} \rceil}$ and consequently, we have at most $\binom{N-1}{\lceil \frac{N-1}{2} \rceil}$ repetitions of each $p_j$ in the above double summation. Knowing that $\sum_{j \in S_1 \cup ... \cup S_k} p_j \leq N$, leads to the bound $\binom{N-1}{\lceil \frac{N-1}{2} \rceil} N\epsilon$.

Applying the Stirling's approximation, we can achieve the following,

$$\left| \mathbb{P}_{\mathtt{maj}}(\mathcal{E}, \theta) - \widehat{\mathbb{P}_{\mathtt{maj}}}^t(\mathcal{E}, \theta) \right| \leq \frac{2^N}{\sqrt{N}} N \epsilon = 2^N \sqrt{N} \epsilon.$$

Applying the concentration result in Eq. (A.35), we obtain,

$$\mathbb{P} \left( \left| \mathbb{P}_{\mathtt{maj}}(\mathcal{E}, \theta) - \widehat{\mathbb{P}_{\mathtt{maj}}}^t(\mathcal{E}, \theta) \right| \leq 2^N \sqrt{N} \epsilon \right) \geq \mathbb{P} \left( \left| \hat{p}_i^t - p_i \right| \leq \epsilon, \forall i \in [N] \right) \geq 1 - 2N \exp \left( -\frac{t \epsilon^2}{2 \sigma^2} \right).$$

Now, requiring the probability to be at least $1 - \delta$, we obtain,

$$\mathbb{P} \left( \left| \mathbb{P}_{\mathtt{maj}}(\mathcal{E}, \theta) - \widehat{\mathbb{P}_{\mathtt{maj}}}^t(\mathcal{E}, \theta) \right| \leq \epsilon_0 \right) \geq 1 - \delta. \tag{A.36}$$

where,

$$\epsilon_0(t) := \sqrt{\frac{2N 4^N \sigma^2}{t} \log \left( \frac{2N}{\delta} \right)}.$$

Let us consider two possible cases at round $t$; (i) $\mathbb{P}_{\mathtt{maj}}(\mathcal{E}, \theta^*) - \mathbb{P}_{\mathtt{maj}}(\mathcal{E}, \theta^t) > 2\epsilon_0(t)$ and (ii) $\mathbb{P}_{\mathtt{maj}}(\mathcal{E}, \theta^*) - \mathbb{P}_{\mathtt{maj}}(\mathcal{E}, \theta^t) \leq 2\epsilon_0(t)$. Next, we show that case $i$ won't happen with high probability. To this end, using the concentration in (A.36), we imply that the following events hold with probability at least $1 - 2\delta$.

$$\left| \mathbb{P}_{\mathtt{maj}}(\mathcal{E}, \theta^*) - \widehat{\mathbb{P}_{\mathtt{maj}}}^t(\mathcal{E}, \theta^*) \right| \leq \epsilon_0(t),$$

$$\left| \mathbb{P}_{\mathtt{maj}}(\mathcal{E}, \theta^t) - \widehat{\mathbb{P}_{\mathtt{maj}}}^t(\mathcal{E}, \theta^t) \right| \leq \epsilon_0(t).$$

Combining the above inequalities with the assumption of case (i), i.e., $1 \geq \mathbb{P}_{\mathtt{maj}}(\mathcal{E}, \theta^*) - \mathbb{P}_{\mathtt{maj}}(\mathcal{E}, \theta^t) > 2\epsilon_0(t)$, implies $\widehat{\mathbb{P}_{\mathtt{maj}}}^t(\mathcal{E}, \theta^t) < \widehat{\mathbb{P}_{\mathtt{maj}}}^t(\mathcal{E}, \theta^*)$. This is a contradiction with $\theta^t \in \arg \max_\theta \widehat{\mathbb{P}_{\mathtt{maj}}}^t(\mathcal{E}, \theta)$. Thus, with probability $1 - 2\delta$, case (ii) occurs. By letting $\delta = 1/(2T)$, we could bound the regret as follows,

$$R_T = \sum_{t=1}^{T} \left( \mathbb{P}_{\mathtt{maj}}(\mathcal{E}, \theta^*) - \mathbb{P}_{\mathtt{maj}}(\mathcal{E}, \theta^t) \right) \leq \sum_{t=1}^{t_0} 1 + \sum_{t=t_0+1}^{T} \left( \mathbb{P}_{\mathtt{maj}}(\mathcal{E}, \theta^*) - \mathbb{P}_{\mathtt{maj}}(\mathcal{E}, \theta^t) \right)$$

$$\leq t_0 + \frac{1}{T}(T - t_0) + (1 - \frac{1}{T}) \sum_{t=t_0+1}^{T} 2\epsilon_0(t)$$

$$\leq t_0 + 1 + (1 - \frac{1}{T}) \sqrt{8N 4^N \sigma^2 \log(4TN)} \int_{t_0}^{T} \frac{1}{\sqrt{t}} dt \in \mathcal{O} \left( 2^N \sqrt{NT \log(NT)} \right).$$

where $t_0 := 2N 4^N \sigma^2 \log(4TN)$. Note that during the initial phase, when $t \leq t_0$, we bound the instantaneous regrets by their worst scenario that is 1 as the utility function is a probability. □

**Remark 5.** *Let consider $N = 3$ experts with competencies $p_1, p_2, p_3$ and weights $\theta = (1, 1, 1)$. In this case, $\mathcal{S}(\theta) = \{\{1, 2\}, \{1, 3\}, \{2, 3\}\}$. Moreover, let $\hat{p}_i = p_i + \epsilon$ for all $i$, where $\epsilon \leq \min_i p_i$ then*

$$|\mathbb{P}_{maj}(\mathcal{E}, \theta) - \widehat{\mathbb{P}_{maj}}(\mathcal{E}, \theta)| = |p_1 p_2 - \hat{p}_1 \hat{p}_2 + p_1 p_3 - \hat{p}_1 \hat{p}_3 + p_2 p_3 - \hat{p}_2 \hat{p}_3 - 2(p_1 p_2 p_3 - \hat{p}_1 \hat{p}_2 \hat{p}_3)|$$

$$= |(p_1 + p_2)\epsilon + (p_1 + p_3)\epsilon + (p_2 + p_3)\epsilon + 3\epsilon^2 - 2(p_1 p_2 + p_1 p_3 + p_2 p_3)\epsilon - 2(p_1 + p_2 + p_3)\epsilon^2 - 2\epsilon^3|$$

$$= |2(p_1 + p_2 + p_3 - (p_1 p_2 + p_1 p_3 + p_2 p_3))\epsilon - (2(p_1 + p_2 + p_3) - 3)\epsilon^2 - 2\epsilon^3|.$$

*The introduced bound is $\binom{N-1}{\lceil \frac{N-1}{2} \rceil]}\left(\sum_{j \in S_1 \cup S_2 \cup S_3} p_j\right)\epsilon = 2(p_1 + p_2 + p_3)\epsilon$. Now, lets compare the exact value and the bound for when $p = (.02, .03, .04)$. In this case,*

$$\binom{N-1}{\lceil \frac{N-1}{2} \rceil]}\left(\sum_{j \in S_1 \cup S_2 \cup S_3} p_j\right)\epsilon - |\mathbb{P}_{maj}(\mathcal{E}, \theta) - \widehat{\mathbb{P}_{maj}}(\mathcal{E}, \theta)| = 0.18\epsilon - |0.1748\epsilon + 2.82\epsilon^2 - 2\epsilon^3|$$

*Note that the above expression can be arbitrary small as $p_i$s tends to zero.*

### A.17   Scaling Variance of $1/n_i$ for Bernoulli Variables

**Lemma A.4.** *For estimates sums of Bernoulli variables $X_i \in \{0, 1\}$, where $p_i$ is the probability of success for variable $X_i$, $\hat{p}_i = \frac{k_i}{n_i}$ is the empirical estimate, and $n_i$ is the empirical count of successes, it holds that,*

$$\mathbb{P}\left(\left|\sum_{i=1}^{N} \hat{p}_i - \sum_{i=1}^{N} p_i\right| \geq \epsilon\right) \leq 2\exp\left(-\frac{2\epsilon^2}{\sum_{i=1}^{N} \frac{1}{n_i}}\right).$$

*Proof.* **Scaling $1/n_i$:** Since each $X_{i,j} \in [0, 1]$, the range of each scaled variable $\frac{X_{i,j}}{n_i}$ is contained in $\{0, 1/n_i\}$. Each original Bernoulli random variable $X_{i,j}$ takes values in $\{0, 1\}$, thus,

$$X_{i,j} \in [0, 1].$$

When forming the empirical mean,

$$\hat{p}_i = \frac{1}{n_i}\sum_{j=1}^{n_i} X_{i,j},$$

each summand inside the sum is scaled by $1/n_i$. Therefore,

$$\frac{X_{i,j}}{n_i} \in \left\{0, \frac{1}{n_i}\right\} \subseteq [0, 1/n_i].$$

This implies the range of each scaled variable is reduced from length 1 to length $1/n_i$. This scaling is allows us to applying Hoeffding's inequality, as the concentration bound depends on the sum of the squared lengths of these ranges,

$$\sum_{i=1}^{N}\sum_{j=1}^{n_i}\left(\frac{1}{n_i} - 0\right)^2 = \sum_{i=1}^{N} n_i \cdot \frac{1}{n_i^2} = \sum_{i=1}^{N} \frac{1}{n_i}.$$

Hence, the variance proxy in the concentration bound is governed by $\sum_i \frac{1}{n_i}$.

Applying Hoeffding's inequality for sums of independent bounded variables, for any $\epsilon > 0$:

$$\mathbb{P}\left(|S_\Sigma - \mu| \geq \epsilon\right) \leq 2\exp\left(-\frac{2\epsilon^2}{\sum_{i=1}^{N} n_i \left(\frac{1}{n_i} - 0\right)^2}\right).$$

Simplifying the denominator,

$$\sum_{i=1}^{N} n_i \left(\frac{1}{n_i}\right)^2 = \sum_{i=1}^{N} \frac{1}{n_i}.$$

Hence,

$$\mathbb{P}\left(\left|\sum_{i=1}^{N} \hat{p}_i - \sum_{i=1}^{N} p_i\right| \geq \epsilon\right) \leq 2\exp\left(-\frac{2\epsilon^2}{\sum_{i=1}^{N} \frac{1}{n_i}}\right).$$

$\square$

# B  Algorithms

## B.1  Greedy Algorithm for Constructing Optimal Egalitarian Committee

---
**Algorithm 3** Greedy Algorithm - Construct Optimal Committee with Accuracy $\mathbb{P}_{\texttt{maj}}$
---
**Require:** Set of experts $\mathcal{E}$ and $\mathcal{E}_{\texttt{cand}} = \emptyset$
**Ensure:** Optimal $\mathbb{P}_{\texttt{maj}}$ and $\mathcal{E}_{\texttt{OEC}}$
 1: Sort all experts by their accuracy measure $p_i$, resulting in $\texttt{sorted}(\{p_i\})$.
 2: $\mathcal{E}_{\texttt{cand}} \leftarrow e_1$.
 3: **for** $j = 2$ to $N$ **do**
 4:     **if** $\mathfrak{F}(\mathcal{E}_{\texttt{cand}}, \mathcal{E}') \geq 0, \quad \forall \mathcal{E}' \in \texttt{TopK}(\mathcal{E}_{\texttt{cand}}^{\leq j})$ **then**          ▷ Check the advantage function.
 5:         $\mathcal{E}_{\texttt{cand}} \leftarrow \mathcal{E} \cup \mathcal{E}'$.
 6:         $\mathbb{P}_{\texttt{maj}} \leftarrow \mathbb{P}_{\texttt{maj}}(\mathcal{E}_{\texttt{cand}})$
 7:     **end if**
 8: **end for**
 9: $\mathcal{E}_{\texttt{OEC}} \leftarrow \mathcal{E}_{\texttt{cand}}$
10: **return** $\mathbb{P}_{\texttt{maj}}, \mathcal{E}_{\texttt{OEC}}$

---

## B.2  Zooming Algorithm

---
**Algorithm 4** Zooming Algorithm for Lipschitz Bandits
---
 1: **Input:** Metric space $(X, d)$, Lipschitz constant $L > 0$, time horizon $T$
 2: **Initialization:** Set active arms Active $= \emptyset$
 3: **for** $t = 1, 2, \ldots, T$ **do**
 4:     **Update Confidence Intervals:**
 5:     **for** $x \in$ Active **do**
 6:         Compute empirical mean reward:

$$\hat{\mu}_t(x) = \frac{1}{n_t(x)} \sum_{s=1}^{n_t(x)} r_s(x)$$

 7:         Compute confidence radius:

$$r_t(x) = \sqrt{\frac{2 \log T}{n_t(x)}}$$

 8:     **end for**
 9:     **Activate New Arms:**
10:     **for** $x \in X \setminus$ Active **do**
11:         **if** $\min_{y \in \text{Active}} d(x, y) > r_t(y)$ **then**
12:             Add $x$ to Active
13:         **end if**
14:     **end for**
15:     **Select Arm to Play:**
$$x_t = \arg \max_{x \in \text{Active}} (\hat{\mu}_t(x) + r_t(x))$$
16:     **Play Arm $x_t$:**
17:     Observe reward $r_t$ and update $n_t(x_t)$
18: **end for**

---

The zooming algorithm from  Kleinberg, Slivkins, and Upfal [39]  adaptively explores and exploits strategies in a metric space by maintaining confidence intervals around the estimated rewards of active arms, serving as an appropriate benchmark bandit algorithm for when the reward function lacks convexity. It dynamically activates new arms that are sufficiently far from existing ones (based on their confidence radii) and prioritizes arms with high upper confidence bounds. In non-convex spaces, the algorithm leverages the Lipschitz condition to generalize observations across nearby arms, ensuring efficient exploration while adapting to the structure of the problem instance.

# C Experiment Details

## C.1 LLM Offline Performance Benchmarking (i.e. Expert Competencies)

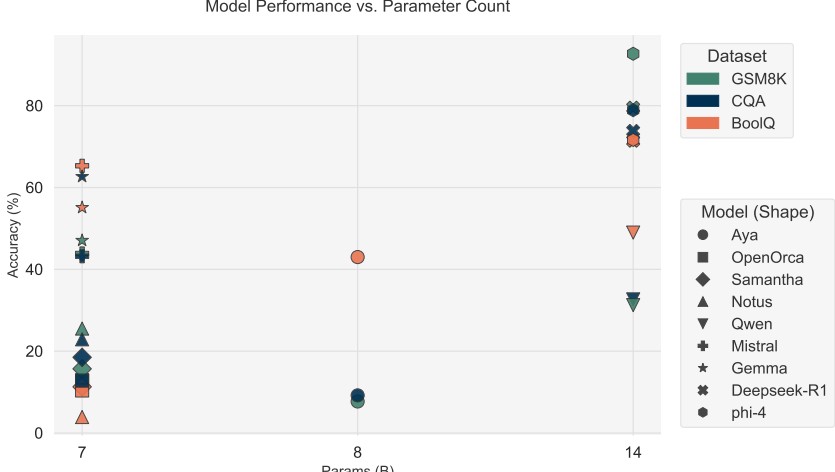

Figure 4: Visualization of the performance comparison of language models across three benchmarks (GSM8K, CQA, BoolQ) as a function of parameter count (7B, 8B, 14B).

| Model | GSM8K Acc. (%) | CQA Acc. (%) | BoolQ Acc. (%) | Params. (B) |
|---|---|---|---|---|
| Aya | 7.71 | 9.20 | 43.00 | 8 |
| OpenOrca | 12.90 | 12.80 | 10.40 | 7 |
| Samantha | 15.69 | 18.50 | 11.30 | 7 |
| Notus | 25.53 | 22.90 | 3.91 | 7 |
| Qwen | 31.25 | 32.66 | 48.98 | 14 |
| Mistral | 43.88 | 43.20 | 65.31 | 7 |
| Gemma | 47.07 | 62.68 | 55.10 | 7 |
| Deepseek-R1 | 79.52 | 73.83 | 71.43 | 14 |
| phi-4 | 92.69 | 78.90 | 71.63 | 14 |

Table 2: **Model performance ranking.** Models listed by their accuracy across various domains in the offline setting, alongside their respective parameter count (billions).

## C.2 Description of Datasets for Task Generation in Online Simulation

| Dataset | Description | Testing Capabilities and Limitations |
|---|---|---|
| GSM8K | Collection of 8.5k middle school-level math problems from Cobbe et al. [36] solvable by elementary operations and logical reasoning. | The collections of questions are relatively basic and relevant to mathematics. The expert is allowed to propose any answer they would like (in this case limited to integers). |
| CommonsenseQA | Open-source from benchmark dataset from Talmor et al. [37] containing 9.26k questions with common sense yes/no questions. | This dataset benchmarks the reasoning capabilities of LLMs, and provides a catalogue of candidate answers to choose from. |
| BoolQ | A question-answering dataset from Clark et al. [38] that focuses on logical reasoning and evaluation of boolean expressions. It is part of the BIG-Bench Hard (BBH) suite. | This dataset examines the logical reasoning capabilities of LLMs and provides a catalogue of possible answers *true* and *false*. |

Table 3: Overview of selected evaluation datasets.

## C.3    Computing Specifications

| Component | Specification |
| --- | --- |
| CPU | Intel$^{®}$ Core™i9-9900K @ 3.60 GHz |
| Cores/Threads | 16 |
| GPU | NVIDIA GeForce RTX 2080 |
| VRAM | 8 GB GDDR6 |
| CUDA Version | 11.8 |
| System Memory | 64 GB DDR4 |
| Storage | NVMe SSD |
| Python Version | 3.8.10 |
| Gurobi Optimizer | 9.5.2 |

Table 4: Computing hardware and software library specifications.

## C.4 Chain-of-Thought Prompting with Constrained Output Formatting

**Structured Chain-of-Thought Prompting:** Our approach implements a constrained variant of chain-of-thought (CoT) prompting [40] with strict output formatting rules. We provide a prompt template for each of the domains.

### C.4.1 GSM8K Prompt Template

```
### Instructions:
1. Read the question carefully and identify what is being asked.
2. Solve the problem methodically, showing each step clearly.
3. Double-check your calculations before finalizing the answer.
4. Your final output MUST follow EXACTLY this format:

### Reasoning:
[Your step-by-step reasoning here]

### Final Answer: [Numerical Value]

### Required Output Format Rules:
- Only numbers allowed in final answer (e.g., 42, 3.14, 2/3)
- If uncertain, return ### Final Answer: 0
- No additional text after final answer
- Final answer must be the last line

### Question:
{question}

### Choices:
{choices}

### Reasoning:
```

### C.4.2 CommonsenseQA Prompt Template

```
### Instructions:
1. Read the question carefully and identify what is being asked.
2. Solve the problem methodically, showing each step clearly.
3. Double-check your calculations before finalizing the answer.
4. Your final output MUST follow EXACTLY this format:

### Reasoning:
[Your step-by-step reasoning here]

### Final Answer: One of 5 catagories [A, B, C, D, E]

### Required Output Format Rules:
- Only numbers are allowed in the final answer (e.g., A, B, C, D, E)
- If you cannot determine the answer, you MUST pick a random answer from [A, B, C, D, E].
- No additional text, explanations, or characters after the final answer.
- The final answer line must be the very last line of your response.

### Question:
{question}

### Choices:
{choices}

### Reasoning:
```

### C.4.3 BoolQ Prompt Template

```
### Instructions:
1. Read the question carefully and identify what is being asked.
2. Solve the problem methodically, showing each step clearly.
3. Double-check your calculations before finalizing the answer.
4. Your final output MUST follow EXACTLY this format:

### Reasoning:
[Your step-by-step reasoning here]

### Final Answer: One of 2 catagories [true, false]

### Required Output Format Rules:
- Only numbers are allowed in the final answer (e.g., true, false)
- If you cannot determine the answer, you MUST pick a random answer from [true, false].
- No additional text, explanations, or characters after the final answer.
- The final answer line must be the very last line of your response.

### Question:
{question}

### Reasoning:
```

### C.4.4 Output Parsing Algorithm

The response parsing algorithm enforces strict numerical output constraints through regular expression matching:

---

**Algorithm 5** Response Parser

---

1: **procedure** PARSERESPONSE($R$)
2:     *pattern* ← "\\\\Final Answer:\s*(-?\d+\.?\d*—[-+]?\d+/\d+)"
3:     *match* ← regex_search($R$, *pattern*, IGNORECASE)
4:     **if** *match.success* **then**
5:         **return** *match.group*$(1)$
6:     **else**
7:         **return** "0"
8:     **end if**
9: **end procedure**

---

## C.5 Extended Empirical Results

| Algorithm | $N$ | $|\mathcal{E}^*|$ | $\mathbb{P}^*_{\text{maj}}$ | $\tilde{\epsilon}$ | $R_T$ | $\%R\downarrow$ | $\hat{P}_{\text{maj}}$ | Domain | Config. Id. |
|---|---|---|---|---|---|---|---|---|---|
| SEE | 20 | 9 | 0.992 | 0.010 | **546.2** | 0.921 | **0.980** | Bernoulli | SE3 |
| Comb. UCB | 20 | 9 | 0.992 | 0.010 | 6932 | - | 0.370 | Bernoulli | SC3 |
| SEE | 15 | 3 | 0.985 | 0.040 | **649.8** | 0.867 | **0.969** | Bernoulli | SE3 |
| Comb. UCB | 15 | 3 | 0.985 | 0.040 | 4906 | - | 0.549 | Bernoulli | SC2 |
| SEE | 5 | 3 | 0.883 | 0.010 | **535.3** | 0.719 | **0.871** | Bernoulli | SE1 |
| Comb. UCB | 5 | 3 | 0.883 | 0.010 | 1905 | - | 0.759 | Bernoulli | SC1 |
| $\theta$-WMV | 9 | 9 | 0.644 | 0.004 | **217.4** | 0.545 | **0.606** | Bernoulli | WV3 |
| Zooming | 9 | 9 | 0.644 | 0.004 | 478.4 | - | 0.449 | Bernoulli | WZ3 |
| $\theta$-WMV | 6 | 6 | 0.911 | 0.004 | **312.4** | 0.331 | **0.853** | Bernoulli | WV2 |
| Zooming | 6 | 6 | 0.911 | 0.004 | 467.0 | - | 0.765 | Bernoulli | WZ2 |
| $\theta$-WMV | 3 | 3 | 0.881 | 0.106 | **287.0** | 0.474 | **0.835** | Bernoulli | WV1 |
| Zooming | 3 | 3 | 0.881 | 0.106 | 546 | - | 0.681 | Bernoulli | WZ1 |
| $\theta$-WMV | 9 | 9 | 0.897 | 0.023 | **200.1** | 0.080 | **0.828** | Bernoulli | WS4 |
| SEE | 9 | 9 | 0.897 | 0.023 | 216.0 | - | 0.806 | Bernoulli | WE5 |
| $\theta$-WMV | 5 | 5 | 0.811 | 0.05 | **203.8** | 0.114 | **0.732** | Bernoulli | WS5 |
| SEE | 5 | 5 | 0.811 | 0.05 | 230.1 | - | 0.707 | Bernoulli | WE5 |
| $\theta$-WMV | 3 | 3 | 0.89 | 0.02 | **202.6** | 0.484 | **0.817** | Bernoulli | WS6 |
| SEE | 3 | 3 | 0.68 | 0.05 | 392.7 | - | 0.427 | Bernoulli | WE6 |
| $\theta$-WMV | 9 | 9 | 0.927 | 0.027 | **587.0** | 0.829 | **0.927** | GSM8K | WG3 |
| Zooming | 9 | 9 | 0.927 | 0.027 | 3443 | - | 0.604 | GSM8K | ZG3 |
| $\theta$-WMV | 6 | 6 | 0.927 | 0.132 | **420.6** | 0.353 | **0.918** | GSM8K | WG2 |
| Zooming | 6 | 6 | 0.927 | 0.132 | 650.2 | - | 0.891 | GSM8K | ZG2 |
| $\theta$-WMV | 3 | 3 | 0.439 | 0.057 | **302.0** | 0.660 | **0.433** | GSM8K | WG1 |
| Zooming | 3 | 3 | 0.439 | 0.057 | 886.0 | - | 0.369 | GSM8K | ZG1 |
| SEE | 9 | 4 | 0.807 | 0.036 | **328.2** | 0.939 | **0.805** | CommonsenseQA | CS3 |
| Comb. UCB | 9 | 4 | 0.807 | 0.036 | 5437 | - | 0.293 | CommonsenseQA | CC3 |
| SEE | 5 | 3 | 0.810 | 0.031 | **676.8** | 0.364 | **0.810** | CommonsenseQA | CS2 |
| Comb. UCB | 5 | 3 | 0.810 | 0.031 | 1065 | - | 0.664 | CommonsenseQA | CC2 |
| SEE | 3 | 1 | 0.810 | 0.051 | **339.0** | 0.65 | **0.807** | CommonsenseQA | CS1 |
| Comb. UCB | 3 | 1 | 0.810 | 0.051 | 970 | - | 0.763 | CommonsenseQA | CS1 |
| $\theta$-WMV | 9 | 9 | 0.763 | 0.0085 | **210.1** | 0.402 | **0.675** | BoolQ | WB3 |
| Zooming | 9 | 9 | 0.763 | 0.0085 | 351.4 | - | 0.483 | BoolQ | ZB3 |
| $\theta$-WMV | 5 | 5 | 0.735 | 0.013 | **360.9** | 0.512 | **0.735** | BoolQ | WB2 |
| Zooming | 5 | 5 | 0.735 | 0.013 | 740.3 | - | 0.694 | BoolQ | ZB2 |
| $\theta$-WMV | 4 | 4 | 0.735 | 0.04 | **183.1** | 0.225 | **0.667** | BoolQ | WB1 |
| Zooming | 4 | 4 | 0.735 | 0.04 | 236.4 | - | 0.586 | BoolQ | ZB1 |

Table 5: Multi-armed bandit experiment results showing consistent performance across environments, with time horizon $T \leq 10^4$. Performance metrics show reduced cumulative regret ($R_T$) and increased empirical accuracy ($\hat{P}_{\text{maj}}$) across all baselines.

### C.5.1 Empirical Results: Successive Expert Elimination with Bernoulli Experts

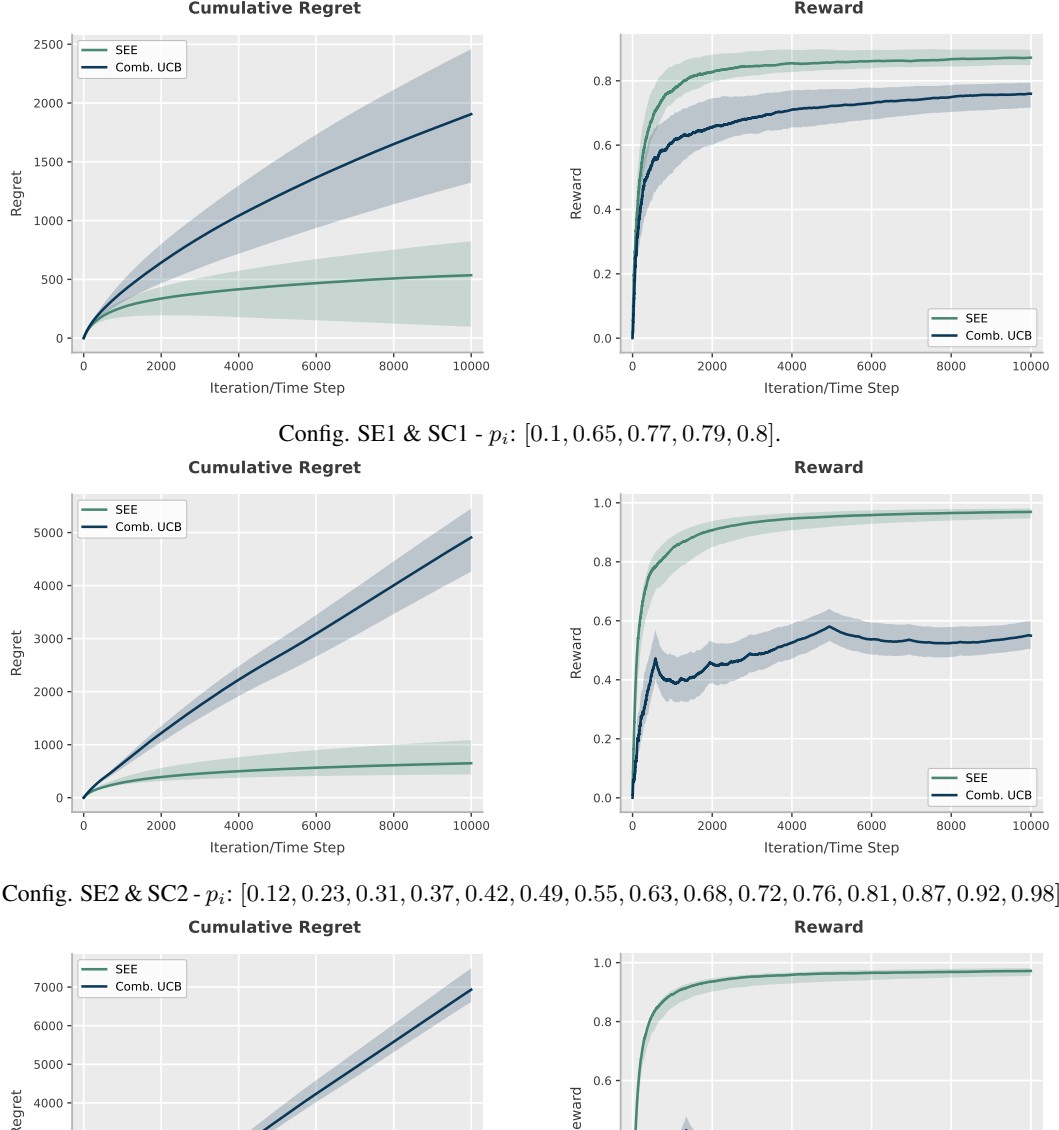

Config. SE1 & SC1 - $p_i$: $[0.1, 0.65, 0.77, 0.79, 0.8]$.

Config. SE2 & SC2 - $p_i$: $[0.12, 0.23, 0.31, 0.37, 0.42, 0.49, 0.55, 0.63, 0.68, 0.72, 0.76, 0.81, 0.87, 0.92, 0.98]$.

Config. SE3 & SC3 - $p_i$: $[0.04, 0.07, 0.09, 0.12, 0.15, 0.19, 0.22, 0.26, 0.30, 0.33, 0.70, 0.73, 0.76, 0.78, 0.81,$ $0.84, 0.86, 0.89, 0.90, 0.91]$.

Figure 5: Mean values are calculated over 1,000 trials, with finite time horizon $T = 10,000$, with shaded regions representing confidence intervals of $\pm$UCB.

## C.5.2 Empirical Results: $\theta$-Weighted Majority Voting with Bernoulli Experts

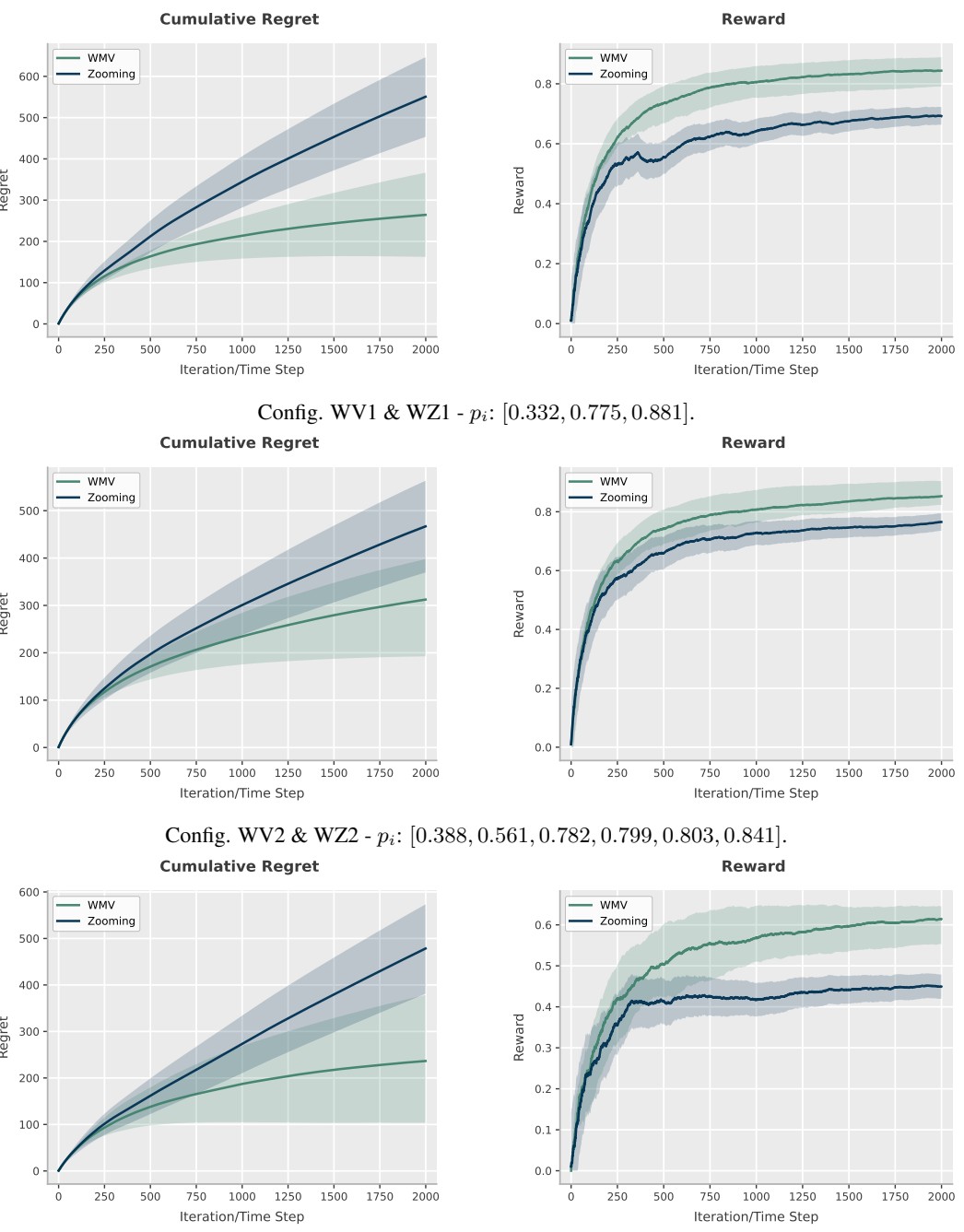

Config. WV1 & WZ1 - $p_i$: $[0.332, 0.775, 0.881]$.

Config. WV2 & WZ2 - $p_i$: $[0.388, 0.561, 0.782, 0.799, 0.803, 0.841]$.

Config. WV3 & WZ3 - $p_i$: $[0.261, 0.370, 0.382, 0.499, 0.503, 0.511, 0.542, 0.616, 0.634]$.

Figure 6: Mean values are calculated over 1,000 trials, with finite time horizon $T = 2000$, with shaded regions representing confidence intervals of $\pm$UCB.

## C.5.3 Empirical Results: $\theta$-WMW vs. SEE Voting with Bernoulli Experts

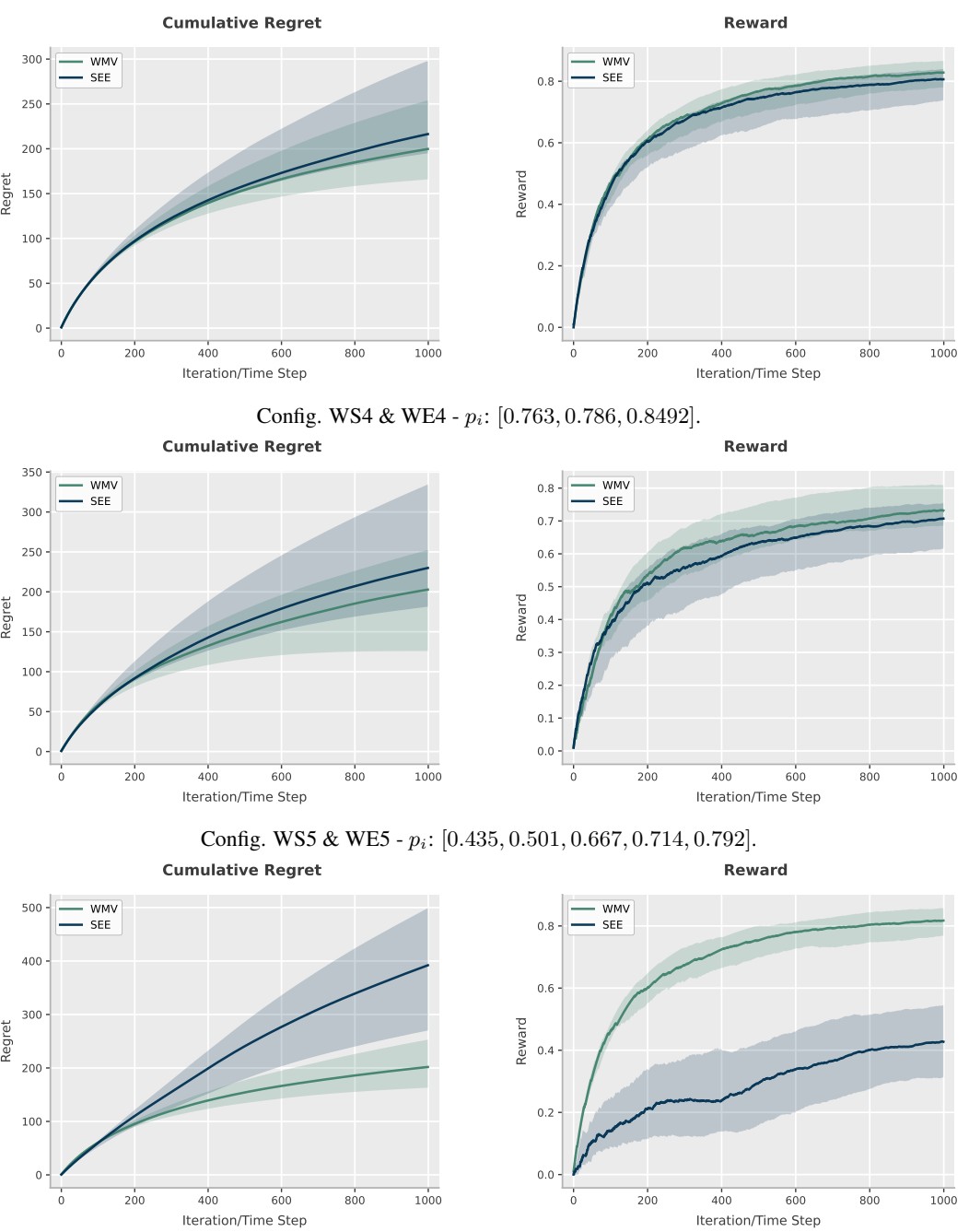

Config. WS4 & WE4 - $p_i$: $[0.763, 0.786, 0.8492]$.

Config. WS5 & WE5 - $p_i$: $[0.435, 0.501, 0.667, 0.714, 0.792]$.

Config. WS6 & WE6 - $p_i$: $[0.121, 0.232, 0.319, 0.374, 0.428, 0.498, 0.552, 0.637, 0.681]$.

Figure 7: Mean values are calculated over 1,000 trials, with finite time horizon $T = 1000$, with shaded regions representing confidence intervals of $\pm$UCB. To note, for a small batch of experts, the regret performances are near-identical as, the optimal committee consists of a single best expert among a few. The learning complexity is not high, and both algorithms converge to the optimal solution quickly. As the number of experts increases, $\theta$-WMV demonstrates a clear advantage in regret minimization, aligning with our theory from Lemma 2.3, that the weighted majority voting committee will always yield a superior solution compared to the egalitarian committee.

## C.5.4 Empirical Results: θ-Weighted Majority Voting with GSM8K Tasks

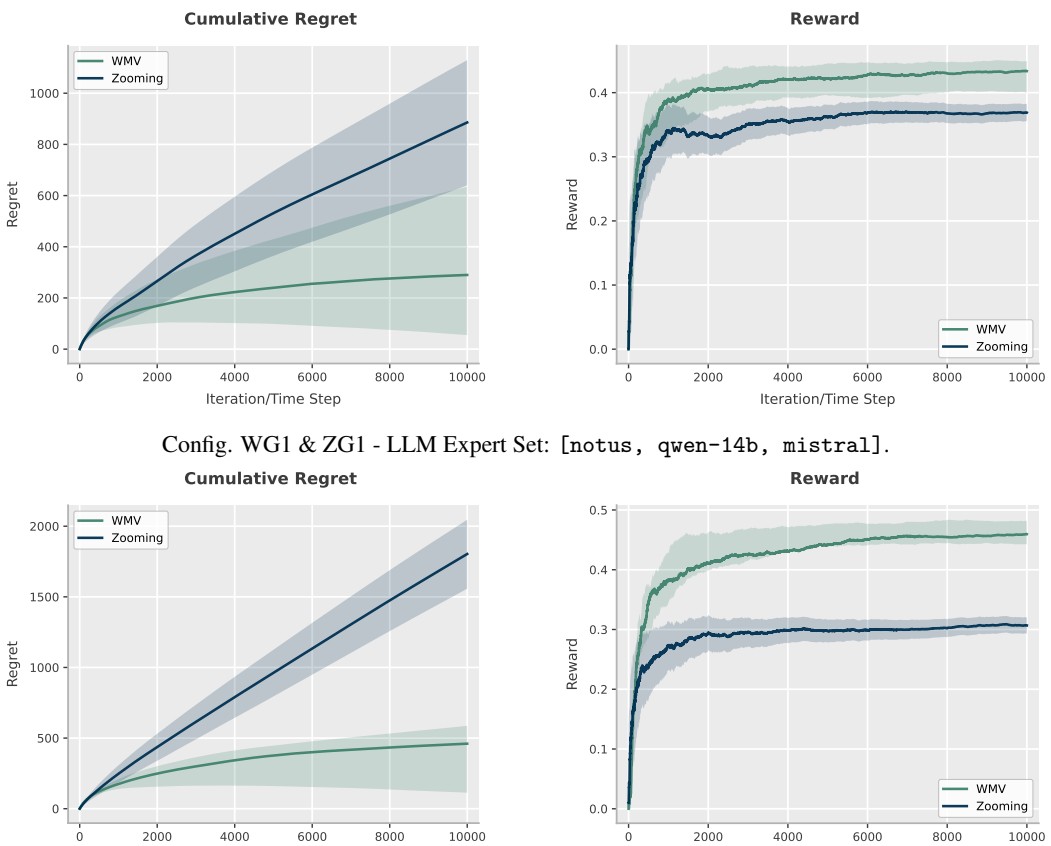

Config. WG1 & ZG1 - LLM Expert Set: [notus, qwen-14b, mistral].

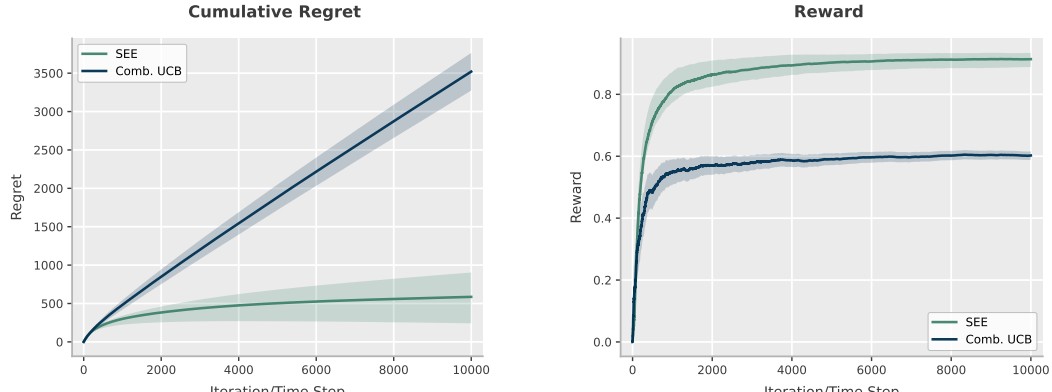

Config. WG2 & ZG2 - LLM Expert Set: [samantha-mistral, notus, qwen-14b, mistral, gemma-7b, deepseek-r1-14b].

Config. WG3 & ZG3 - LLM Expert Set: [aya, mistral-openorca, samantha-mistral, notus, qwen-14b, mistral, gemma-7b, deepseek-r1-14b, phi4].

Figure 8: Questions were sampled from the GSM8K dataset [36] . Mean values are calculated over 1,000 trials, with finite time horizon $T = 10,000$, with shaded regions representing confidence intervals of $\pm$UCB.

### C.5.5 Empirical Results: Successive Expert Elimination Voting with CommonsenseQA Tasks

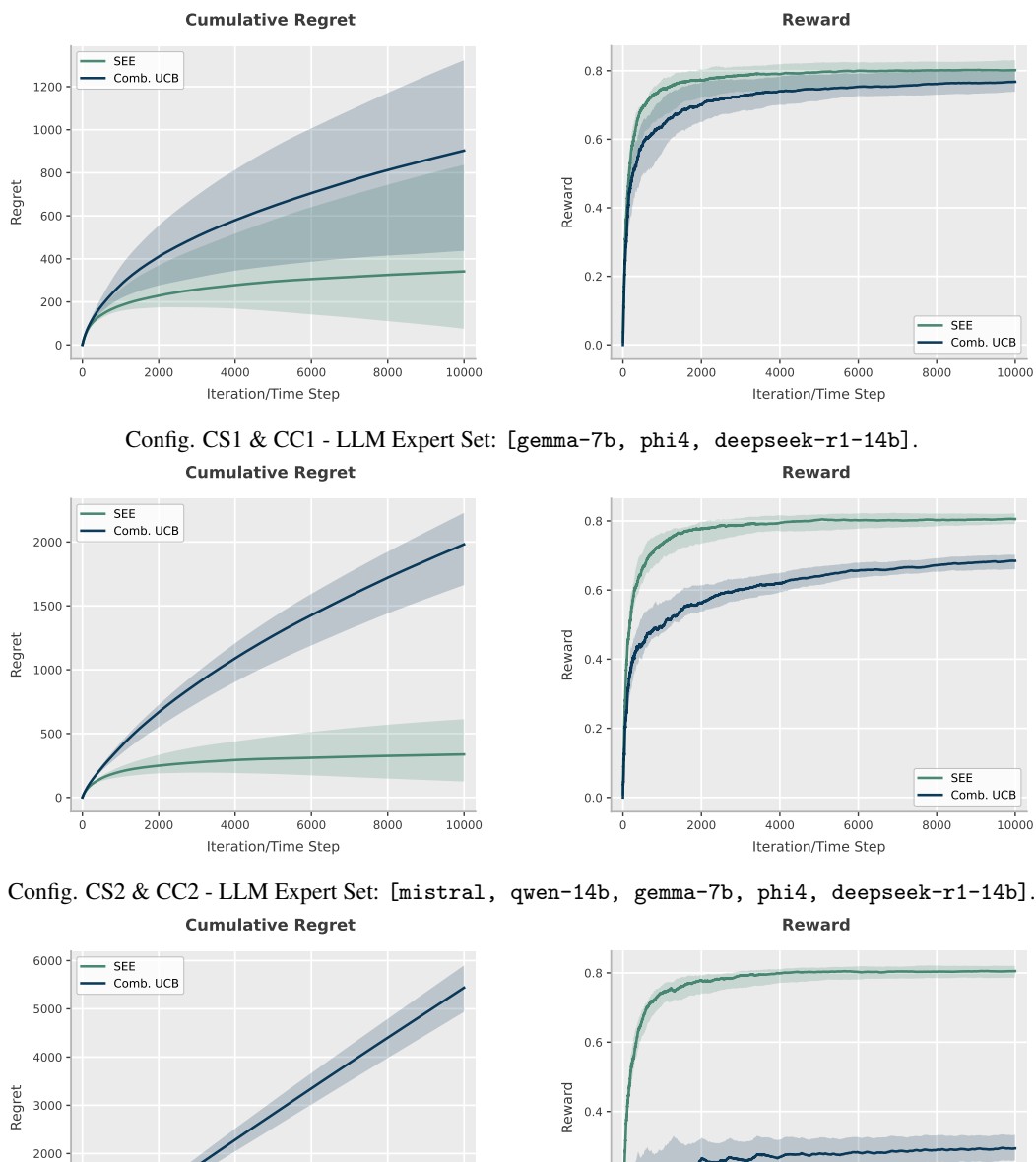

Config. CS1 & CC1 - LLM Expert Set: [gemma-7b, phi4, deepseek-r1-14b].

Config. CS2 & CC2 - LLM Expert Set: [mistral, qwen-14b, gemma-7b, phi4, deepseek-r1-14b].

Config. CS3 & CC3 - LLM Expert Set: [aya, mistral-openorca, samantha-mistral, notus, mistral, qwen-14b, gemma-7b, phi4, deepseek-r1-14b].

Figure 9: Questions were sampled from the CommonsenseQA dataset [37]. Mean values are calculated over 1,000 trials, with finite time horizon $T = 10,000$, with shaded regions representing confidence intervals of $\pm$UCB

## C.5.6 Empirical Results: Successive Expert Elimination Voting with BoolQ Tasks

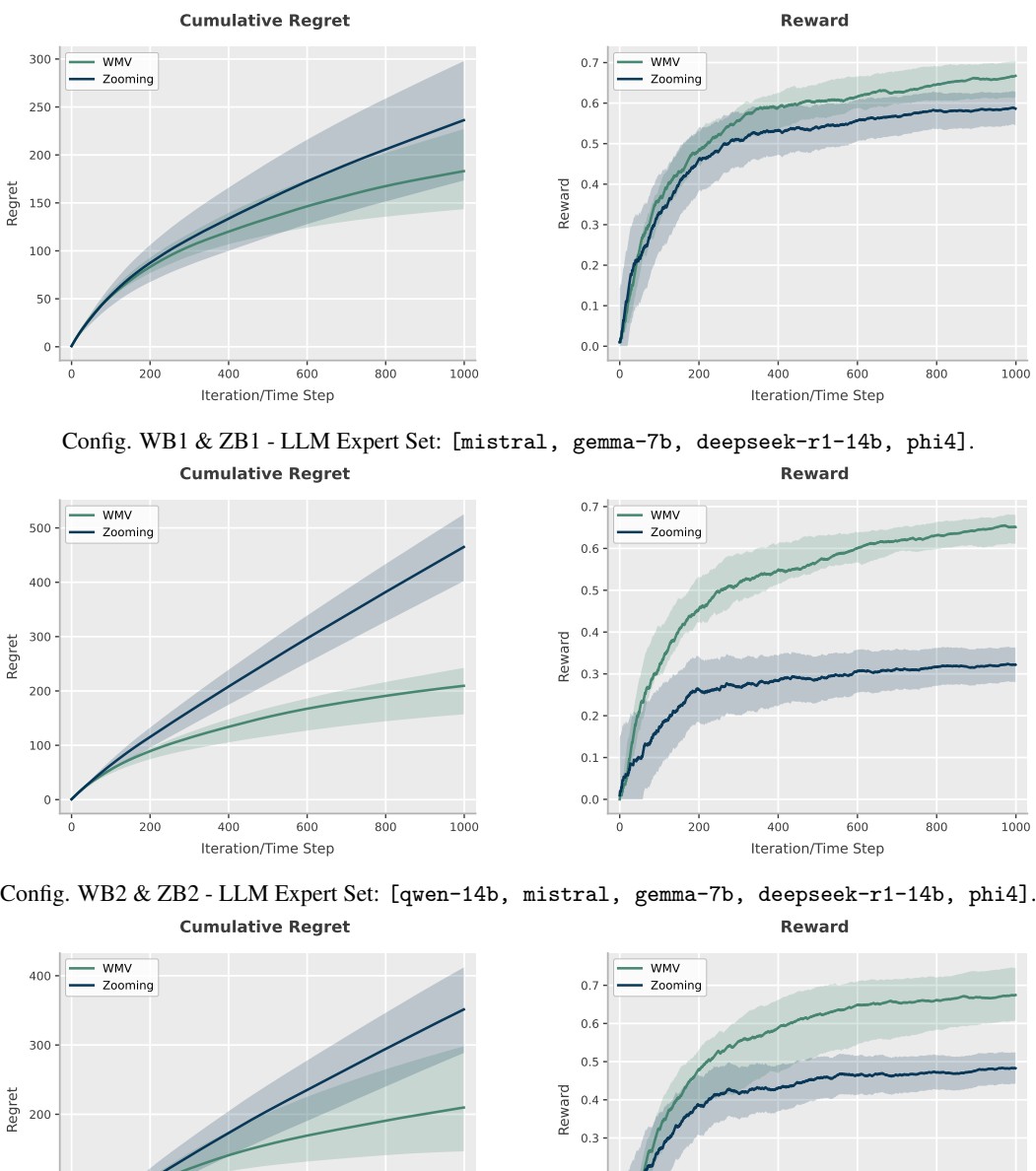

Config. WB1 & ZB1 - LLM Expert Set: [mistral, gemma-7b, deepseek-r1-14b, phi4].

Config. WB2 & ZB2 - LLM Expert Set: [qwen-14b, mistral, gemma-7b, deepseek-r1-14b, phi4].

Config. WB3 & ZB3 - LLM Expert Set: [samantha-mistral, qwen-14b, mistral-openorca, notus, aya, mistral,gemma-7b, deepseek-r1-14b, phi4].

Figure 10: Questions were sampled from the BoolQ dataset [38] . Mean values are calculated over 1,000 trials, with finite time horizon $T = 1,000$, with shaded regions representing confidence intervals of $\pm$UCB.

