# OpenReview forum: "Online Mixture of Experts: No-Regret Learning for Optimal Collective Decision-Making"
_NeurIPS.cc/2025/Conference — NeurIPS 2025 poster_

### Official Review · Reviewer_P2T7 · 2025-06-28

**Clarity:** 3
**Significance:** 3
**Originality:** 3
**Rating:** 5
**Confidence:** 3

**Summary:**

This paper presents two settings for decision making with expert sets. The first setting does this by finding the optimal egalitarian expert committee (OEC), i.e., a set of experts whose majority vote outperforms any individual expert. While the experts in the OEC all receive one vote, the second setting proposes finding the optimal weighted voting committee -- where the votes of the experts are weighted by some measure of their competencies. The paper provides optimal solutions in both scenarios when the competencies of the experts are known. It then discusses how these settings can be used for the online Mixture of Experts (MOE) problem, with a new notion of regret in terms of the best set of experts (as opposed to the best individual expert, in standard no regret algorithms for MOE). Two algorithms are proposed corresponding to each setting -- Successive Expert Elimination (SEE) where the OEC is found by eliminating experts with estimated low competencies and $\theta$-weighted Majority Voting ($\theta$-WMV) where the voting weights are found using optimistic estimates of competencies. Regret bounds for both the algorithms are derived. Finally, empirical results are presented for online fine-tuning of LLM experts on real datasets.

**Questions:**

Questions:
1. How should one choose between the two proposed algorithms in practice? Are there scenarios when one should be prefered over the other?
2. What is the difference between the "average scoring function" (Line 70) and the "scoring-aggregation function" (Line 81)?
3. Line 105: $X_i$ be created conditioned on $c_i(x)$, which is the output prediction. Should it instead not be conditioned on the prediction being correct / incorrect?
4. Should the condition on $\theta$ in Equation 2.9 be two-sided (as on line 176)?
5. Algorithm 1, line 5: "in descending order" of what?

Other minor comments:
1. The abstract reads "We propose three algorithms ..." (line 4) but I only see two algorithms discussed in the paper (SEE and $\theta$-WMV)
2. Line 44: "provides convergence best collective" -> "provides convergence to best ..."
3. Lines 47-48: "LLM's" -> "LLMs"
4. Lines 222-223: The sentence "Recent work on online ... is key" doesn't read right.
5. Line 235: "difference between to expected ..." doesn't read right.
6. Line 323: "amount of experts" -> "number of experts"

I am willing to increase my overall score if the questions/comments mentioned above are addressed and Section 4 (empirical study) is updated with enough detail (see weaknesses for specifics).

**Ethical Concerns:**

["NO or VERY MINOR ethics concerns only"]

**Final Justification:**

I find the new notion of regret with respect to a set of experts novel and very interesting. The proposed algorithms are explained in enough detail -- the regret bounds look good and the experiments demonstrate empirical usefulness.

With the assumption that the Experiments section will be updated with the details on the baselines, the typos resolved, and the plots updated for clarity, I think the changes would be minor and would like to recommend acceptance.

**Limitations:**

yes

**Quality:**

3

**Strengths And Weaknesses:**

**Strengths**

1. Clarity: The paper is well-written and easy to understand. The key insights for the SEE and $\theta$-WMV algorithms are also presented well. My only qualm is with the discussion of the experiments and empirical results, as mentioned in more detail under weaknesses.

2. Well-motivated problem: The problem with the existing notion of regret (w.r.t the best expert) for online Mixture of Experts is well-motivated and the two settings, the egalitarian expert committee and majority voting with a set of experts, are novel in the context of the MOE problem and are discussed in enough detail.

3. The proposed algorithms seem useful in practice for a large enough number of experts (up to 20 in the experiments, Table 1), while providing meaninful regret guarantees (theorems 1-2).

4. Moreover, the proposed algorithms report lower cumulative regret values when compared to baselines in both model-based (with Bernoulli experts) and data-derived settings (with LLM experts) (Table 1).

**Weaknesses**

1. Rushed experiments section: Section 4 (Empirical Study) could use some more details about
    - The baselines that the proposed algorithms are compared against: These are mentioned in Figure 3 and Table 1 with the results but not discussed at all in the main text (what are they? what the key differences when compared to the proposed algorithms?)
    - A brief discussion of the results: The results summary paragraph (Line 312 onwards) merely mentions the setup very briefly. The captions for Figure 3 and Table 1 do not discuss the implications of the results either. It would be very helpful to discuss the results in at least a couple of lines to give some more context. It would also be interesting to see what the egalitarian committees found empirically look like in the LLM setup (are the optimal committees found?) and what the weights found for the majority voting approach indicate (which LLMs are weighted higher and why?).

2. Figure 3, with the main comparison of cumulative regret across techniques, is very difficult to parse. The colors are too similar to be differentiated on the graph. I would suggest increasing the sizes of these graphs, using a more diverse color pallette for distinguishing algorithms, and briefly discussing the key takeaway in the caption.

Overall, I am pleased with the clarity of most of the main text (except the experiments section). I also find the expert set notion of regret and the use of top-k bandits with the two proposed algorithms in this setting original. I think that the problem is well-motivated and the application to MOE indicates that this work is indeed significant. This should encourage future work on no regret algorithms with expert sets and MOE. Please see the Questions section below for other questions and suggestions to further improve the paper.

---

> ### Author Rebuttal · Authors · 2025-07-30
>
> We thank the reviewer for your positive appraisal of our work - we provide a response to address some of the minor weaknesses and questions regarding our paper.
>
> ### Addressing Weaknesses:
>
> 1. **Empirical Experiments Clarification (Figure 3 and Table 1):** Due to space constraints, we had to shorten the exposition on describing the empirical performance of the models. In brief, we intend to demonstrate that our algorithms, when compared to their respective baselines, are superior in terms of regret minimization (lower is better).
>
> The two baselines we compared our problem setting against were:
>
> - **SEE vs. Combinatorial UCB.** We compare the SEE algorithm with the combinatorial UCB bandit algorithm. The combinatorial UCB is a general algorithm and considers all possible combinatorial combinations of expert committees - even possibly ones that are not likely to be optimal. SEE seeks to eliminate potential experts early, if they are highly unlikely to be optimal in a majority vote. Therefore, combinatorial UCB serves as a reasonable baseline, and the superior performance is likely due to SEE’s gradual elimination possibilities in the action space - making the algorithm more efficient.
>
> - **$\theta$-WMV vs Zooming (baseline):** The problem of online learning in repeated games where the reward function (in this case our scoring function) is in some general unknown format, is generally difficult. The zooming algorithm (Kleinberg et. al. 2008 [36]) appropriate for no-regret learning in repeated games for general continuous multi-dimension Lipschitz action spaces, accompanied with theoretical guarantees, provides a good general baseline. The zooming algorithm does not assume convexity of the reward function, and is one of the few algorithms which offer theoretical guarantees for this type of problem setting.
>
>     Due to zooming’s generality, its no-regret guarantee is weaker than the no-regret learning guarantee of our $\theta$-WMV (Alg. 2). This can be confirmed both theoretically and empirically, from our experiments. Feel free, also refer to the work of (Kleinberg et. al. 2008 [36]) for details. We use the zooming algorithm as a baseline to compare with our designed algorithm, $\theta$-WMV (Alg. 2), that addresses the online majority voting problem setting.
>
> 2. **Brief Discussion of Results:** Due to space constraints, we relegated some discussion of results and procedural details to  Appendix C. We provide the details on a larger ablation study, with the chain-of-thought prompting strategy, and LLM individual benchmarks. We believe the results are relatively easy to interpret, basically our algorithm is superior to the aforementioned baselines in terms of regret minimization. The differences are relatively stark in Fig. 3a and 3b), nevertheless we agree that we should produce a new plot for 3c) as too many overlapping shades are distorting the interpretation of the plots.
>
>     An clearer way to compare the plots in its current form would be “blue vs green”. For each dataset (eg. CQA, Bool etc.) we provide comparison between the **green (our paper’s algorithm)** and the **blue (the baselines)**. It is clear that our algorithms exhibit superior performance in terms of regret minimization. Further detailed views, and additional ablations, are provided in Appendix C.5.
>
> - **Updates to Plots:** Unfortunately due to the restrictions on providing PDF updates we could not upload new charts (unlike last year’s NeuRIPS), however, we hope the additional comprehensive experiments in Appendix C.5 notes show more clarity, and we will take your suggestion to polish up our plots to make them more interpretable - should we have a chance for make a camera ready version.
>
> ### Answers to Questions:
>
> **Q1. Conditions where one algorithm is preferred over another:** Lemma 2.3 posits that given the same committee of experts, utilizing the $\theta$-WMV (Alg. 2) to aggregate experts’ output will always yield a stronger result than using egalitarian voting. Therefore, if the computational resources are available we should use the $\theta$-WMV algorithm for better predictive accuracy. However, if there are limited computational resources, or if the goal is to learn an efficient representation of experts (i.e. a sparse set of experts) then perhaps SEE (Alg. 1) algorithm is a better choice to retain efficiency.
>
> **Q2: Average scoring function vs score-aggregation function:** The average scoring function is the expectation over the composition of the aggregation first $A(\cdot)$, followed by the scoring function $G(\cdot)$, according to distribution $P(x)$ over all contexts. The score-aggregation function is actually just the aggregation function $A(\cdot)$ (we will update our draft to avoid confusion).
>
> **Q3. W.r.t. To Eq. 2.3,**  the set $\mathcal{X}_i$ is defined as a subset of contexts for which a given expert i’s prediction, denoted by $c_i$, returns the correct answer. This allows us to partition the space of contexts depending on whether the expert was correct or not. Each expert has its own partitions, and subsequently allows us to rewrite the average scoring function in line 108.
>
> **Q4. Eq. 2.9 Two-sided:** To be consistent we should correct this to the same two sided condition as in line 276, however, in its current form it makes no difference, due to Lemma 2.2, because the maximizing solution can be replaced with $|| \theta || = 2Q$, so even if $|| \theta_1 || < 2Q$ it could be suboptimal, as there is an equivalent maximizing solution at $|| \theta || = 2Q$. (Nevertheless, we will make the minor update to be consistent.)
>
> **Q5. Descending Order (Alg. 1 Line 5):** This refers to the descending order of the expert’s competencies $p_i$. We perform the removal test by adding incrementally weaker experts to a shadow committee, and removing them entirely for future consideration should their theoretical contributions to the committee yield suboptimal results, under optimistic estimate
>
> **Typos:** Thank you for pointing out minor typos in our paper. We will definitely address them in an update to our paper.
>
> **Updates to Plots:** Unfortunately due to the restrictions on providing PDF updates we could not upload new charts (unlike last year’s NeuRIPS), however, we hope the additional comprehensive experiments in Appendix C.5 notes show more clarity, and we will take your suggestion to polish up our plots to make them more interpretable - should we have a chance for make a camera ready version.
>
> ---
>
> We thank the reviewer once again for your insightful comments and suggestions. Should further clarifications and/or discussions be required we would be happy to oblige. We hope the additional details and response will lead to an increase in overall assessment of our paper.

---

> > ### Comment · Reviewer_P2T7 · 2025-08-05
> >
> > Thank you for the response. I appreciate the detailed discussion on the experimental baselines here and would strongly encourage the authors to include some version of this in the Experiments section -- I understand that the Appendix might have more details but the main section should be fairly self-contained, with maybe some interesting ablations / additional results in the Appendix.
> >
> > Thank you for the answers to the questions as well.
> >
> > With the assumption that the Experiments section will be updated with the details on the baselines, the typos resolved, and the plots updated for clarity, I am happy to increase my score.
> >
> > I have no further questions.

---

### Official Review · Reviewer_BNid · 2025-07-01

**Clarity:** 3
**Significance:** 3
**Originality:** 3
**Rating:** 4
**Confidence:** 3

**Summary:**

This paper proposes a novel framework called Online Mixture of Experts (OMoE) for aggregating multiple expert predictions in real time to achieve superior collective decision-making, as opposed to selecting a single best expert. The authors introduce two main algorithms: (1) Successive Expert Elimination (SEE) for egalitarian voting, and (2) $\theta$-Weighted Majority Voting ($\theta$-WMV) for scenarios where experts should be weighted by predictive quality. The key contribution lies in combining techniques from online learning and social choice theory to provide provable no-regret guarantees. The framework is motivated by applications such as online aggregation of large language models (LLMs), where leveraging the "wisdom of the crowd" may outperform any single model.

**Questions:**

Please comment on or discuss the above weakness parts.

**Ethical Concerns:**

["NO or VERY MINOR ethics concerns only"]

**Final Justification:**

The authors solved most of my concerns regarding the MoE term, the efficiency, and the uncertainty of the scoring function. So I raise my score accordingly.

**Limitations:**

Yes.

**Paper Formatting Concerns:**

No.

**Quality:**

2

**Strengths And Weaknesses:**

**Strengths:**

- **Original Problem Formulation:**

    The paper targets a gap in standard online learning frameworks like EXP4, which are designed to identify the single best expert. In contrast, OMoE explicitly models **group decision quality** and allows the learner to benefit from aggregating multiple expert outputs. This is a compelling direction, especially in settings like LLM ensemble inference.

- **Sound Algorithm Design and Analysis:**
    - **SEE** leverages a bandit-based successive elimination strategy for pruning low-performing experts under an egalitarian majority-voting regime.
    - **$\theta$-WMV** introduces a more expressive model where experts are weighted by their predictive power. The use of a UCB framework with mixed-integer programming to compute optimal voting weights is clever, albeit potentially computationally intensive.

        Both algorithms are analyzed rigorously, with regret bounds stated clearly and supported by PAC-style analysis.

- **Bridging Online Learning and Social Choice Theory:**

    The connection to optimal voting rules and top-K ordinal aggregation offers an intellectually satisfying perspective and could inspire future work across disciplines.


**Weaknesses:**

1. **Forced Fit with MoE Terminology:**

    Although framed as a mixture of experts problem, the setting is arguably closer to **ensemble LLM decision aggregation** than traditional MoE architectures (which involve conditional routing and sparsity). The terminology may overpromise or mislead readers familiar with MoE in deep learning.

2. **High Inference Overhead:**

    Since the proposed algorithms require querying all (or most) experts at each round, the approach may be **infeasible in latency- or cost-sensitive settings**, particularly when each expert is a large model like an LLM. This undermines the practicality of the method in online LLM inference tasks.

3. **Weak Modeling of Real-World Feedback:**

    The formulation assumes access to a scoring function GG that reflects correctness of the experts’ outputs. However, in real-world LLM applications (e.g., open-ended QA), determining whether an output is "correct" is often **ambiguous or costly**. This introduces challenges in reward signal design, supervision cost, and label fidelity.

4. **Catalogue Assumption Limits Generality:**

    The reliance on a predefined catalogue of possible outputs restricts scalability to **structured prediction tasks** with limited output space. In more realistic NLP tasks—like summarization, image captioning, or generative QA—this assumption is too restrictive and may lead to poor performance or inapplicability.


**Writing and Presentation Issues:**

- The **abstract mentions three algorithms**, though only two are actually proposed. This may confuse readers and should be corrected.
- The notation is at times **inconsistent or overloaded**, e.g., the symbol $P$ appears with different meanings in different parts of the paper. A careful notation pass is recommended.

---

> ### Author Rebuttal · Authors · 2025-07-30
>
> We sincerely appreciate the reviewer’s diligent review and valuable insights. In response to the points raised, we aim to provide further clarification to address their concerns.
>
> 1. When it comes to multi-agent learning, we believe the **term Mixture of Experts** can capture a wide variety of meanings. The first time this terminology was introduced was in (Nowland and Hinton 1990 [3]), and it was later adopted in conditional routing systems introducing sparsity (Anthony et al. 2024 [4], Shazeer et al. 2017 [5], Dai et al. 2024 [6]), etc. We refer to MoE in its more native introductory form, as in (Nowland and Hinton 1990 [3]). Specifically, we address the online learning problem for MoEs, whereas [4,5,6] address the offline learning problem and, most critically, do not provide theoretical guarantees for their methods. If the terminology is one of the major deficits of our paper, we’d be happy to reword it to “ensemble model aggregation” or “ensemble learning” with LLM applications, as opposed to MoE—nevertheless, we do not see this as a major impasse to either the theory or application of our paper.
>
> 2. With respect to **sparsity and high inference overhead**, in fact, our current algorithms (SEE and $\theta$-OMV) do accommodate model sparsity by targeting a specific number of experts used at any given time. This reduces overhead by limiting the number of experts employed to reduce or regulate computational overhead. We have already included an analysis of this mode of operation and proved in Corollary 3.1 that this increases the complexity of no-regret learning by a factor of $N/m$, while retaining the exact complexity with respect to $T$, or the number of samples.
>
> - Please refer to the Target-$m$ Online Majority Expert Voting algorithm and detailed analysis in A.11.
>
> 3. **Open Ended QA:** We do not focus on the problem setting where open-ended questions or RL alignment problems exist in general. Although we recognize that RL alignment, robust reward design, and open-ended QA pose rich and interesting questions in the RL space, we are focused on our core problem and seek to provide rigorous, provable guarantees.
> Saliently, our **core problem setting deals with epistemological ensemble learning**—with extensions to Condorcet’s jury theorem in an online MoE setup. We provide a discussion of this in lines 92–97 of our paper. We believe that this contribution already underscores valuable insights and applications for both the online learning and social choice communities—as well as provides new algorithms for ensemble learning in application.
>
> 4. **Our framework is designed for structured prediction tasks**. However, there is no specific limitation regarding the cardinality of the output space. We make a deliberate choice to address the problem of majority voting for multi-class prediction accuracy under expert competence uncertainty—a problem that is non-trivial and largely unsolved.
>
> Overall, our paper aims first to provide provable guarantees for a well-defined problem in online MoE (or ensemble learning) via majority voting. We provide rigorous theoretical guarantees, accompanied by avenues for applications in MoE LLMs. Although several heuristic-based and empirically designed algorithms do exist, provable solutions to this current MoE problem do not exist, to the best of our knowledge. We hope this lends additional appreciation for our current paper.
>
> Furthermore, thank you for bring up minor inconsistencies with our notation and typos, we would be happy to fix this up in an update to our draft.
>
> ---
> We would be glad to offer further clarification if needed and hope our response will lead to an improved assessment of our paper.

---

> > ### Comment · Reviewer_BNid · 2025-08-03
> >
> > Thanks for the detailed feedback. I am generally satisfied with their responses to points 1, 2, and 4.
> >
> > Regarding point 3, I would like to clarify my original concern. I did not mean to suggest that an Open QA task is required. Rather, my point is that in practical scenarios, the exact score of user feedback is often unavailable or noisy due to environmental uncertainty or human bias. My question is whether the current framework can accommodate such randomness or bias in the scoring function. Would this require significant modifications to the proposed approach?

---

> > > ### Author Response · Authors · 2025-08-04
> > > **Noisy and Missing Feedback in Expert-Based Decision-Making**
> > >
> > > Note that our framework—like the learning-from-expert-advice setting (Cesa-Bianchi and Lugosi 2006 [9], specifically Ch. 5–6)—relies on receiving feedback from experts to evaluate them and improve the overall decision-making. We break down our problem into two scenarios: a) feedback is unavailable, and b) feedback is noisy.
> > >
> > > a) **If user feedback is unavailable**, this means the model would require more samples until it converges to the optimal solution. Suppose user feedback is unavailable $p_x$ percent of the time; then, our expected number of samples would inevitably have to increase to obtain the expected number of trials necessary for $T$ plays. This simply requires proportionally longer interaction periods with the experts to obtain the same theoretical guarantees (i.e., if the scoring function is unavailable $p_x$ of the time, then on expectation we would have to draw $T p_x$ more samples to obtain the same guarantees as originally $T$). Therefore, our method remains applicable in general.
> > >
> > > b) **Noisy user feedback.** The no-regret guarantees are measured with respect to a ground-truth scoring function $G(\cdot)$. Accounting for noise in the observed feedback, it is safe to assume that the noisy feedback is from a single user who, with probability $p_x$, mislabels an output as correct or incorrect. To model this noise, suppose we have a worst-case imperfect user feedback (scoring function) $G'$. Next, we introduce a *swap function* $g(\cdot)$ that applies a randomized perturbation to the expert predictions. Specifically, $g(c(x))$ swaps any correct prediction $c(x)$ with a uniformly random alternative class with probability $p_x$ (simulating feedback noise)—this serves as worst-case noise (we do not assume noise improves predictions). The resulting noisy feedback leads to adjusted competencies $\tilde{p}_i$, and we can establish equivalent expressions:
> > >
> > > $$
> > > \tilde{p}_i = \int_x G' (c_i(x)) dP(x) = \int_x G (g \circ c_i(x)) \, dP(x).
> > > $$
> > >
> > > Here, the scenario of noisy user feedback is expressed equivalently as a scenario under perfect feedback with a noisy swapping function $g(\cdot)$, preserving the structure of the analysis for Theorems 1 and 2. It is worth noting that, since we are introducing noise to a perfect scoring function, any type of noise would only reduce the competencies of the experts in this representation.
> > >
> > > Even with noisy feedback, the algorithms would run as-is, without modifications. The worst-case theoretical guarantees would still hold, with some caveats.
> > >
> > > > **Theorem 1** still holds with the adjusted competencies $\tilde{p}_i$, as the noise is now quantified as part of the experts' competencies. However, we must compute $\tilde{\epsilon}^2$ (the minimum competency gap between two experts) based on the adjusted competencies $\tilde{p}_i$. There is a possibility that $\tilde{\epsilon}^2$ becomes smaller, which would increase the regret guarantee by a constant scaling factor.
> > >
> > > > **Theorem 2** remains unchanged, as the regret guarantees are independent of the expert competencies $p_i$. To provide intuitive insight, for $\theta$-OMV, what matters more is a ranking over the experts' competencies—since noise reduces all experts' competencies equally, their relative rankings remain the same.
> > >
> > > **To summarize**, adding user feedback noise is worst-case equivalent to adding a perturbing swap function for the experts' outputs under true feedback. The algorithms are unchanged, and the theoretical regret guarantees remain largely the same due - but the introduction of noise could diminish regret minimization performance by a constant factor.
> > >
> > > ---
> > > We hope this clarifies the concerns you’ve raised, and we would be happy to elaborate further where needed.

---

> > > > ### Comment · Reviewer_BNid · 2025-08-04
> > > >
> > > > Thanks for addressing point 3. I have no further questions at this time and will determine whether to adjust my score shortly.

---

### Official Review · Reviewer_HeLJ · 2025-07-03

**Clarity:** 3
**Significance:** 2
**Originality:** 3
**Rating:** 3
**Confidence:** 3

**Summary:**

This paper proposes an online learning framework for expert aggregation via majority or weighted voting, aiming to outperform single expert through collective decision-making. It introduces two algorithms, Successive Expert Elimination (SEE) and $\theta$-Weighted Majority Voting ($\theta$-WMV), with theoretical regret guarantees. Experiments on synthetic data and multi-LLM tasks demonstrate improved performance over baselines.

**Questions:**

1) In Section 2 (line 66), the scoring function is defined independently of the context $x$. Does it imply the optimal aggregated output is the same across all queries? This is counterintuitive in settings like LLMs, where the optimal outputs should be context-dependent.

2) How is the scoring function defined in practice, particularly for open-ended tasks?

**Ethical Concerns:**

["NO or VERY MINOR ethics concerns only"]

**Final Justification:**

After multiple rounds of discussion during the rebuttal phase, I remain conservative on this submission. As noted previously, my main concerns are: (1) the lack of references on online MoE or online ensemble learning, (2) the need for a more careful comparison with CMAB methods, and (3) the more critical issue of the context-aware scoring function.

For the first two points, I believe the authors should strengthen the literature review and provide a more thorough comparison with existing work. Regarding the third point, after several rounds of questions and clarifications, the authors eventually acknowledged that their original scoring function definition is not suitable for the LLM setting. While I accept that the theoretical guarantees may not be significantly affected, the scoring function is a foundational component of their method. Given its central role, a careful re-examination is necessary.

**Limitations:**

Yes.

**Quality:**

2

**Strengths And Weaknesses:**

Strengths
1) The paper introduces a new perspective on expert selection via voting-based aggregation, moving beyond traditional online single expert selection.
2) The proposed algorithms are accompanied by formal regret analyses.

Weaknesses
1) The problem studied is not aligned with standard Mixture-of-Experts (MoE) models used in modern LLMs (e.g., DeepSeek-MoE, Mixtral). The paper’s claim that MoE selects only a single expert is inaccurate. This mismatch could mislead readers and should be explicitly clarified.
2) The citations for "online MoE" primarily cover general bandit algorithms (e.g., EXP4), without discussing more directly relevant work such as combinatorial bandits, Top-K selection, or their applications to expert or LLM ensemble selection, which makes it difficult to assess the significance.
3) While theoretically sound, both proposed algorithms adapt standard techniques like UCB-based elimination without introducing fundamentally new algorithmic ideas.
4) The experiments are restricted to synthetic Bernoulli simulations and simple multi-LLM voting tasks. The rationale for including baselines like Zooming and “Comb. UCB” (not properly referenced) is unclear. The empirical section does not thoroughly analyze why the proposed methods outperform these baselines.

---

> ### Author Rebuttal · Authors · 2025-07-30
>
> 1. To clarify, we do not claim that the current implementations of MoE are limited to selecting only a single expert, with reference to [4, 5, 6, 7, 8]. For example the work in (Shazeer et al 2017 [5]) selects the optimal 2 experts. We simply state that the MoE architecture is generally used in LLM inference, and that traditionally these models are learned offline via supervised learning (lines 24-26). Furthermore, the vast majority of these problem settings (whether its online, or offline learning), are not accompanied by theoretical guarantees.
>
> - However, in the context of online learning the vast expanse of scientific literature, via algorithms, such as EXP4 [13, 14, 9], are theoretically guaranteed to converge only to the best single expert, such methods are bound to the performance of the best expert. In this work, based on the classical result on social choice models and voting theory, we consider a majority or weighted voting in a sub-committee of experts which can result in even a better performance than the best expert.
>
> 2. We assert that a significant variety of bandit algorithms was mentioned in our paper. In lines 221-233, we provide an ample literature review on Top-K combinatorial bandits, as well as other methods to address our problem (eg. submodular bandits, satisficing bandits, duelling bandits etc.) . Further, in lines 179 -182, we discuss the combinatorial complexity of solving our online problem. In fact, the SEE method we propose uses strategies devised from Top-K combinatorial bandits. We would like to understand what exactly the reviewer means when by “difficulty asses the significance” and also specifically where the LLM ensemble selection is unclear per se.
>
> 3. A natural approach to decision-making problems is to frame them as variants of the multi-armed bandit framework. By adapting established algorithms—such as UCB or Thompson Sampling—one can then derive tailored solutions for specialized settings (e.g. causal bandits, or top-k ranking). Consequently, the key innovation in such work—mirroring our own—resides not only in the problem formulation but also in the detailed analysis of the adapted bandit algorithm.
>
> - To reiterate, our online algorithms (Alg 1 and Alg 2) converge to the best optimal subset of experts, unlike prior work which only guarantees convergence to the best single or top-K experts [9, 10, 11, 12, 13,14]. Our no-regret guarantees required a nontrivial and carefully constructed analysis (see Appendix A.7, A.10), as optimizing for the best voting committee—under both egalitarian and weighted settings—poses inherent complexities that go deep beyond conventional UCB analysis.
>
> 4. We provide discussion on Top-K combinatorial bandits earlier lines 171-183. To be more clear, Combinatorial UCB runs regular UCB, but with a combinatorial selection of arms rather than each arm independently. This is a principled no-regret learning algorithm, but suffers from combinatorial explosion of arms. Our successive elimination algorithm does not exhibit this characteristic - and therefore would exhibit improved regret minimization advantage.
>
> - The zooming algorithm (Kleinberg et. al. 2008 [36]) is used for no-regret learning in repeated games for general continuous multi-dimension Lipschitz action spaces. The zooming algorithm is a general solution to the Lipschitz bandit problem in continuous action space setting, with a general non-convex reward function. Notably, the zooming algorithm is one of the few scarce algorithms where no-regret guarantees exist. Our algorithm leverages additional information from the online majority voting problem setting - because of this we expect it to perform better than the zooming algorithm, when used as a baseline.
>
> ### Addressing the specific questions:
>
> 1. **Contextual information** is passed into each expert (predictor) and the predictions are aggregated parametrically - given the structured approach via majority voting. It is important to note that the scoring function evaluates the accuracy or quality of the prediction. As indicated in Eq. 2.1, the prediction itself may be context-dependent. In the current implementation, contextual information is used to learn the optimal parameters for expert aggregation. As discussed in lines 298-303, we draw tasks (contexts) from a diverse set of task domains (e.g. GSM8K, CommonsenseQA, BoolQ etc.) - we provide more detailed descriptions in the Appendix C. This is a probability distribution, where certain types of contexts are emitted from. Our goal is to design an algorithm that learns the optimal parameters for the aggregation algorithm.
>
> - Not having the context be directly visible to the expert aggregation algorithm presents some advantages in the context of privacy in machine learning. In the LLM setting, the current algorithm(s) do not need to observe the input of the prompts, and in principle, can receive an one-way encrypted message from each expert. Therefore our methods provide meaningful applications when context (prompt) privacy is of a concern.
>
> - We do acknowledge that a per-context MoE predictor could align more with the standard LLM based literature. Possibly, this is interesting for future work, but deserves a separate treatment using other learning algorithms. Our paper could provide insights for researchers who wish to pursue this direction of research.
>
> 2. **The scoring function** used is the label of the extracted task (which could also be one-way encrypted). As discussed in lines 298–303, each database—GSM8K [33] (mathematical reasoning), CommonsenseQA [34] (implicit knowledge), and BoolQ [35]—contains labeled answers for each task (math, commonsense reasoning, etc.). We use the labels for each task in the dataset to provide a score for the LLM’s responses. More detailed descriptions of the prompting strategy, baseline LLM performances, and descriptions of each dataset are provided in Appendices C.1–C.4.
>
> ---
> We are grateful to the reviewer for their thoughtful comments and for recognizing the contributions of our work. If further clarification would be helpful, we are happy to provide it. We would appreciate it if our response leads to a more favorable reassessment.

---

> ### Comment · Reviewer_HeLJ · 2025-08-06
>
> Thank you for the detailed response. I appreciate the clarifications and the additional discussions. I still have a few remaining concerns regarding the literature, significance, and the scoring function design.
>
> 1) The citations for online MoE (e.g., [9, 14]) are general-purpose online learning textbooks and not specific to online MoE settings. Including more relevant work on online ensemble learning, expert aggregation, or top-K voting would better situate the contribution within the context of online LLM or ensemble selection.
>
> 2) I appreciate the discussion on combinatorial UCB, but would like to clarify my concern. Modern CMAB algorithms (e.g., Chen et al. 2013; Wang & Chen 2017; Merlis & Mannor 2019) are specifically designed to mitigate combinatorial explosion. Thus, I disagree with the claim that combinatorial UCB inherently suffers from this issue. In fact, many CMAB formulations can naturally model subcommittee selection, even without explicitly incorporating voting. Without a direct comparison or clearer justification of novelty beyond existing CMAB techniques, it remains difficult to assess the significance of the proposed methods.
>
> 3) I remain unconvinced by the context-independence assumption in the scoring function. My original question—if the score of a prediction $c \in C$ is independent of the query (or context) $x$, does that imply the optimal aggregated output is the same across all queries?—was not directly addressed. In practice, LLM outputs are typically query-dependent, so using a fixed scoring function across all contexts seems counterintuitive.
>
> 4) (Minor) I agree with Reviewer BNid that the problem setting is more aligned with ensemble-based LLM decision aggregation than with traditional MoE architectures. It may be helpful to revise the title or framing to better reflect the focus on voting-based expert aggregation.
>
> 5) (Minor) I understand that for classification-style tasks (e.g., GSM8K, BoolQ), ground-truth labels serve as the scoring signal. However, in open-ended tasks where no labels are available, how might the current scoring framework be extended? While this is not a major concern, I think it is a meaningful direction for future work.
>
> (Chen et al. 2013) Chen, Wei, Yajun Wang, and Yang Yuan. "Combinatorial multi-armed bandit: General framework and applications." In International conference on machine learning, pp. 151-159. PMLR, 2013.
>
> (Wang & Chen 2017) Wang, Qinshi, and Wei Chen. "Improving regret bounds for combinatorial semi-bandits with probabilistically triggered arms and its applications." Advances in Neural Information Processing Systems 30 (2017).
>
> (Merlis & Mannor 2019) Merlis, Nadav, and Shie Mannor. "Batch-size independent regret bounds for the combinatorial multi-armed bandit problem." In Conference on Learning Theory, pp. 2465-2489. PMLR, 2019.

---

> > ### Author Response · Authors · 2025-08-07
> >
> > 1. **Literature Review:** We highlight that we have already cited many relevant topics in our paper. For LLM and MoE see refs. [1 to 8], ensemble learning see refs. [9 to 13]. We also included several references to bandit literature see Sec. 3, refs. [23 to 32], including Top-K combinatorial bandit in [30,31,32]. While we have already provided these references in our current paper, we plan to incorporate additional relevant references, in line with your suggestion.
> >
> > 2. **CMAB Methods and the (α,β)-Oracle Limitation** The suggested works concerning CMAB (Wang & Chen 2017; Merlis & Mannor 2019; Chen et al. 2013) do indeed illustrate innovative and powerful methods to combat combinatorial explosion in a general CMAB environment - we are glad you brought this up.
> >
> > - However, per one critical subtle aspect. **The suggested CMAB works all require the existence of an (α,β)-oracle, an assumption which is not theoretically valid in our problem setting**. These CMAB methods assume the learning algorithm has access to an offline (α,β)-approximation oracle where, for every parameter set, with probability β, it outputs a solution whose value is at least α portion of the max reward. For example, when the reward function is monotone and submodular, such an oracle can be defined with α=1-1/e, as described in (Merlis & Mannor 2019).
> >
> > - In our setting, the learner does not have access to such an (α,β)-oracle, as our reward function is not guaranteed to be submodular with respect to the composition of the set-based action space. For instance, when additional arms are pulled w.r.t. an existing subset of arms (or when additional experts are added to an existing committee), this does not always result in a submodular increase in reward in our majority expert voting setting. This is a key subtlety that differentiates our SEE algorithm and the CMAB algorithms you’ve provided.
> >
> > - Moreover, the absence of an (α,β)-oracle in the suggested CMAB papers precludes us from running these algorithms as benchmarks in our setting, because the oracle is a key step in their algorithmic procedures (refer to Line 6 in Alg. 1 of Merlis & Mannor 2019 and Line 7 in Alg. 1 of Chen et al. 2013).
> >
> > 3. **Aggregated Output and Scoring Mechanism:** Regarding concerns about aggregated output, the optimal scoring output should depend on the nature of the query—some of which have ground-truth answers. This is referred to as “truth-tracking” in epistemological social choice and we discuss this further in lines 86–97, with supporting references [15–19].
> >
> > - In our experiments, the optimal score occurs when the aggregated prediction matches the true label from set $\mathcal{C}$ (e.g., $\{\text{dog}, \text{cat}\}$ in the set of all pets). Here, we assume a fixed set of true labels in $\mathcal{C}$, and the score measures agreement between the aggregated prediction and the true label. Since human labeling is costly, we designed a static scoring function as a practical way to measure performance in terms of regret minimization - but importantly doesn’t hinder the extension to context dependent scoring.
> >
> > - **Making the scoring function context-dependent (i.e. varying with input $x$) would not affect the theoretical guarantees in our work.** Referring to Eq. 2.2 in our paper, the original scoring function $G(\cdot)$ is context-independent. If $G(\cdot)$ is modified to be context-dependent, we would adjust Eq. 2.2 to accept two arguments (instead of one),
> >
> > $$
> > p_i:= \int_{x} G \left(c_i(x), x \right) \, dP(x).
> > $$
> >
> > This change would accommodate:
> > - Subjective, context-dependent queries (e.g., "What is your favorite color?")
> > - As well as, objective queries with absolute answers (e.g., "What is the capital of France?")
> >
> > Crucially, **this modification does not impact the computation of expert competency** $p_i$, preserving all theoretical guarantees under the new feedback mechanism. Practically, the only difference lies in who provides the labels (e.g., ground-truth annotations vs. human LLM interactions). We think this is a good point you’ve raised, and we’d be happy to discuss this item in a revision to our paper.
> >
> > **Minor items:**
> >
> > 4. As also replied for reviewer BNid If the terminology is one of the major deficits of our paper, we’d be happy to reword it to “ensemble model aggregation” or “ensemble learning” with LLM applications, as opposed to MoE—nevertheless, we do not see this as a major impasse to neither the theory or application of our work.
> > 5. Our framework can be extended to incorporate user feedback in open-ended tasks by defining the scoring function, $G(\cdot)$, as a context-dependent mapping from $\mathcal{C} \times \mathcal{X}$ to $\mathbb{R}$. This allows $G(\cdot)$ to evaluate outputs in a context-aware manner (consistent with the modification in point 3). This extension is well-suited, as user-provided labels arrive online in real time, per context. So long as the feedback-giver doesn't change, our existing theoretical guarantees continue to hold.

---

> > > ### Comment · Reviewer_HeLJ · 2025-08-07
> > >
> > > Thank you for the clarifications. I appreciate the thoughtful responses. I still have some remaining concerns.
> > >
> > > 1) To clarify, my concern is specifically about the lack of references on *online* MoE or *online* ensemble learning. The cited works in your introduction (e.g., [9], [13], [14]) and Section 3 primarily cover general bandit algorithms, but do not address settings where *expert selection* or *aggregation* is performed in an online manner. In particular, references [9–13] focus on *offline* or *static* MoE, and [23–32] are standard bandit papers without MoE-specific considerations. I recommend including more relevant literature on online ensemble learning or online expert aggregation, which would better contextualize your contribution.
> > >
> > > 2) In my view, the reliance on an offline oracle is not the fundamental limitation in adapting CMAB methods (submodularity is not a strict requirement). If the underlying offline optimization is intractable, then it is unlikely that any efficient online algorithm can be designed under uncertainty. More relevant challenges may arise from the availability of *semi-bandit feedback* and whether the reward function satisfies the *bounded smoothness condition* (Wang & Chen 2017), both of which are key to regret analysis in CMAB literature. I would encourage the authors to think about these aspects more carefully to clarify the distinctions and better justify the necessity of a new framework.
> > >
> > > 3) I remain unconvinced by the justification for a context-independent scoring function, particularly given the emphasis on LLM applications of this paper. The analogy to “truth-tracking” in epistemic social choice may not be appropriate here. In practice, LLM responses are highly context-sensitive, and so is the notion of correctness.\
> > > Could you please clarify the following with a concrete example? Suppose the correct answer to query 1 is “dog” and the correct answer to query 2 is “cat.” Then, the optimal aggregated prediction should be (“dog”, “cat”). However, if your scoring function is context-independent, does that mean only one of them (e.g., “dog” or “cat”) can be optimal across all queries? In your experiments, is it the case that all queries share the same ground-truth label (e.g., “dog”), making the context-independent assumption effectively valid in that case? If so, I think this should be clearly stated, and the limitations acknowledged.\
> > > Moreover, if a context-dependent scoring function can be incorporated without affecting the theoretical guarantees, then it would be valuable to actually include it in the algorithm and regret definition. Otherwise, the current regret metric may misalign with real-world LLM tasks, where context plays a central role in evaluating correctness.

---

> > > > ### Author Response · Authors · 2025-08-08
> > > >
> > > > We thank the reviewer for continuing to engage with our work with thoughtful feedback.
> > > >
> > > > 1. Frankly, to the best of our knowledge, there is very sparse literature providing formal guarantees on the online Mixture of Experts (MoE) setting. With respect to the literature on online expert selection, we refer to [9], which discusses several "learning from expert advice" algorithms, such as EXP4. Most literature, even recent ones, dealing with this "learning from expert advice" framework converges to the performance of the “best single expert”—but we aim to converge to the “best committee of experts” —which is the focal point of our paper (see also response to reviewer nmuN). We would be happy to provide further discussion of additional relevant literature on online MoE in an updated revision of our paper.
> > > >
> > > > 2. As we’ve cited [refs. 3–8], many offline MoE algorithms do not use majority voting as their aggregation method. In some cases, these offline MoE problems can be intractable (they may have increased the complexity of their aggregation model, and relinquished theoretical guarantees). Thus, in those cases, a learned representation serves as the best approximation of their context-to-output strategy, and model performance is based primarily on empirical assessment.
> > > >
> > > > - Regarding our online MoE algorithm, Successive Expert Elimination (SEE), the **offline version of the MoE optimization problem is tractable**. When given the batch offline data (query + label) beforehand, the solver can produce a solution with O(N) complexity, where N is the number of experts. Specifically, this is the solution to Eq. 2.9, where θ ∈ {0, 1} (i.e., inclusion or exclusion) for the egalitarian committee. We solve for the assignment of θ that maximizes historical performance by checking the inclusion/exclusion of experts ranked by their empirical competencies. Given the structure of the problem, we need not explore the entire (2^N) assignment space; instead, we start by including the best expert based on the highest competency estimate and subsequently add weaker experts in descending order, computing $P_{maj}$ in Eq. 2.8.
> > > >
> > > > - As you correctly pointed out, the bounded smoothness condition from (Wang & Chen, 2017) is satisfied in our problem setting. However, without a submodular reward function with respect to the composition of the set-based action space, we do not have an (α,β)-oracle. Thus, we cannot execute the key steps in these CMAB algorithms (refer to Line 6 in Alg. 1 of Wang & Chen, 2017), nor do we necessarily inherit their guarantees. We emphasize that **these CMAB methods (Wang & Chen, 2017) cannot be applied to our MoE setting out-of-the-box.**
> > > >
> > > > - To shed more light, the CMAB approach from (Wang & Chen, 2017) is unlikely to yield superior performance to our specially tailored online majority voting algorithm because it does not exploit problem-specific information to simplify the problem. Our problem is more akin to the Top-K Combinatorial Bandit setup [refs. 30–32], particularly the "successive elimination algorithm," which we discuss in our paper (lines 254–265). SEE is based on the principles of successive elimination, adapted with modifications specific to the properties of majority voting.
> > > >
> > > > - Nevertheless, we agree that providing additional discussion of (Wang & Chen, 2017; Merlis & Mannor, 2019; Chen et al., 2013) would benefit our paper, in line with your suggestions.
> > > >
> > > > 3. If we understand correctly, in the example you presented (where "dog" is optimal for query 1 and "cat" is optimal for query 2), the scoring function, G(·), will reflect this ground truth. It is **not the case that all queries share the same ground-truth label** (e.g., "dog" across all queries); this is not what we mean by context-independence. What we mean is that the score given to the aggregation of outputs is independent of external factors—not that the output must be the same for all queries (e.g., if the context provides unrelated information about the weather, it would not change the fact that the optimal query response is "dog"). **It is not the case that all queries share the same ground-truth label**, each context could have a different label - as is reflected in the datasets (i.e. GSM8K, BoolQ etc.). In line with your suggestion, we agree that updating our paper to clarify this point would be very beneficial.
> > > >
> > > > - We also believe incorporating a discussion on the context-dependent scoring function—which do not affect the theoretical guarantees—would be a prudent addition to our paper, and we agree that it plays a central role in human-LLM interaction. Model performance relies on a label provided either by the ground truth or a human labeller, and in either case, our current framework applies.

---

> > > > > ### Comment · Reviewer_HeLJ · 2025-08-08
> > > > >
> > > > > Thank you for the follow-up clarification.
> > > > >
> > > > > Referring to your point 3, if we look back at your definition of the scoring function $G: C \to \mathbb{R}$ in line 66, this mapping depends only on the prediction and not on the query (context). Strictly following this definition, the score assigned to a given prediction cannot vary across different queries.
> > > > >
> > > > > A more reasonable formulation—consistent with what you described in your earlier rebuttal—would be $G: C \times X \to \mathbb{R}$ (e.g., $G(c_i(x), x)$), so that the score can depend on both the prediction and the query. Without this, a static scoring function $G: C \to \mathbb{R}$ cannot correctly determine whether a prediction matches the ground-truth label for different queries.
> > > > >
> > > > > I am therefore unsure how the current context-independent scoring function operates in your setting. In your experiments, is it the case that any output in the true label set $C$ is assumed to have the optimal score? Or is there some implicit mechanism by which the context still influences scoring under the current definition?
> > > > >
> > > > > I appreciate your engagement in this discussion. I am particularly interested in this point as the scoring function is a core component of your method, and I want to ensure there is no mismatch or ambiguity for readers.

---

> ### Author Response · Authors · 2025-08-08
>
> Yes, the current expression in line 66 should be updated to reflect that $G: C \times X \mapsto \mathbb{R}$. Consequently, Eq. 2.2 should be revised to:
>
> $$
> p_i := \int_{x} G \left( c_i(x), x \right) dP(x).
> $$
>
> Similarly, Eq. 2.1 should be updated so that $G(\cdot)$ takes both $c$ and $x$ as arguments. This ensures that the score varies across queries, as as accurately reflected in our experiments. The current static scoring function Eq. 2.,2, does not reflect the actual operation of the scoring function, and it is a minor error of expression which will be corrected.
>
> In our experiments, for each context, the true label (i.e. correct answer) will receive an of score 1 from the scoring function - as it applies to the datasets (GSM8K, BoolQ etc.). Beyond this, there is no additional implicit mechanism by which the context still influences scoring - perhaps that’s where some misunderstanding could have arisen.
>
> We will clarify these points in an update to our paper, and we thank the reviewer for bringing this to our attention.

---

### Official Review · Reviewer_nmuN · 2025-07-03

**Clarity:** 3
**Significance:** 2
**Originality:** 3
**Rating:** 4
**Confidence:** 2

**Summary:**

The paper is concerned with merging output of experts, which are typically LLMs, in the on-line mode. If I understood the setup correctly, the predictions are rankings on finite sets and we want to compete against optimal weighted committees of the experts. The paper proposes two algorithms, one working through expert elimination and assigning weights to experts. Regret bounds are proven for the algorithms. The papers refers to them as 'no-regret', which, I presume, stands to mean $o(T)$ regret. There is empirical evaluation using LLMs.

**Questions:**

Are we still talking about rankings or 0-1 predictions in Section 3? If it is about ranking, is the number of categories fixed or bounded? Do we count only one top vote from an expert, or do other choices matter?

My ultimate goal is to understand how the result of the paper compares with well-known Weighted Majority of Littlestone and Warmuth, 1994.

**Ethical Concerns:**

["NO or VERY MINOR ethics concerns only"]

**Final Justification:**

Upon reading the discussion, I am inclined to keep my assessment of the paper.

The authors' use of terminology seems awkward and causes some confusion (prediction with expert advice vs voting) but the result seems interesting.

**Limitations:**

Yes

**Quality:**

3

**Strengths And Weaknesses:**

This is an interesting and practical results.

It may be just me, but I found the organisation of the paper rather confusing. Section 2 is quite detailed, but as far as was able to ascertain, it deals with batch setup and was included mainly for motivation. The original contribution of the paper is in Section 3, and it is rather terse. I would like to clarify a few points and see a clearer comparison with the extensive machinery of prediction with expert advice.

---

> ### Author Rebuttal · Authors · 2025-07-30
>
> We thank the reviewer deeply for the time and effort taken to review our paper, and your overall positive appraisal.
>
> To clarify the questions, we build on the work of (Littlestone and Warmuth [13]) by proving finite-time regret bounds, whereas (Littlestone and Warmuth [13]) provide only an error guarantee. In this work, we do not consider the rank over experts’ predictions but rather their top prediction. Our work provides a finite-time regret guarantee. The successive elimination algorithm we employ is based on established successive elimination bandit algorithms from (Even-Dar et al. [30]) and (Rejwan and Mansour [32]).
>
> Furthermore, the guarantee from (Littlestone and Warmuth [13]) is **limited to binary predictions**, where voters can only produce one of two outputs ({0,1}). Similar work can also be found in (Ntizan and Paroush [21]). We expand on these ideas by **providing no-regret guarantees for an optimal committee of experts** that covers the space of multi-class (i.e., N) prediction outputs, not just binary outputs. We provide two solution strategies for this method: one based on successive elimination, in Section 3.1, which can be employed to target model efficiency by limiting the group of selected experts and another based on optimal weighted voting, in Section 3.2, which extracts the optimal majority voting weights (parameters) under imperfect information.
>
> ---
> We hope this clarifies your questions and thank you once again for the thoughtful feedback. We would be glad to provide further clarification if needed. If our response helps to address any remaining concerns, we would appreciate a adjustment of the overall assessment.

---

> > ### Comment · Reviewer_nmuN · 2025-08-06
> >
> > Thank you for the clarification. I see that the setup is quite different from Littlestone and Warmuth, 1994 and presumably there is no direct comparison. You aim to compete against committees rather than experts.

---

### Author Response · Authors · 2025-08-09
**Authors' Summary of Rebuttal and Key Contributions**

We thank the AC and reviewers sincerely for their valuable time and feedback provided during this review process.

---
To summarize the areas which were covered in the rebuttal:

1. **Optimal Majority Voting Committee:** To clarify the online learning objective (rev. nmuN), we build on the classical “weighted majority algorithm” from (Littlestone and Warmuth [13]) to extend it from the *binary decision problem* to the *multi-class classification problem*, via online majority voting. Our goal is to converge to the best majority voting committee of experts (which could be an egalitarian committee, or a weighted voting committee) while providing formal finite-time no-regret guarantees, requiring additional technical analysis and derivations.

2. **Scoring Function and Context-Dependence:** Reviewer HeLJ pointed out that the scoring function $G(\cdot)$, introduced in line 66 of our paper, appeared static across queries — which is unrealistic for LLMs. To address this point, we corrected notation to G(c(x), x), instead of G(x), ensuring ground-truth labels are context-aware. Further, we argue that this modification preserves the existing no-regret theoretical guarantees provided in the paper.

3. **Additional topics:** that were discussed include a clarification about the strength and suitability of the baseline bandit algorithms (rev. P2T7, HeLJ), the appropriateness of mixture-of-experts (MoE) terminology (rev. HeLJ, BNid), computational overhead (rev. BNid), and the interpretation of the empirical results (rev. P2T7), etc. — which can be found in the details of the author–reviewer discussion.

---
To highlight the contributions of our paper:

1. **Gap in Online Ensemble Learning:** Our paper addresses a key intersection area that combines the theoretical components of online learning with epistemological social choice — where in recent literature, few theoretical contributions have been made, if any. This provides value to the application and understanding of online ensemble learning and MoE models in general.

2. **Online MoE-LLMs:** As mentioned in our paper, lines 25–31, the MoE-LLM framework is often addressed as an offline learning problem (please see refs. [3 to 8]). Although offline MoE can be powerful, offline MoE-LLM literature focuses less on formal guarantees and more on practical performance driven by empirical results. In contrast, the online MoE algorithms we present are more natural for instances where user-feedback arrives sequentially — we seek to provide new principled learning methodologies for the online optimization of LLMs.

3. **No-Regret Guarantees:** We provide online MoE algorithms that are accompanied by formal no-regret guarantees. Our first algorithm, Successive Expert Elimination (SEE), extends the “successive elimination bandit algorithm” (Even-Dar et al. [30], Rejwan and Mansour [32]) to the egalitarian majority voting setting. The second algorithm, $\theta$-Weighted Majority Voting ($\theta$-WMV), extends UCB-based bandit algorithms to the setting of weighted majority voting. Leveraging problem-specific information, we construct a no-regret learning algorithm for repeated games in general continuous multi-dimensional action spaces. The guarantees were rigorously derived, where key technical challenges had to be overcome to ensure sublinear regret performance, aligning with the empirical results we presented.

---
We hope this summary offers a clear and concise clarification of the salient points. We once again thank the reviewers for their thoughtful feedback, which has enabled a constructive and productive discussion.

---

### Decision · Program_Chairs · 2025-09-17

**Decision:**

Accept (poster)

**Comment:**

The paper proposes a novel online learning framework, Online Mixture of Experts (OMoE), for collective decision-making through expert aggregation, particularly in settings involving large language models (LLMs). It introduces two algorithms: Successive Expert Elimination (SEE) for egalitarian majority voting and θ-Weighted Majority Voting (θ-WMV) for weighted voting based on expert competencies. The key contributions include rigorous no-regret guarantees for these algorithms under a new regret formulation—defined with respect to optimal expert committees rather than single best experts.

The reviewers appreciated the original formulation and theoretical grounding, highlighting that the approach extends classical online learning and voting mechanisms in meaningful ways. However, concerns were raised regarding: (1) the alignment of the problem with traditional MoE terminology (R2, R3); (2) incomplete or unclear experimental details and baseline descriptions (R4); (3) context-independence in the scoring function which seemed unrealistic for LLMs (R2); and (4) literature positioning especially in relation to CMAB and ensemble learning (R2).

Overall, I recommend acceptance. Despite some open concerns—mainly around scope (e.g., generalizability to broader NLP tasks) and literature framing—the paper presents a novel and well-argued contribution to the field of online expert aggregation. It offers both solid theoretical insights and promising empirical performance. Please incorporate the final suggestions of the reviewers in the camera-ready version as promised.